# Acute stress causes sex-specific changes to ventral subiculum synapses, circuitry, and anxiety-like behavior

Carley N. Miller [1], Yuan Li [2], Kevin T. Beier [2,3,4,5] & Jason Aoto [1]✉

Experiencing a single severe stressor is sufficient to drive sexually dimorphic psychiatric disease development. The ventral subiculum (vSUB) emerges as a site where stress may induce sexually dimorphic adaptations due to its sex-specific organization and pivotal role in stress integration. Using a 1 h acute restraint stress model in mice, we uncover that stress causes a net decrease in vSUB activity in females driven by adrenergic receptor signaling. By contrast, males exhibit a net increase in vSUB activity that is driven by corticosterone signaling. We further identified sexually dimorphic changes in vSUB output to the anterior bed nucleus of the stria terminalis and in anxiety-like behaviors in response to stress. These findings reveal striking changes in psychiatric disease-relevant brain regions and behavior following stress with sex-, cell-type, and synapse-specificity that contribute to our understanding of sexually dimorphic adaptations that may shape stress-related psychiatric disease risk.

Stress exposure is a prominent risk factor for developing psychiatric diseases, and stress-related psychiatric diseases are more common in females[1]. Starting in early adolescence, stress-related psychiatric diseases exhibit profound sex differences in prevalence and symptomology, making it critical to understand the mechanisms that shape sex-specific vulnerability[2,3]. Chronic stress is known to contribute to psychiatric disease development in both sexes[4] and causes synaptic adaptations within the ventral hippocampus (vHipp) in preclinical and clinical models[5,6]. Interestingly, severe acute stress is a critical risk factor for stress-related disorder development, which disproportionately affects females, intimating that there may be sex differences in how acute stress is encoded at synapses relevant to the pathogenesis of stress-related disorders[7–10]. However, the cellular mechanisms by which acute stress induces vulnerability to psychiatric disease remain obscure, and even less is understood regarding how acute stress impacts vHipp synapses while considering sex as a biological variable.

Excitatory/inhibitory (E/I) imbalance within vHipp has been firmly established in chronic stress and psychiatric disease models in both sexes[11], revealing altered synaptic plasticity at excitatory synapses and in the excitability of parvalbumin (PV) expressing inhibitory interneurons[12–15]. The ventral subiculum (vSUB) subregion of vHipp emerges as a promising site of sex-specific disease pathogenesis due to its basal sexually dimorphic local circuit organization and its regulation of stress integration and anxiety-like behavior via projections to the anterior bed nucleus of the stria terminalis (aBNST)[16–19]. Although vSUB is sexually dimorphic and resides at the intersection of stress regulation and anxiety-like behavior, vSUB is grossly understudied.

Distinct from other vHipp subregions, vSUB primarily consists of pyramidal excitatory cells that are classified based on their firing pattern as burst (BS) or regular spiking (RS). BS and RS cells are accepted to be functionally nonoverlapping, especially in psychiatric disease pathogenesis[19–24] and exhibit basal sex differences in PV inhibitory synaptic strength and connectivity[19]. These cells express receptors targeted by the main stress effector pathways, the adrenergic system and hypothalamus-pituitary-adrenal gland axis (HPA)[25–28]. While the adrenergic system and HPA activity are vital to respond appropriately to environmental stressors, dysfunction of either system can contribute to the development of psychiatric disease[29,30]. Together, the

[1]Department of Pharmacology, University of Colorado Anschutz Medical Campus, Aurora, CO 80045, USA. [2]Department of Physiology and Biophysics, University of California, Irvine, CA 92697, USA. [3]Department of Neurobiology and Behavior, University of California, Irvine, CA 92697, USA. [4]Department of Biomedical Engineering, University of California, Irvine, CA 92697, USA. [5]Department of Pharmaceutical Sciences, University of California, Irvine, CA 92697, USA. ✉e-mail: jason.aoto@cuanschutz.edu

inherent sexual dimorphism and stress responsivity of vSUB make this subregion well-positioned as a site of sex-specific stress integration and stress disorder pathogenesis; however, no studies have systematically examined the vSUB local circuit after acute stress with cell-type and sex specificity.

Here, using multidisciplinary approaches, we systematically examined the impact of 1 h acute restraint stress on the functional properties of vSUB synapses in ex vivo slices, vSUB activity in vivo, and anxiety-like behaviors in sexually mature adolescent female and male mice. Acute restraint stress induced sex-, cell-type-, and synapse-specific adaptations to vSUB principal cells that resulted in net inhibition of BS cells in females and net excitation of BS cells in males. Importantly, these sex-specific functional adaptations in response to stress were mechanistically distinct as adrenergic receptor signaling was necessary in females while corticosterone signaling was required in males. Moreover, our electrophysiological and in vivo $Ca^{2+}$ imaging identified sex-specific stress-induced changes in vSUB output to the aBNST. In parallel with these ex vivo and in vivo synaptic findings, stress produced robust anxiety-like behaviors in female, but not male mice, which provides the framework to interpret our findings. Finally, we report that the ex vivo and in vivo sex-specific adaptations to local vSUB circuitry findings in response to stress occur across a broad range of ages. Our results support the notion that the integration of acute stress is sex-specific in vSUB and represent initial steps to understand the mechanisms by which sex differences in stress-induced psychiatric disease prevalence, treatment responsivity, and symptomology may occur.

## Results

### Stress impairs vCA1-vSUB-BS excitatory synapses in females

To begin to assess the impact of acute restraint stress (referred to as "stress") on the vSUB local circuit, we first interrogated the strength of vCA1-BS and vCA1-RS excitatory synapses 24 h after a 1 h stress paradigm in postnatal day 42–60 (PND42-60) sexually mature adolescent male and female mice[31]. In vSUB, vCA1 provides the major source of glutamatergic input onto vSUB BS and RS principal neurons. To selectively activate vCA1 afferents, we placed an extracellular stimulation electrode in the alveus/stratum oriens border of vCA1 (Fig. 1a). We then monitored electrically evoked excitatory postsynaptic currents (EPSCs) from electrophysiologically identified postsynaptic BS and RS neurons in ex vivo acute vHipp slices (Fig. 1a, b). In females, stress reduced vCA1-BS synaptic strength by over 50% according to their input-output relationship (Fig. 1c, d). We next measured strontium-evoked asynchronous EPSCs (aEPSCs) and EPSC paired-pulse ratios (PPRs) to identify the synaptic locus of this stress-induced decrease in excitatory transmission. aEPSC amplitude is commonly believed to reflect postsynaptic strength[32] while PPRs are an indirect measure of presynaptic release. In females, stress reduced aEPSC amplitudes at vCA1-BS synapses without altering the PPR, indicating that stress selectively decreased postsynaptic strength at these synapses (Fig. 1e, f). At vCA1-RS synapses, stress induced a less significant (30%) decrease in postsynaptic strength (Fig. 1g–j). Together, these stress-induced changes in vCA1-vSUB synapses manifest postsynaptically and were greater in BS cells, consistent with the current notion that BS cells are more susceptible to stress-induced perturbations[27,33,34].

The prominent stress-dependent reduction in excitatory synaptic transmission at vCA1-BS synapses in females motivated us to examine whether similar stress-induced synaptic adaptations occur in male mice. In contrast to females, stress did not impact basal vCA1-vSUB synaptic strength (Fig. 2a–j), consistent with other models of acute stress[27,35].

### Stress causes sexually dimorphic adaptations to vSUB PV-BS inhibition

We next probed whether stress drives changes in PV inhibition of BS cells to compensate for or exacerbate the robust impairment of basal excitatory transmission in females. PV interneurons exhibit sex-specific connectivity in vSUB, are more stress-sensitive than other hippocampal interneurons, and exert major inhibitory control over vSUB pyramidal cells[19,36–41]. Despite this, the impact of stress on PV inhibition in vSUB at the synaptic level remains unexamined. To isolate PV-mediated synaptic transmission, we injected a Cre-dependent channelrhodopsin variant (AAV-DIO-ChIEF) into the vSUB of PND21 PV-Cre mice and assessed light-evoked PV inhibitory postsynaptic currents (IPSCs) from BS and RS cells in acute slices from sexually mature adolescent mice (PND42-60) after stress (Fig. 3a, b). In females, stress significantly increased PV-BS inhibition by enhancing postsynaptic strength without altering presynaptic release (Fig. 3c–f). By contrast, stress did not significantly change PV-RS inhibition (Fig. 3g–j). In males, although excitatory synapses made by vCA1 were unaffected by stress, stress surprisingly reduced the postsynaptic strength of inhibitory PV-BS synapses without altering presynaptic release (Fig. 4a–f). Thus, although stress induced distinct sex-specific changes in the strength of inhibitory PV-BS synapses, both phenotypes share a postsynaptic locus. Similar to females, stress did not impact PV-RS inhibition in males (Fig. 4g–j). Together, stress increases the strength of PV-BS inhibitory synapses and reduces vCA1-BS excitatory synaptic strength, which may result in a robust overall net reduction in vSUB BS activity in females. By contrast, stress reduces PV-BS inhibition without changing vCA1-BS excitatory strength in males which may shift vSUB BS cells to a more active state and provide a synaptic explanation for why the general activity of vSUB is increased after stress in males[26,42].

### vSUB local circuit changes are not developmentally dependent

While the sex-specific synaptic adaptations in vSUB following stress in PND42-60 mice are striking, we next asked whether these adaptations are limited to sexually mature adolescents or whether stress can induce similar synaptic adaptations in young adult animals (PND 70-77)[43,44]. Consistent with the effects of stress on vCA1-vSUB excitatory synapses in adolescent females (Fig. 1), the strength of the excitatory synapses made by vCA1 onto BS and RS cells was reduced in stress-exposed PND70-77 females (Supplementary Fig. 1a–h). This reduction in vCA1 excitatory input was more pronounced in BS cells, akin to what was observed in adolescent females. Further, stress in PND70-77 females recapitulated the postsynaptic enhancement of PV-BS inhibition observed in adolescents (Fig. 3; Supplementary Fig. 1i–p). In males, we found that neither vCA1-BS nor vCA1-RS synapses were impacted by stress in adolescent or PND70-77 mice (Fig. 2 and Supp Fig. 2a–h). Additionally, stress exposure significantly reduced the strength of PV-BS inhibitory synapses via postsynaptic mechanisms in PND70-77 males (Supplementary Fig. 2i–p), which matches our results in adolescent males (Fig. 4). Taken together, these prominent sex-specific changes observed after stress in vSUB occur both in sexually mature adolescent and young adult mice, which indicates that these local circuit adaptations are not restricted to a specific developmental window but instead manifest over a broad age range.

### Stress impairs long-term potentiation at vCA1-BS synapses in females

Stress can alter the activity of brain regions by modulating the intrinsic excitability of cells, however, these studies primarily used male rodents[45–47]. To examine whether stress drives sexually dimorphic adaptations to the intrinsic excitability of vSUB neurons, we measured the rheobase of BS, RS, and PV neurons in vSUB in mice from both sexes. In females, stress did not alter the intrinsic excitability of vSUB BS, RS, or PV cells (Supplementary Fig. 3a–c). Similarly, in males, stress did not alter the intrinsic excitability in BS or RS cells (Supplementary Fig. 4a, b). However, stress increased the rheobase of PV cells in males, indicating a reduction in the intrinsic excitability of this population of inhibitory neurons (Supplementary Fig. 4c). While these changes likely impair the sustained firing of male PV interneurons, they are unlikely to

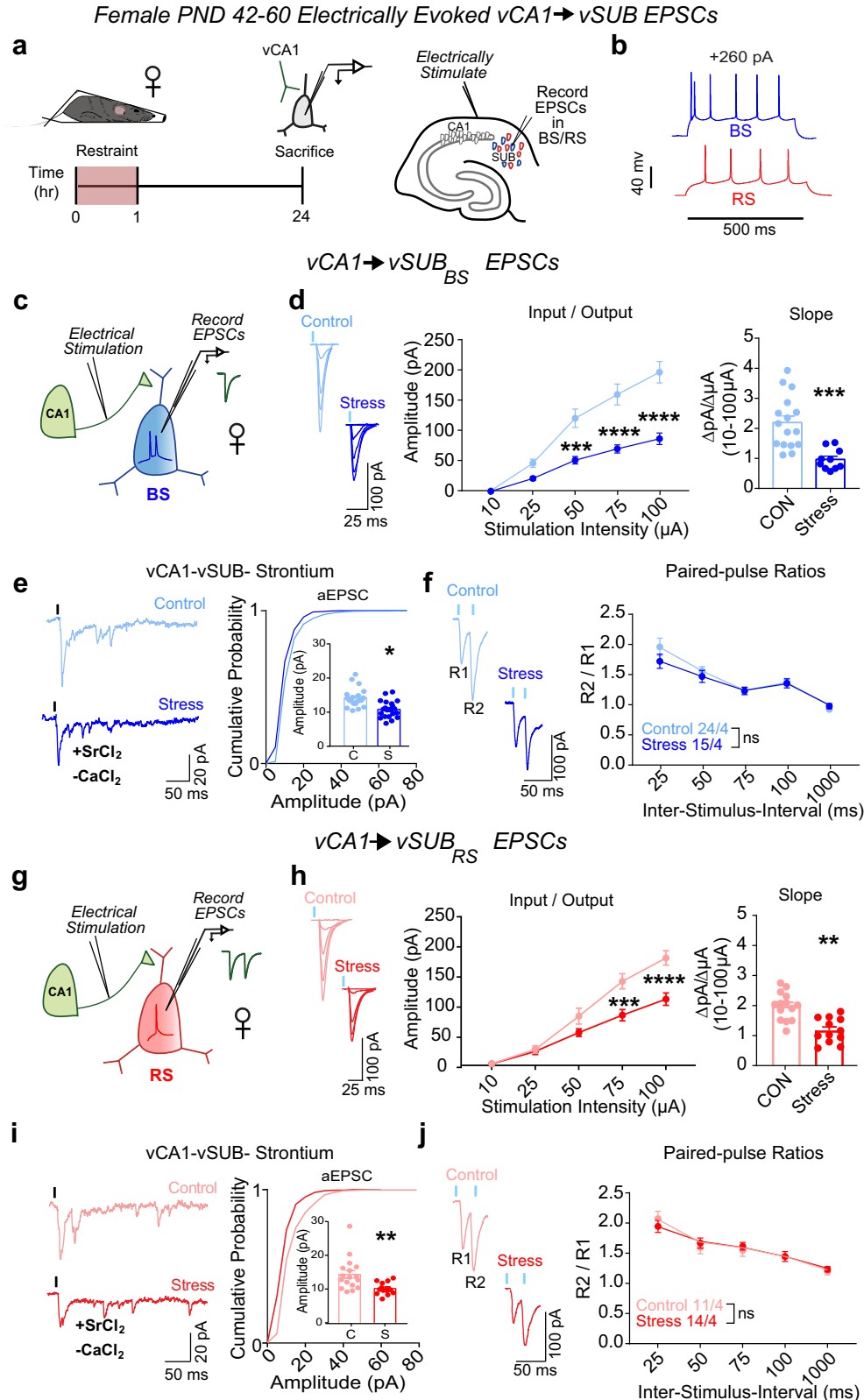

**Female PND 42-60 Electrically Evoked vCA1 ➤ vSUB EPSCs**

directly account for the reduction in the amplitudes of single light-evoked IPSCs at PV-BS synapses because our data indicate that this phenotype is driven by a reduction in postsynaptic strength.

Hippocampal plasticity occurs in response to major stressors and the dysfunction of plasticity is a hallmark of psychiatric diseases[48–50]. Long-term potentiation (LTP) in vSUB manifests pre- or post-synaptically depending on the firing pattern of the postsynaptic cell.

CA1-BS LTP occurs presynaptically due to increased presynaptic release probability, while CA1-RS LTP manifests postsynaptically through classical NMDA-receptor-dependent mechanisms[23,51]. Using an established induction protocol[23,51], we assayed LTP at vCA1-BS and vCA1-RS synapses in females after stress. Stress impaired presynaptic LTP in BS cells (Supplementary Fig. 3d–f) without altering postsynaptic LTP in RS cells (Supplementary Fig. 3g–i), reinforcing a stress-sensitive

**Fig. 1 | Stress weakens vCA1-vSUB basal excitatory synaptic strength in females.**
**a** Electrically evoked excitatory postsynaptic currents (EPSCs) from vCA1 were recorded in vSUB neurons 24 h after stress in female mice (PND42-60).
**b** Representative traces of action potential firing patterns in burst-spiking (blue) or regular-spiking (red) cells. **c, g** Experimental schema to record EPSCs from burst-spiking or regular-spiking cells. **d** Input-output representative traces (left), summary graph (middle), and slope (right) for EPSCs recorded in burst-spiking cells ((**d**) Stim Intensity x Stress Group $F_{(4, 96)} = 13.62$, ****$p < 0.0001$; 10 μA, $p > 0.9999$; 25 A, $p = 0.5756$; 50 μA, ***$p = 0.0007$; 75 μA, ****$p < 0.0001$; 100 μA, ****$p < 0.0001$; slope, ***$p = 0.0003$ ($t = 4.212$, $df = 24$); control n = 16/4, stress $n = 10/4$) or (**h**) regular-spiking cells ((**h**) Stim Intensity x Stress Group $F_{(4, 96)} = 10.82$, ****$p < 0.0001$; 10 μA, $p > 0.9999$; 25 μA, $p = 0.9997$; 50 μA, $p = 0.1651$, 75 μA, ***$p = 0.0001$; 100 μA, ****$p < 0.0001$; slope, **$p = 0.0043$ ($t = 4.149$, $df = 7$); control

$n = 15/4$, stress $n = 12/5$). **e, i** Representative traces of strontium-mediated aEPSCs after electrical stimulation (left) and aEPSC amplitude (right) for burst-spiking ((**e**) *$p = 0.0352$ ($t = 3.129$, $df = 4$); control $n = 20/3$, stress $n = 21/3$) or regular-spiking cells ((**i**;) **$p = 0.051$ ($t = 3.040$, $df = 28$); control $n = 16/3$, stress $n = 14/3$).
**f, j** Representative PPR trace (50 ms) (left) and PPR quantification (right) from burst-spiking ((**f**) ISI x Stress Group $F_{(4, 148)} = 1.236$, $p = 0.2982$; control $n = 24/4$, stress $n = 15/3$) or regular-spiking cells ((**j**) ISI x Stress Group $F_{(4, 92)} = 0.5969$, $p = 0.6658$; control $n = 11/4$, stress $n = 14/4$). Data are mean ± SEM calculated from the total number of cells indicated in graphs. Numbers in the legend represent the numbers of cells/animals. Statistical significance was determined by 2-way repeated measures ANOVA followed by Šidák's multiple comparisons test ((**d**) (middle), (**f, h**) (middle), (**j**)) or a nested unpaired Student's *t*-test (two-tailed) ((**d**) (right), (**e, h**) (right), (**i**)). Source data are provided as a Source Data file.

characterization of BS cells. By contrast, LTP was intact at vCA1-BS and vCA1-RS synapses in males (Supplementary Fig. 4d–i). Our results indicate that stress renders the vCA1-BS synapse incapable of activity-dependent presynaptic LTP selectively in females. Together, stress causes sex-specific excitatory synaptic adaptations such that excitatory basal synaptic transmission (Fig. 1) and activity-dependent plasticity (Supplementary Fig. 3) are remarkably impaired after stress in the vSUB of only females, and vSUB BS synapses are dominantly impacted.

### vSUB-BS cells disproportionately project to aBNST

The aBNST is a highly sexually dimorphic nucleus that controls stress and anxiety states[52,53]. The vSUB-aBNST circuit is known to be critical for stress hormone (i.e. corticosterone) release[18] and anxiety-like behaviors[16,17,54], but its cell type-specific connectivity remains undefined. To determine the cellular identity of aBNST-projecting cells within the vSUB, we stereotaxically injected mRuby expressing retrograde AAV2 (AAV2$_{rg}$-hSyn-mRuby) into the aBNST of PND21 mice. Examination of retrograde mRuby expression in the vHipp revealed that vSUB sends the majority (88%) of projections to aBNST, consistent with recent reports, however, to our knowledge, the cellular identities of aBNST-projecting vSUB neurons had never been investigated[55,56] (Supplementary Fig. 5a–c). To gain cell-type resolution of these aBNST-projecting vSUB neurons, we electrophysiologically defined mRuby-positive vSUB neurons in PND42-60 ex vivo acute brain slices. Despite the vSUB containing BS and RS in equal proportions[21,57], we found that, independent of sex, ~80% of aBNST-projecting neurons in the vSUB exhibited burst-spiking properties (Fig. 5a–d). Given this notable BS cell over-representation, we assessed whether there were differences in intrinsic excitability between BS and RS cells that project to aBNST that may compensate for this bias. We found no differences in BS or RS cell intrinsic excitability in either sex (Fig. 5e, f). The over-representation of BS cells in vSUB-aBNST projection circuitry is intriguing considering our data indicating that BS neurons are uniquely stress-sensitive and manifest sex-specific synaptic adaptations in the vSUB local circuit.

### Stress induces sex-specific changes to vSUB-aBNST pathway activity in vivo

To expand on our ex vivo findings, we employed fiber photometry in awake, behaving (PND45-55) mice to monitor in vivo $Ca^{2+}$ signals in aBNST-projecting vSUB cells, which is a measure of $Ca^{2+}$ activity, an indirect proxy of neuronal activity[58]. To selectively monitor the $Ca^{2+}$ dynamics of aBNST-projecting vSUB neurons, of which ~80% are BS cells (Fig. 5d), we utilized an intersectional viral approach where we stereotaxically injected aBNST with a retrograde AAV2 encoding Cre recombinase and tdTomato (AAV$_{rg}$-hSyn-Cre-P2A-tdTomato) and stereotaxically injected an AAV encoding a Cre-dependent $Ca^{2+}$ indicator (AAV$_1$-hSyn-FLEx$^{loxP}$-jGCaMP7f)[59] into vSUB. The recording fiber was positioned directly above vSUB to measure aBNST-projecting vSUB cells. We assessed activity in two ways. One, we monitored $Ca^{2+}$ activity time-locked to movement initiation before (i.e., control conditions)

and after stress and time-locked to struggle initiation during stress (Fig. 5g–l) to assess behavior-evoked $Ca^{2+}$ activity. Second, we assessed neuronal activity during the recordings by measuring the number of events, Z-score (the deviation of the ΔF/F signal from its mean), and the area under the curve (AUC) from -3s to +3 s relative to the time-locked behavior, as performed previously[59–62]. In females, stress increased the maximum Z-score amplitude, assessed during the window 10 s before and 20 s after a motion event, by ~30% compared to control (Fig. 5i–m), which represents increased neuron activity during struggling. After stress, the Z-score amplitudes during movement were not different from the control period (Fig. 5m). Mirroring these changes, AUC increased during stress and returned to control levels in the recovery period (Fig. 5n). Although the Z-score amplitudes and AUC were significantly increased during stress, we did not observe a significant concomitant increase in the frequency of $Ca^{2+}$ transients during the whole recording (Fig. 5o).

In males, the Z-score magnitude increased during stress relative to the control period upon movement by ~20%, which returned to baseline during the stress recovery period (Fig. 5p–t). Like females, the AUC significantly increased during stress, but unlike females, this increase persisted through the stress recovery period (Fig. 5u). Although the Z-score amplitudes and AUC increased during stress, the frequency of $Ca^{2+}$ transients non-significantly increased during stress but significantly decreased in the recovery period after stress (Fig. 5v). Collectively, stress increases the activity of aBNST-projecting vSUB cells when time-locked to struggle in both sexes. However, elevated cellular activity persists only in males during the recovery period while it returns to control levels in females. Critically, because we do not observe any changes in intrinsic excitability of these pyramidal cells (Supplementary Figs. 3–4 and Fig. 5e, f), the changes we observe in neural activity are presumed to be driven by synaptic input. Thus, the persistent elevation in neuronal activity in males post-stress are consistent with and provide insight into the impact of the reduced PV inhibition we uncovered post-stress (Fig. 4d). We did not observe any sex differences in struggling behavior and, importantly, the changes we observed in in vivo $Ca^{2+}$ activity cannot be explained by gross changes in locomotion during the recordings (Supplementary Fig. 5d-i).

### Stress induced sex-specific changes to vSUB-aBNST pathway activity in vivo are not developmentally dependent

To examine whether the effects on in vivo cell activity were not restricted to adolescence, we next assessed the impact of stress on this circuit in PND70-77 young adult mice. Consistent with sexually mature adolescent mice, PND70-77 mice did not exhibit sex differences in the number of struggles during restraint (Supplementary Fig. 6b). Similar to adolescent mice, adult animals of either sex exhibited increased Z-score amplitudes and in vivo $Ca^{2+}$ activity during stress and similar trends in frequency changes. Further, we observed a persistence in in vivo $Ca^{2+}$ activity during the stress recovery period in males but not females, agreeing with adolescent data (Supplementary Fig. 6c–p). Again, the changes we observed in

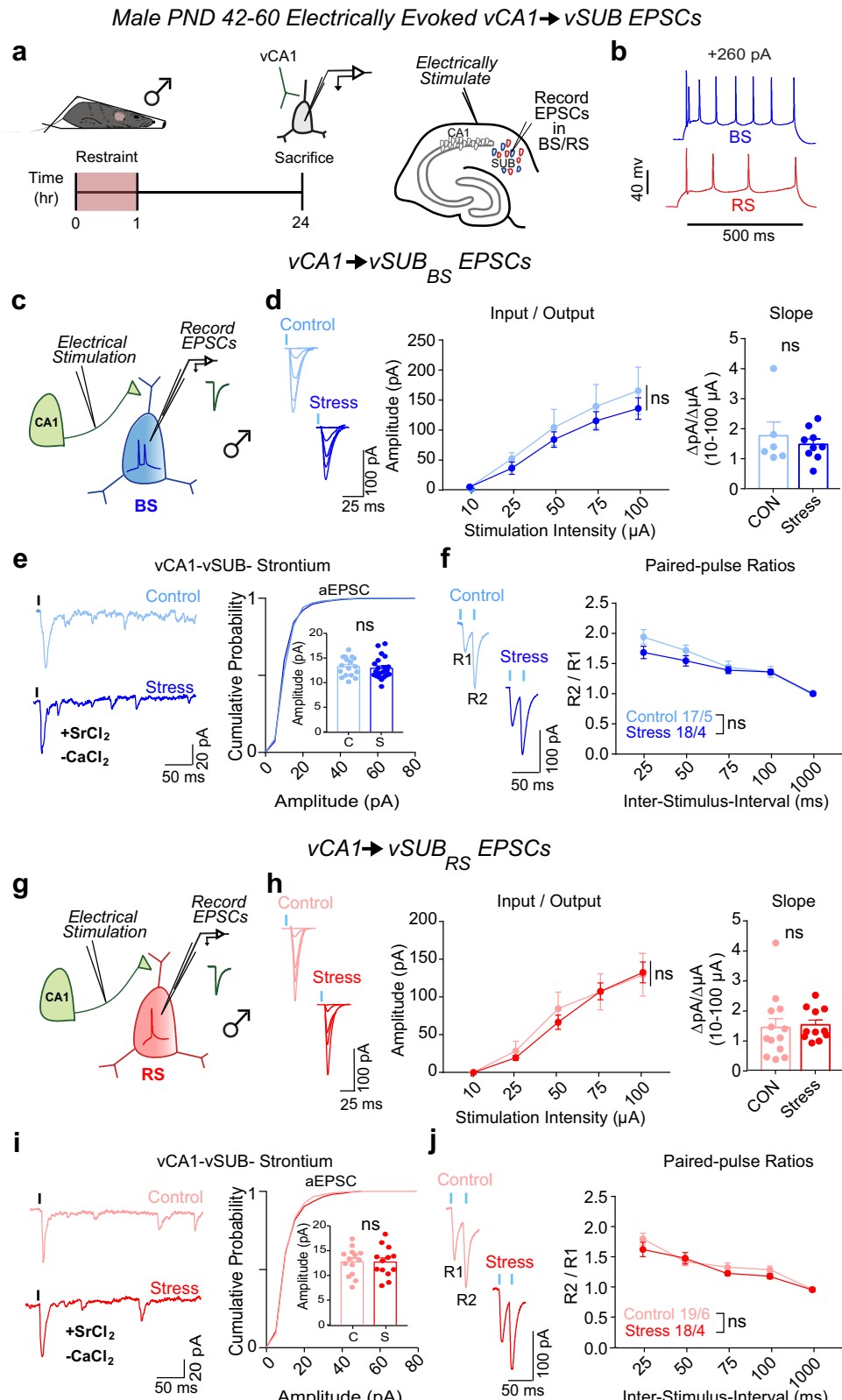

*Male PND 42-60 Electrically Evoked vCA1 → vSUB EPSCs*

*vCA1 → vSUB_BS EPSCs*

*vCA1 → vSUB_RS EPSCs*

Ca²⁺ activity cannot be explained by gross changes in locomotion during the recordings (Supplementary Fig. 6q–t). Thus, in sexually mature adolescent (PND45-55) and young adult (PND70-77) males, the activity of aBNST-projecting vSUB neurons is elevated during the period following stress exposure, while activity levels in females returns to baseline. These sex-specific findings are consistent with our electrophysiological analyses at each developmental period,

which revealed stress enhances PV-mediated inhibition and decreases excitatory input onto BS cells in females (Figs. 1, 3 and Supplementary Fig. 1) while selectively reducing the strength of PV-BS synapses in males (Figs. 2, 4 and Supplementary Fig. 2). Together, the stress sensitivity of aBNST projecting vSUB neurons and the sex-specific changes in their post stress in vivo Ca²⁺ activity occur across a broad range of mouse development.

**Fig. 2 | Stress does not alter vCA1-vSUB basal excitatory synaptic strength in males. a** Electrically evoked EPSCs from vCA1 were recorded in vSUB neurons 24 h after stress in male mice (PND42-60). **b** Representative traces of firing patterns in burst-spiking (blue) or regular-spiking (red) cells. **c, g** Experimental schema to record EPSCs from burst-spiking or regular-spiking cells. **d** Input-output representative traces (left), summary graph (middle), and slope (right) for EPSCs recorded in burst-spiking (Stim Intensity x Stress Group $F_{(4, 52)} = 0.4200$, $p = 0.7934$; slope, $p = 0.6195$ ($t = 0.5234$, $df = 6$); control $n = 6/3$, stress $n = 8/3$) or (**h**) regular-spiking cells (Stim Intensity x Stress Group $F_{(4, 88)} = 0.3494$, $p = 0.8438$; slope, $p = 0.8081$ ($t = 0.2458$, $df = 22$); control $n = 13/3$, stress $n = 11/4$). **e, i** Representative traces of strontium-evoked asynchronous EPSCs (aEPSCs) and aEPSC amplitude summary graph (right)

for burst-spiking ((**e**) $p = 0.7469$ ($t = 0.3458$, $df = 4$); control $n = 17/3$, stress $n = 14/3$) or regular-spiking cells (**i**; $p = 0.9882$ ($t = 0.01494$, $df = 26$); control $n = 14/3$, stress $n = 14/3$). **f, j** Representative PPR traces (50 ms) (left) and PPR measurements (right) from burst-spiking ((**f**) ISI x Stress Group $F_{(4, 132)} = 1.257$, $p = 0.2901$; control n = 17/5, stress n = 18/4) or regular-spiking cells ((**j**) ISI x Stress Group $F_{(4, 140)} = 0.9487$, $p = 0.4378$; control $n = 19/6$, stress $n = 18/4$). Data are mean ± SEM calculated from the total number of cells indicated in the graphs. Numbers in the legend represent the numbers of cells/animals. Statistical significance was determined by 2-way repeated measures ANOVA followed by Šidák's multiple comparisons test ((**d**) (middle), (**f, h**) (middle), (**j**)) or a nested unpaired Student's $t$-test (two-tailed) (**d**) (right), (**e, h**) (right), (**i**)). Source data are provided as a Source Data file.

## Stress exerts sex-specific effects on excitatory vSUB-aBNST synapses

In vSUB, stress-sensitive BS cells overwhelmingly innervate aBNST cells (Fig. 5a–d), which raised the intriguing possibility that vSUB-aBNST synapses also exhibit functional adaptations to stress. To test this, we injected AAV-ChIEF into the vSUB of PND21 mice and electro-physiologically monitored light-evoked vSUB EPSCs from postsynaptic aBNST neurons on PND42-60. In females, stress reduced the excitatory strength of vSUB-aBNST connections. To determine whether the locus of the reduction in synaptic transmission is pre-synaptic, we measured PPRs. We observed a significant increase in PPR, indicating that the stress-induced synaptic adaptation is driven by decreased presynaptic release probability from vSUB (Fig. 6a–c). In contrast, stress strengthened vSUB-aBNST synapses in males, which was driven by increased presynaptic release probability from vSUB (Fig. 6d, e). Together, in addition to the sex-specific changes to the vSUB local circuit, stress causes sexually dimorphic changes in vSUB output to the aBNST; a projection that critically regulates anxiety-like behavior and corticosterone release[16,18].

## Stress causes anxiety-like behaviors in females

Given the remarkable sex differences in ex vivo synaptic transmission and in in vivo Ca$^{2+}$ imaging in the vSUB-aBNST circuit following stress, we next examined whether stress impacted behaviors that model stress-related disorders like anxiety, which has a sex-biased incidence. First, using the elevated zero maze (EZM) to measure anxiety-like behavior 24 h post stress, we observed increased anxiety-like behavior in females but not males. Additionally, stress did not alter male or female locomotion on the EZM (Fig. 6f–j). To assess the robustness of this phenotype in females, we further assessed anxiety-like behavior with the open field test (OFT). In agreement with our EZM findings, stress increased anxiety-like behavior in females in the OFT without impacting locomotion (Supplementary Fig. 7a–c). In sum, females exhibit a robust anxiety-like phenotype across a battery of anxiety-like behavioral measurements following stress exposure. Given the strong anxiety-like behavior detected in females following stress, we also examined whether stress impacted coping and helplessness behavior with the forced swim test (FST). Stress increased immobility in females, which reflects an increase in helplessness and passive coping behaviors (Supplementary Fig. 7d)[63]. These data mirror clinical work that identify sex-specific vulnerability to stress disorders[1] and preclinical work that indicate females are more likely to exhibit anxiety-like behaviors after acute stressors[64–66]. In parallel with the sex-specific changes in synaptic and cellular activity we uncovered after stress, we demonstrate that stress also produces a sex-specific outcome in behavioral adaptations that is in line with clinical and preclinical literature.

## Sex-specific mechanisms underlie stress-induced adaptations of PV-BS inhibition

To begin to understand the mechanisms that may contribute to these sex-specific synaptic and behavioral phenotypes after stress, we examined the roles of systemic adrenergic and corticosterone signaling. In response to environmental stressors, the locus coeruleus (LC)

releases norepinephrine (NE), which activates α and β adrenergic receptors (ARs), and the HPA axis signals the release of corticosterone, which activates glucocorticoid (GR) and mineralocorticoid (MR) receptors[67]. Although understudied, vSUB cells robustly express ARs, MRs, and GRs[25–28]. To prevent adrenergic and corticosterone signaling, we pre-treated animals with propranolol (β-AR blocker) and phento-lamine (α-AR blocker) or metyrapone (corticosterone synthesis inhi-bitor), respectively.

In female controls, pretreatment with AR inhibitors did not impact basal PV inhibition (Fig. 7a–d). However, pretreatment with AR inhi-bitors 30 min before stress prevented the significant stress-induced enhancement of PV-BS postsynaptic inhibition in females (Fig. 7e–h). Interestingly, inhibiting corticosterone synthesis before stress did not prevent stress-induced augmentation of PV-BS postsynaptic inhibition (Fig. 7i–l). Blocking adrenergic or corticosterone signaling did not alter the basal or post-stress strength of PV-RS inhibition (Supplementary Fig. 8). Together, these data surprisingly indicate that adrenergic sig-naling plays a more critical role than corticosterone signaling in mediating the synapse-specific enhancement of PV-BS inhibition in female mice following stress.

Similar to females, pretreatment with AR or corticosterone blockers alone did not impact basal PV-BS inhibition in males (Fig. 8a–d). However, distinct from females, pretreatment with AR blockers did not prevent the stress-mediated depression of inhibi-tion at PV-BS synapses (Fig. 8e–h). By contrast, and further high-lighting the synaptic sex differences induced by stress, the stress-mediated depression of PV-BS synapses was absent when male mice were pretreated with metyrapone (Fig. 8i–l). PV-RS inhibition, which is not impacted by stress (Fig. 4h), was predictably unaltered by the blockade of either adrenergic or corticosterone signaling (Supple-mentary Fig. 9). In sum, our data implicate two distinct signaling pathways in the generation of stress-induced adaptations in PV-BS inhibition: while AR signaling drives stress-induced PV-BS inhibitory adaptations in females, corticosterone signaling plays a larger role in males.

## Discussion

Sex differences in brain circuitry represent potential drivers of sexually dimorphic behavioral responses to stress, with an emphasis on stress-related anxiety disorders[64,68–70]. Clinical and preclinical evidence strongly indicate that major acute stressors are particularly potent at inducing stress-related psychiatric disorders in females at greater rates than males[8–10,64,71], necessitating investigation into how each sex encodes acute stress. Here, we interrogated the circuitry of vSUB, a major output of the vHipp that provides critical top-down regulation of the HPA axis[18,72]. We recently demonstrated that vSUB exhibits basal sexual dimorphism[19], however, despite the known sex-specific responses to stress, sex as a biological variable has previously not been considered when studying the impact of stress on vSUB. Here, we provide a systematic synaptic interrogation of the impact of stress on the vSUB local circuit and its output to aBNST with cell-type- and synapse-specific resolution in male and female mice. We report that acute stress induces sex-specific functional adaptations to vSUB

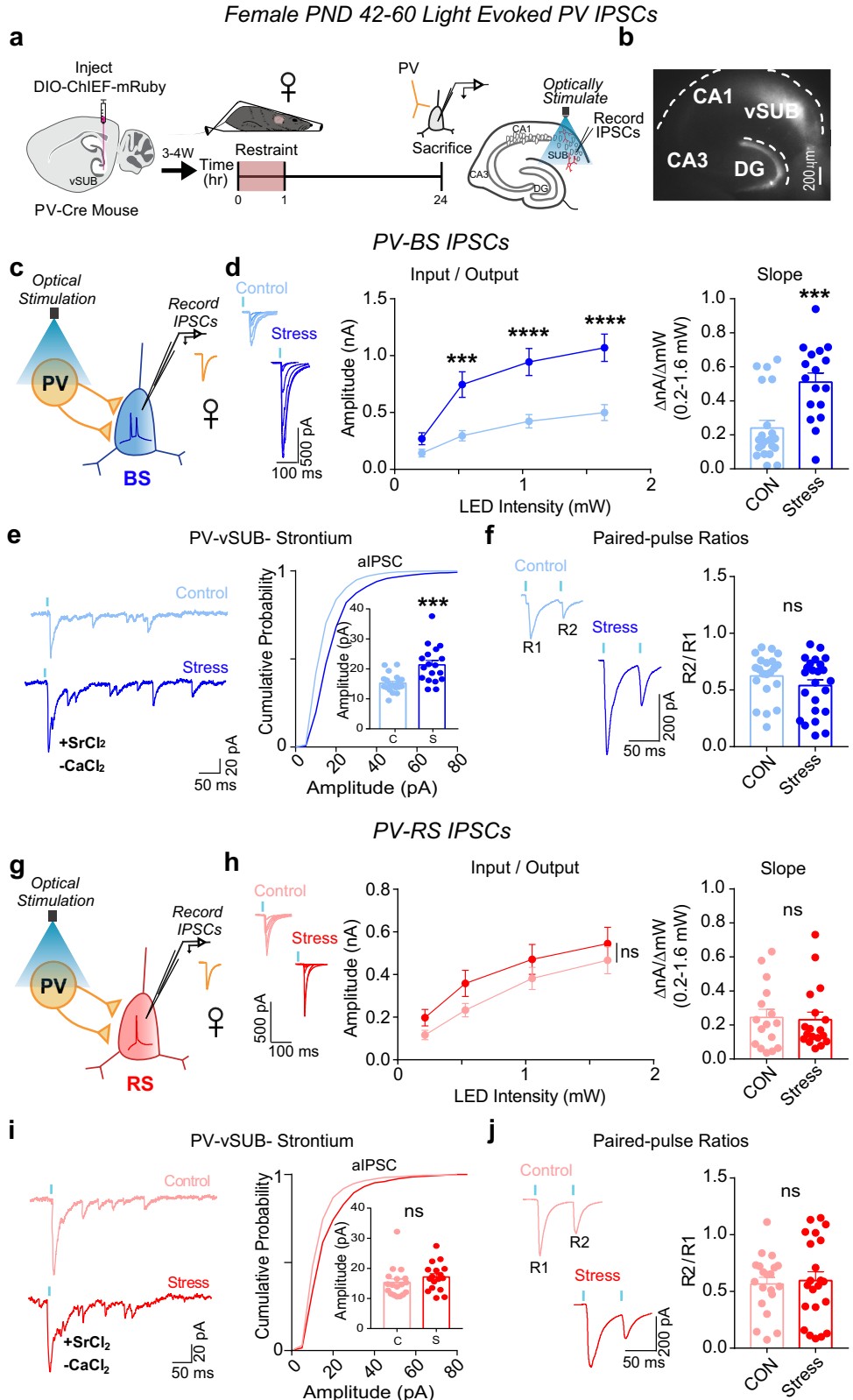

*Female PND 42-60 Light Evoked PV IPSCs*

circuitry across a range of time points, spanning adolescence to adulthood.

Stress induced profound sex-specific synaptic adaptations to PV-mediated inhibition in vSUB. Stress potentiated inhibitory PV-BS synapses in females and depressed PV-BS inhibition in males. Mechanistically, the potentiation of PV-BS synapses is primarily driven by AR signaling in females. In males, depression of the same

synapses is primarily caused by corticosterone signaling. Curiously, these sex-specific adaptations following stress occurred on the same class of principal neurons and manifested via changes in postsynaptic strength which raises two key questions. First, what is the consequence of these sex-specific stress adaptations observed locally within vSUB and in vSUB projections to aBNST? Second, what are the molecular signaling pathways downstream of adrenergic

**Fig. 3 | Stress strengthens parvalbumin-positive interneuron (PV)-burst-spiking (BS) inhibition via increased postsynaptic strength in females. a** A Cre-dependent ChIEF adeno associated virus (AAV) was injected into vSUB of PV-Cre female mice (PND42-60) and optogenetically evoked inhibitory postsynaptic currents (IPSCs) from PV interneurons were recorded from burst-spiking (blue) or regular-spiking (red) cells 24 h after stress. **b** Example image of ChIEF-mRuby expression in the vSUB. **c, g** Optogenetically stimulated IPSCs were recorded in burst-spiking or regular-spiking cells after stress. **d, h** Input-output representative traces (left), summary graph (middle), and slope (right) for IPSCs recorded in burst-spiking ((**d**) LED Intensity x Stress Group $F_{(3,108)} = 15.07$, ****$p < 0.0001$, 0.213 mW, $p = 0.7032$; 0.518 mW, ***$p = 0.0004$; 1.050 mW, ****$p < 0.0001$; 1.640 mW, ****$p < 0.0001$; slope, ***$p = 0.0004$ ($t = 3.908$, $df = 36$); control $n = 21/3$, stress $n = 17/3$) or regular-spiking cells ((**h**) LED Intensity x Stress Group $F_{(3,96)} = 0.2289$, $p = 0.8761$; slope, $p = 0.9034$ ($t = 0.1259$, $df = 7$); control $n = 16/3$, stress $n = 18/3$).

**e, i** Representative traces of strontium-mediated asynchronous IPSCs (aIPSCs) after optogenetic stimulation (left) and aIPSC amplitude (right) for burst-spiking ((**e**) ***$p = 0.0004$ ($t = 3.948$, $df = 36$); control $n = 20/3$, stress $n = 18/3$) or regular-spiking cells ((**i**; $p = 0.6575$ ($t = 0.5781$, $df = 4$); control $n = 18/3$, stress $n = 17/3$).
**f, j** Representative PPR traces (50 ms) (left) and PPR measurements (right) from burst-spiking ((**f**) $p = 0.5518$ ($t = 0.6488$, $df = 4$); control $n = 22/3$, stress n = 24/3) or regular-spiking cells ((**j**) $p = 0.8504$ ($t = 0.1957$, $df = 7$); control 20/3, stress n = 22/3). Data are represented as mean ± SEM; means were calculated from the total number of cells indicated in graphs. Numbers in the legend represent the numbers of cells/animals. Statistical significance was determined by a 2-way repeated measures ANOVA followed by Šidák's multiple comparisons test ((**d**) (middle), (**h**)(middle)) or nested unpaired Student's *t*-test (two-tailed) (**d** (left)-(**f**, **h**) (left)-(**j**)). Source data are provided as a Source Data file.

and corticosterone signaling that might result in sex-specific synaptic adaptations following stress?

Our data indicate that vSUB output to the aBNST, a highly sexually dimorphic region integrated within stress and anxiety circuitry[73–78], is disproportionately dominated by BS cells. aBNST neurons are predominantly inhibitory and participate in feed-forward inhibition of downstream targets that govern the stress response and anxiety. In rodents exposed to aversive contexts, the activity of aBNST-projecting vSUB neurons decreases[17] while experimentally increasing the activity of aBNST-projecting vSUB neurons decreases corticosterone during restraint stress and reduces anxiety-like behavior[16,18]. Thus, an intriguing possibility is that the activity of vSUB BS cells play an influential role in the response to stress. Here, we found that stress produces a diametrically opposite net effect on BS cell activity and in vSUB-aBNST output.

Following stress exposure, PV-mediated inhibition of BS cells is enhanced in females and reduced in males (Figs. 3–4). At vSUB-aBNST synapses, stress significantly decreases the strength of these synapses in females and increases synaptic strength in males (Fig. 6). Together, the shift toward inhibition of BS cells and decreased strength of vSUB-aBNST synapses in females may contribute to a higher anxiety state, consistent with the increased predator-prey avoidance and passive coping behaviors we observed in stress females (Fig. 5 & Supplementary Fig. 7). In contrast, the shift toward excitation of BS cells and potentiated vSUB-aBNST synapses in males may reflect a compensatory response to cope with an anxiogenic environment. Given the several distinct roles this projection contributes to through downstream signaling (i.e., corticosterone regulation, aversion, anxiety), it will be valuable to attain further functional granularity of postsynaptic aBNST neurons. There are at least five electrophysiologically distinct classes of BNST neurons and over 12 functionally distinct subregions[52] and it will be important to characterize vSUB inputs onto these distinct classes of aBNST neurons to gain further resolution. Our findings offer important insight into sexually dimorphic input to aBNST and highlight that in addition to its inherent sex-specific organization, long-range inputs might also encode for the sexually dimorphic properties of aBNST.

Here, we identified that the sex-specific adaptations in PV-BS synapse function are driven by different signaling pathways. In females, the postsynaptic locus of the PV-BS synaptic adaptation requires AR signaling (Fig. 7). AR signaling, largely in response to NE released from the LC, can drive rapid changes in synaptic strength by triggering the phosphorylation of GABA receptors[79–81] and gephyrin[82] and can produce long-lasting changes by altering gene expression[83]. Future studies are needed to determine whether there are sex differences in the expression patterns of AR and/or LC connectivity within vSUB. In males, corticosterone signaling is required for the postsynaptic reduction in PV-BS synaptic inhibition and can act through GR and MR to produce rapid and sustained changes to neuronal activity[84–86]. Future work is required to determine whether there are

sex differences in expression pattern or sensitivity of these receptors. Additionally, these pharmacological manipulations were administered systemically and the functional phenotypes observed may reflect adaptations to peripheral and central signaling, and future studies should gain resolution on this. The systemic administration of the pharmacological agents also raises the important future question whether adrenergic signaling in in females and corticosterone signaling in males activate receptors expressed on vSUB neurons to directly mediate the stress-induced synaptic adaptations or whether the adaptations are an indirect consequence of the activation of these signaling pathways. The sex-specific involvement of the AR and corticosterone signaling agree with clinical work indicating that autonomic dysfunction is more common in females with generalized anxiety disorder[87,88] and males with major depression disorders more frequently exhibit cortisol hyperactivity[89]. Further, preclinical work has identified pronounced sexual dimorphism in LC connectivity, NE activity during puberty, and corticotropin-releasing-factor signaling that may contribute to sex-specific outcomes in stress-related psychiatric disease pathology, highlighting these as appealing signaling pathways for further examination[90].

In addition to the sex-specific adaptations of PV-BS synapses and vSUB-aBNST output, we found that basal excitatory transmission and LTP at vCA1-BS synapses were impaired following stress in females but unaltered in males (Figs. 1–2 and Supplementary Figs. 3–4). Basal transmission at vCA1-BS synapses in females was reduced due to decreased postsynaptic strength, however, LTP at vCA1-BS synapses manifests presynaptically due to enhanced release probability. α2 and β2 ARs and MRs are localized in the pre- and post-synaptic membrane and activation of these presynaptic receptors facilitate presynaptic release, making these signaling pathways unlikely to contribute to the stress-induced impairment of vCA1-BS LTP. However, it is important to note that studies of presynaptic ARs and MRs did not include sex as a biological variable[27,84,91]. It will be important to identify the signaling pathway involved in preventing presynaptic LTP in female vSUB. Our findings contrast with studies using different models of acute or chronic stress in males that alter LTP[27,92,93], thus different physiological stressors appear to be encoded uniquely at hippocampal synapses and it will be important to continue studying how varying stressors impact brain circuitry.

We found that stress induced a robust anxiety-like phenotype in EZM and OFT assays in female mice. Importantly, the anxiety-like behavior in females is consistent with our functional analysis in vSUB and aBNST. The in vivo $Ca^{2+}$ imaging data suggest the in vivo $Ca^{2+}$ activity of aBNST-projecting vSUB neurons returns to baseline after stress in females. By contrast, we observed a persistent increase in $Ca^{2+}$ activity during the post-stress recovery period (Fig. 5 and Supplementary Fig. 6). Additionally, independent of sex, the onset of neural activity coincided with the onset of struggling. Struggling activity is interpreted as active coping and has been attributed to a neural circuit comprised of the insular cortex and BNST[94]. vSUB may participate in

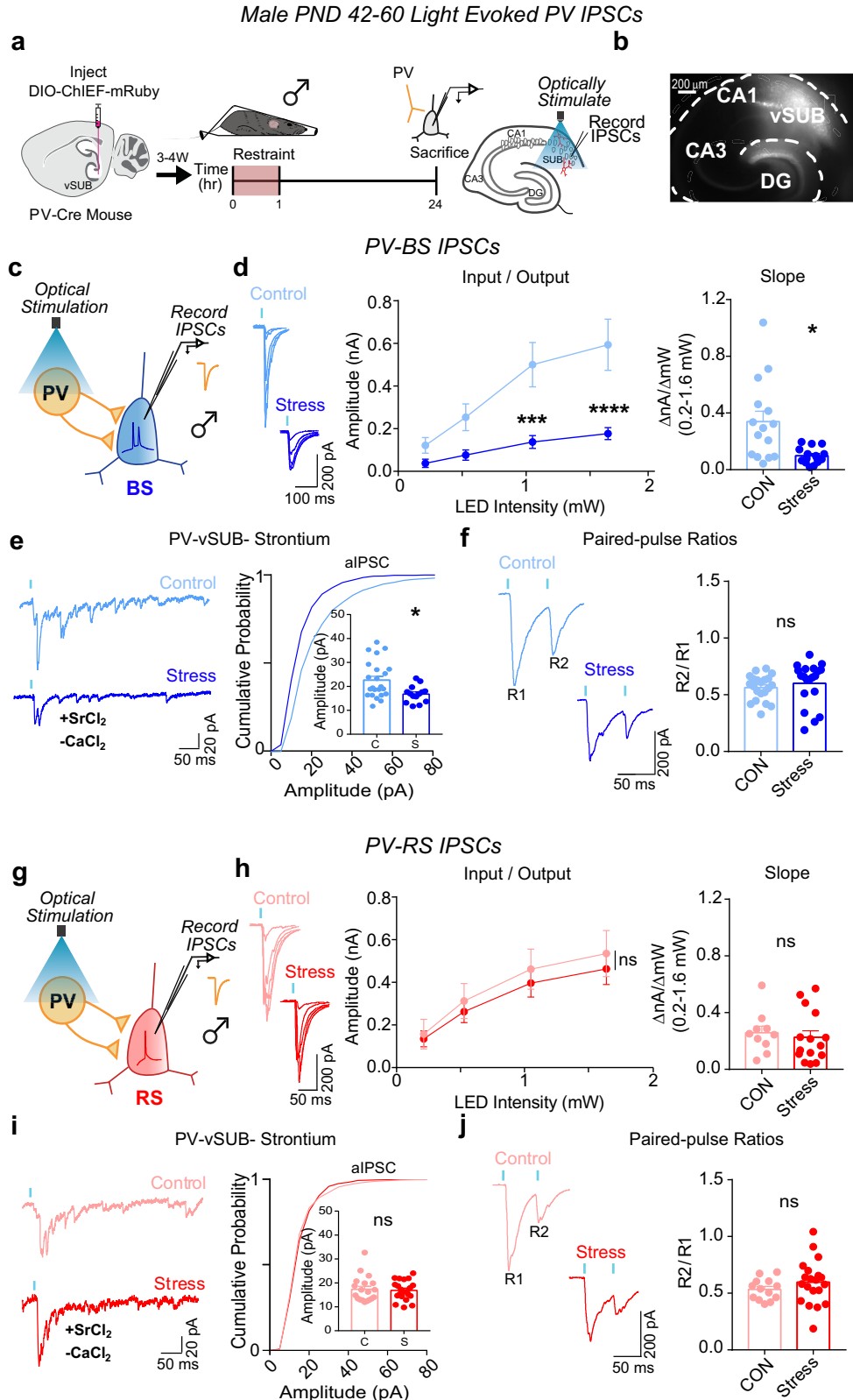

*Male PND 42-60 Light Evoked PV IPSCs*

this coping mechanism and may be downstream of insular cortex and upstream of BNST or represent a parallel pathway to promote the struggle behavior. This would suggest that perhaps vSUB is situated to provide top-down regulation of the HPA axis and participate in stress-coping mechanisms. Our data provide insight into the behavioral impact of our functional analyses and may elaborate on circuits responsible for active coping during stress.

Here, our study primarily used sexually mature mice spanning the end of mid adolescence through late adolescence (PND42-60)[31]. It is important to consider that during this developmental window, adolescent mice undergo dynamic changes, including sexually dimorphic changes in the adrenal system and in the maturation of the dopamine system. For example, the adrenal volume increases to a greater extent in females and females secrete greater amounts of corticosterone than

**Fig. 4 | Stress weakens parvalbumin-positive interneuron (PV)-burst-spiking (BS) inhibition via decreased postsynaptic strength in males. a** A Cre-dependent ChIEF adeno associated virus (AAV) was injected into vSUB of PV-Cre male mice (PND42-60) and optogenetically evoked inhibitory postsynaptic currents (IPSCs) from PV interneurons were recorded from burst-spiking (blue) or regular-spiking (red) cells 24 h after stress. **b** Example image of ChIEF-mRuby expression in the vSUB. **c, g** Optogenetically stimulated IPSCs were recorded in burst-spiking or regular-spiking cells after stress. **d, h** Input-output representative traces (left), summary graph (middle), and slope (right) for IPSCs recorded in burst-spiking ((**d**) LED Intensity x Stress Group $F_{(3, 84)}$ = 9.405, ****$p < 0.0001$, 0.213 mW, $p = 0.8332$; 0.518 mW, $p = 1994$; 1.050 mW, ***$p = 0.0005$; 1.640 mW, ****$p < 0.0001$; slope, *$p = 0.0369$ ($t = 3.080$, $df = 4$); control $n = 15/3$, stress $n = 15/3$) or regular-spiking cells (**h**; LED Intensity x Stress group $F_{(3, 69)}$ = 0.2209, $p = 0.8815$; slope, $p = 0.6163$ ($t = 0.5424$, df = 4); control $n = 10/3$, stress $n = 15/3$). **e, i** Representative traces of strontium-mediated asynchronous IPSCs (aIPSCs) after optogenetic stimulation (left) and aIPSC amplitude (right) for burst-spiking ((**e**) *$p = 0.0109$ ($t = 2.690$, $df = 35$); control $n = 23/3$, stress $n = 14/3$) or regular-spiking cells (**i**; $p = 0.7594$ ($t = 0.3085$, df = 38); control $n = 19/3$, stress $n = 21/3$). **f, j** Representative PPR traces (50 ms) (left) and PPR measurements (right) from burst-spiking ((**f**) $p = 0.4417$ ($t = 0.8531$, $df = 4$); control $n = 21/3$, stress n = 20/3) or regular-spiking cells ((**j**) $p = 0.2826$ ($t = 1.093$, df = 31); control 13/3, stress $n = 20/3$). Data are mean ± SEM calculated from the total number of cells indicated in the graphs. Numbers in the legend represent the numbers of cells/animals. Statistical significance was determined by a 2-way repeated measures ANOVA followed by Šidák's multiple comparisons test ((**d**) (middle), (**h**) (middle)) or nested unpaired Student's $t$-test (two-tailed) ((**d**) (left)-(**f, h**) (left)-(**j**)). Source data are provided as a Source Data file.

males[95]. Additionally, there are notable sex differences in dopamine system maturation that may be important to consider with our findings[96]. The sexually dimorphic developmental differences that occur might represent a permissive developmental window where acute restraint stress is capable of inducing different synaptic adaptations in female and male mice. However, we repeated several key experiments in young adult animals (PND70-77)[31,43,44] and found that not only did the sexually dimorphic synaptic adaptations manifest after stress, but the variability within and between conditions for the adolescent and young adult time points were similar. Together, stress induces sexually dimorphic adaptations that manifest during adolescence through young adulthood, suggesting that sex differences in maturation are not major drivers or sources of variability for the phenotypes we report here.

Despite the clinical impetus to consider sex as a biological variable when studying models of stress, nearly all preclinical work has been conducted solely on male rodents[97]. Here, we focused on sex differences in brain circuitry rather than circulating hormones. Both sexes have circulating levels of estrogen and testosterone, with estrogen fluctuations over a 4 day estrous cycle in female rodents and diurnal testosterone fluctuations with incredible individual variability of up to 20-fold differences between males[98–100]. Although we did not track estrous cycle or testosterone levels here, we observed higher variability in males on basal excitatory, basal inhibitory, and activity-dependent excitatory synaptic strengths. The high individual variability in males is consistent with others[101,102] and may be explained by testosterone levels and/or social hierarchy as the mice were group-housed[103], but more work is necessary to determine this. Our findings here provide strong evidence to evaluate sex as a biological variable when modeling diseases with sexually dimorphic features and illustrate striking sex differences in how acute stress is internalized in a cell-type, synapse-, and behavior-specific manner.

## Methods
### Mouse generation and husbandry
All procedures were conducted in accordance with guidelines approved by Administrative Panel on Laboratory Animal Care at University of Colorado, Anschutz School of Medicine, accredited by Association for Assessment and Accreditation of Laboratory Animal Care International (AAALAC) (00235). For all electrophysiology experiments, male and female mice were bred in house and group-housed (2-5) in ventilated cages enriched with cotton nestlets at the University of Colorado Anschutz. Transgenic PV-IRES-Cre mice (Jackson Laboratories, Stock No: 008069) breeders were kindly provided by Dr. Diego Restrepo and Gt(ROSA)26Sortm9(CAG-tdTomato)Hze mice were purchased from the Jackson Laboratory ("Ai9": Jax 008909). All experimental mice were maintained on congenic B6;129 or B6.Cg mixed genetic backgrounds. Mice were given food and water *ad libitum* and maintained at 35% humidity, 21–23 °C, on a 14/10 light/dark cycle in a dedicated vivarium. Mice

were genotyped in-house, and the sex of the animal was determined by external genitalia.

All fiber photometry, open field test, and forced swim test experiments took place at the University of California, Irvine, and were in accordance with the National Institutes of Health Guidelines for Animal Care and Use. Eight-week-old male and female C57BL/6 J mice were purchased from the Jackson Laboratory ("B6": Jax 000664) and housed in separate cages according to sex for the adult fiber photometry experiments. For the adolescent experiments, mice were bred in house and group-housed according to sex. Mice were housed in standard ventilated cages with corn cob bedding and two cotton cots for environmental enrichment. Lights were on a 12-h on/12-h off cycle (7:30 am–7:30 pm), and room temperature (22 °C +/- 2 °C) and humidity (55–65%) were controlled. Mice had free access to food and water. Mice were housed at the Gillespie Neuroscience Research Facility. For all studies, the experimental group assignment (control vs stress) was counterbalanced across littermates and age-matched. All experiments were conducted in males and females and the data were stratified by sex.

### Restraint stress procedure
Adult animals were habituated to handling for 3–5 days (5 min/day) prior to stress or control procedures. Handling mice consisted of mice freely moving between the experimenters' hands and mice were scooped rather than picked up from their tails to avoid undesired stress. Acute restraint stress was administered for 1 h using commercially available plastic cones (Braintree Scientific Inc) fastened with a twisty tie in a fresh cage. Mice were completely unable to move all limbs during the restraint procedure. Control mice moved freely for 1 h in a fresh cage. Control or stress procedures all occurred between 0800–1000. Control and stress animals were returned to their home cage or sacrificed for electrophysiology experiments immediately after either procedure. For pharmacologic intervention studies, adrenergic blockage was achieved with propranolol (10 mg/kg in saline; i.p.; Sigma) and phentolamine (15 mg/kg in saline; i.p.; Sigma) 30 min prior to restraint or control procedures. Corticosterone synthesis was inhibited by administering metyrapone (75–100 mg/kg in DMSO, i.p; Cayman Chemical) 1 h before restraint or in untreated animals. Doses for adrenergic blockade[104,105] and corticosterone synthesis inhibitors[106,107] were determined by previous literature. All injectable solutions were prepared fresh on the day of the experiment.

### Elevated zero maze
Twenty-four hours after stress or control conditions, adult mice (PND42-60) were tested in the elevated zero maze (EZM) during their light phase of the light/dark cycle to assess anxiety-like behavior[108]. Mice were accustomed to the behavior room for at least 1 h before testing. The mouse test order was counterbalanced by pretreatment and sex. Mild white noise (40–50 dB) was played during the habituation and experimentation period to minimize unwanted disruption.

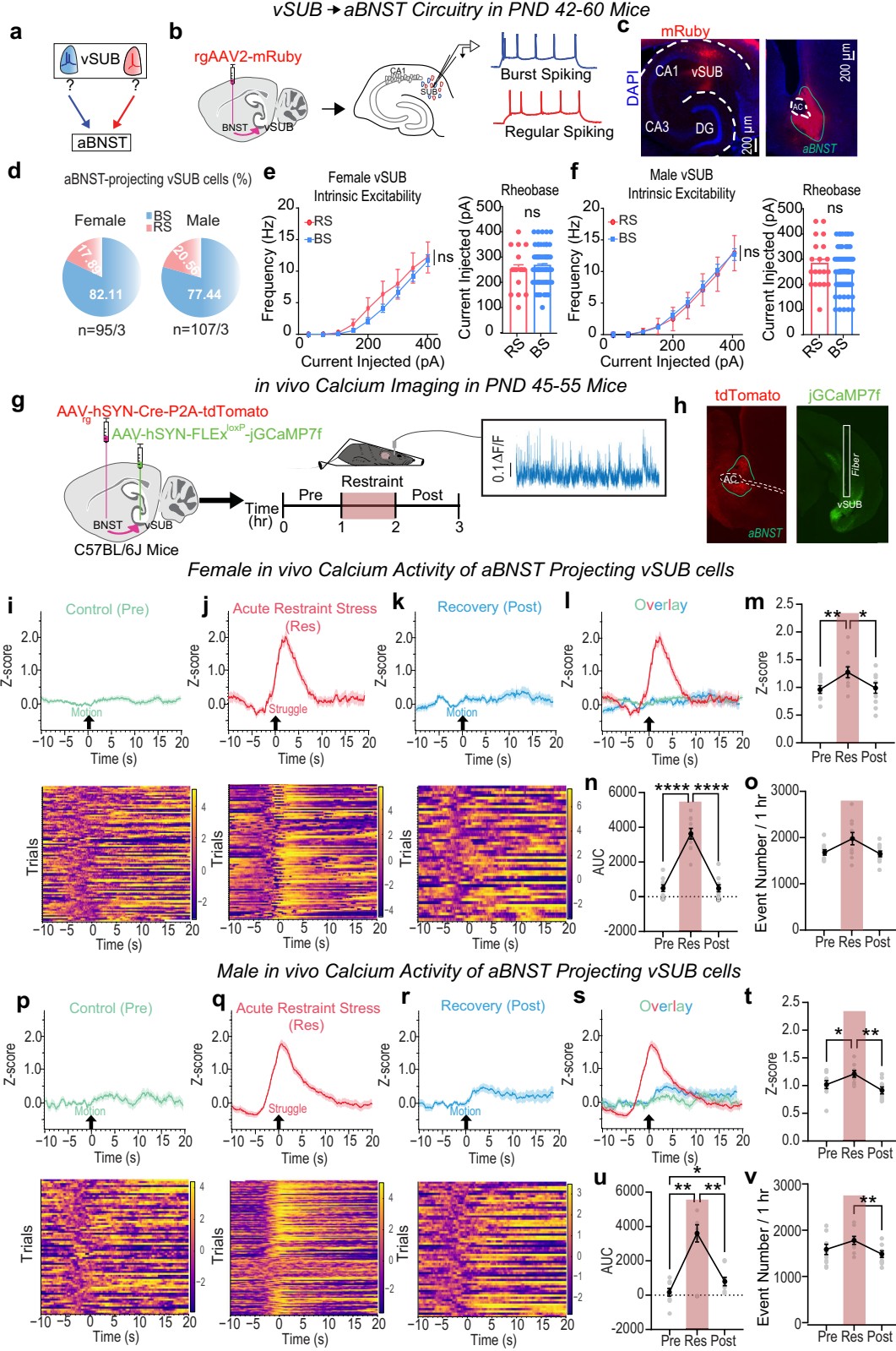

**vSUB → aBNST Circuitry in PND 42-60 Mice**

**in vivo Calcium Imaging in PND 45-55 Mice**

**Female in vivo Calcium Activity of aBNST Projecting vSUB cells**

**Male in vivo Calcium Activity of aBNST Projecting vSUB cells**

Experimenters remained outside of the room during the 10 min of behavior recording. The EZM consisted of 4 quadrants with a 40 cm inner diameter, 2 closed-arm sections, and 2 open-arm sections as illustrated in Fig. 6. Mouse movement was tracked with Ethovision software. The percentage of time in each arm and total distance traveled were analyzed from the entire 10 min of the EZM.

**Open field test**

The animals were assessed for anxiety-related behaviors using the open field test (OFT) (illustrated in Supplementary Fig. 7). For the OFT, the percentage of time spent in the center square (1/3 of the total area of the box) during a 5 min test period was calculated. OFT experiments were conducted under illuminance of ~300 lux.

**Fig. 5 | vSUB burst-spiking (BS) cells primarily project to anterior bed nucleus of the stria terminalis (aBNST) and exhibit sex-specific activity in vivo. a** vSUB-aBNST cartoon. **b** Retrograde labeling schema. **c** Representative retrograde labeling images. **d–f** Identification and quantification of intrinsic excitability and rheobase of aBNST-projecting vSUB cells and in female mice ((**e**) Current Injected x Cell Type $F_{(8, 616)} = 0.7289$, $p = 0.6661$; rheobase, $p = 0.6624$ ($t = 0.4383$, $df = 78$); burst-spiking=64/3 regular-spiking=16/3) or male ((**f**) Current injected x Cell Type $F_{(8, 720)} = 0.3925$, $p = 0.9249$; rheobase, $p = 0.1793$ ($t = 1.353$, $df = 90$); burst-spiking = 72/3, regular-spiking=20/3) mice. **g** Strategy to measure in vivo calcium activity from aBNST-projecting vSUB cells during control (Pre), stress (Res), and stress recovery (Post) periods. **h** aBNST injection site (right) and vSUB fiber placement (left) images. (**i–k, p–r**) Representative traces and heat maps from control (**i** female; **p** male), recovery (**k** female; **r** male) or struggle initiation periods (**j** female; **q** male). **l, s** Overlay of traces for females (**l**) and males (**s**). **m, t** Z-score amplitudes in females ((**m**) $F_{(1.684, 15.16)} = 9.289$, **$p = 0.0033$**, Pre vs Res **$p = 0.0088$**; Pre vs Post $p = 0.9114$; Res vs Post *$p = 0.0348$; $n = 10$) and males ((**t**) $F_{(1.911, 15.29)} = 12.76$, ***$p = 0.0006$, Pre vs Res *$p = 0.0390$; Pre vs Post $p = 0.2682$; Res vs Post **$p = 0.0013$; $n = 9$) mice. **n, u** Overall activity, the area under the curve (AUC), in females ((**n**) $F_{(1.676, 15.08)} = 87.03$, ****$p < 0.0001$, Pre vs Res ****$p < 0.0001$; Pre vs Post $p = 0.9996$; Res vs Post ****$p < 0.0001$; $n = 10$) and males ((**u**; $F_{(1.146, 9.168)} = 28.38$, ***$p = 0.003$, Pre vs Res **$p = 0.0012$; Pre vs Post *$p = 0.0259$; Res vs Post **$p = 0.0023$; $n = 9$). **o, v** Calcium event frequency in females ((**o**) $F_{(1.353, 12.18)} = 6.702$, *$p = 0.0173$, Pre vs Res $p = 0.0574$; Pre vs Post $p = 0.8446$; Res vs Post $p = 0.0529$; $n = 10$) and males ((**v**) $F_{(1.504, 12.03)} = 7.179$, *$p = 0.0127$, Pre vs Res $p = 0.1528$; Pre vs Post $p = 0.5012$; Res vs Post **$p = 0.0012$; $n = 9$). Data: mean ± SEM from the number of cells (electrophysiology) or animals (fiber photometry). Significance: 2-way repeated measures ANOVA followed by Šidák's multiple comparisons test ((**e, f**) intrinsic excitability), unpaired Student's $t$-test (two-tailed) ((**e, f**) rheobase), or 1-way repeated measures ANOVA followed by Šidák's multiple comparisons test (**m–o, t–v**). Source data are provided as a Source Data file.

## Forced swim test

The animals were also assessed for depressive-like behavior using the forced swim test. Cylinders (21.6 cm diameter, 26.7 cm height) were filled with tap water (25 °C) to 2/3 of the cylinder height. Animals were recorded for 6 min, starting when the animal was placed in the water. Behavioral assessments were made based on the immobility during the last 4 min of each recording. A mouse was considered immobile when it stopped struggling and floated motionless in the water, making only minimal movements required to keep its head above the surface[63].

## Stereotactic viral injections

Stereotactic injections were performed on PND21-22 mice (8-14 g) for electrophysiological experiments. Animals were induced with 4% isoflurane, maintained at 1-2% isoflurane, and then head fixed to a stereotactic frame (KOPF). Small holes were drilled into the skull and 0.1-0.2 µL solutions of adeno-associated viruses (AAVs) were injected with glass micropipettes into the brain at a rate of 10 µL/hr using a syringe pump (World Precision Instruments), consistent with standard protocols[19]. Coordinates (in mm) for ventral subiculum (vSUB) were (A-P/M-L/D-V from Bregma and relative to pia): -3.41, +/- 3.17, -3.45. Coordinates (in mm) for anterior ventral bed nucleus of the stria terminalis (avBNST) were: +0.15, +/-1.1, -4.5. All AAVs used in this study were packaged in-house: AAV DJ-hSYN-DIO$_{loxp}$-ChIEF-mRuby, AAV DJ-hSYN-ChIEF-mRuby, and AAV2-Retro-mRuby, consistent with standard protocols[19,51]. All AAV vectors were created from an empty cloning vector with the expression cassette is as follows: left-ITR of AAV2, human synapsin promotor, 5' LoxP site, multiple cloning site, 3' LoxP site, and right ITR. All AAV plasmids were co-transfected with pHelper into HEK293T cells. Cells were then harvested and lysed 72 h post transfection, and the virus was harvested from the 40% iodixanol fraction. Finally, the virus was concentrated in 100 K MWCO ultra-con filter.

## Ex vivo whole-cell electrophysiology

Sexually mature adolescent (PND42-60) or fully adult (PND70-77) mice were deeply anesthetized with isoflurane, decapitated, and their brains were rapidly dissected, consistent with standard protocols[19,51]. Brains were removed and 300 µm horizontal slices containing the vSUB or 250 µm coronal slices containing the aBNST were sectioned with a vibratome (Leica VT1200) in ice-cold high-sucrose cutting solution containing (in mM) 85 NaCl, 75 sucrose, 25 D-glucose, 24 NaHCO$_3$, 4 MgCl$_2$, 2.5 KCl, 1.3 NaH$_2$PO$_4$, and 0.5 CaCl$_2$. Slices were transferred to 31.5 °C oxygenated artificial cerebral spinal fluid (ACSF) containing (in mM) 126 NaCl, 26.2 NaHCO$_3$, 11 D-Glucose, 2.5 KCl, 2.5 CaCl$_2$, 1.3 MgSO$_4$-7H$_2$O, and 1 NaH$_2$PO$_4$ for 30 min, then recovered at room temperature for >1 hr before recordings. Recordings were made at 29.5 °C ACSF, with 3-5 MΩ patch pipettes, and cells were voltage-clamped at -70mV. Pyramidal neurons in the vSUB were visualized with a BX51 microscope with a 40x dipping objective collected on a Hamamatsu ORCA-Flash 4.0 V3 digital camera with an IR bandpass filter. The identity of pyramidal neurons (regular (RS) versus burst spiking (BS)) was determined by current-clamping and injecting a 500 ms depolarizing current in 50 pA steps, as previously described[19]. Cells that exhibited burst firing upon suprathreshold current injection (2-4 spikes with ~10 ms inter-spike-interval) were considered BS and those that did not were labeled RS.

To measure inhibitory currents, NBQX (10 µM) and D-AP5 (50 µM) were added to the ACSF, and a high-chloride internal solution containing (in mM) 95 K-gluconate, 50 KCl, 10 HEPES, 10 Phosphocreatine, 4 Mg$_2$-ATP, 0.5 Na$_2$-GTP, and 0.2 EGTA was used. To measure excitatory currents, picrotoxin (100 µM) was added to the ACSF, and a low-chloride internal solution containing (in mM) 137 K-gluconate, 5 KCl, 10 HEPES, 10 Phosphocreatine, 4 Mg$_2$-ATP, 0.5 Na$_2$-GTP, and 0.2 EGTA was used. For strontium experiments, CaCl$_2$ was replaced with 2.5 mM SrCl$_2$ in the recording ACSF. To assess intrinsic excitability of neurons, cells were current-clamped and given −100 to +400 pA current injections in 20−50:pA steps. To optogenetically stimulate ChIEF-expressing fibers in the vSUB or BNST, slices were illuminated with 470 nm LED light (ThorLabs M470L2-C1) for 3 ms through the 40x dipping objective located directly over the recorded cell. With an illumination area of 33.18 mm2 the tissue was excited with an irradiance of 0.006 to 0.17 mW/mm2. To electrically stimulate pyramidal cells, a bipolar nichrome stimulating electrode was made in-house and placed in the alveus/stratum oriens border of vCA1 and controlled by a Model 2100 Isolated Pulse Stimulator (A-M System, Inc.)[51]. To induce long-term potentiation of vCA1-vSUB synapses, we used a standard protocol of four 100 Hz tetani separated by 10 s[23,51].

## vHipp imaging and quantification

The aBNST of male and female PND21 mice was injected with rgAAV2-mRuby, as described above. On PND42-60, 100 µm acute slices containing vHipp were preserved in a 4% PFA + 4% sucrose solution overnight at 4 °C. Slices were mounted in DAPI Fluoromount-G (Southern Biotech) and imaged on an Olympus VS120 slide scanner (20x objective). Stitched images were then analyzed with FIJI. To identify hippocampal subregions, the DAPI signal was compared to the Allen Brain Explorer and boundaries were manually drawn to demarcate vSUB and vCA1. The mRuby channel was background subtracted and cells in vCA1 and vSUB were counted using the FIJI plugin Cell Counter. The percentage of cell from each vHIPP subregion was calculated and plotted in GraphPad Prism 10. Slices where individual cells in vSUB could not be distinguished due to intense mRuby fluorescence were not included for analysis.

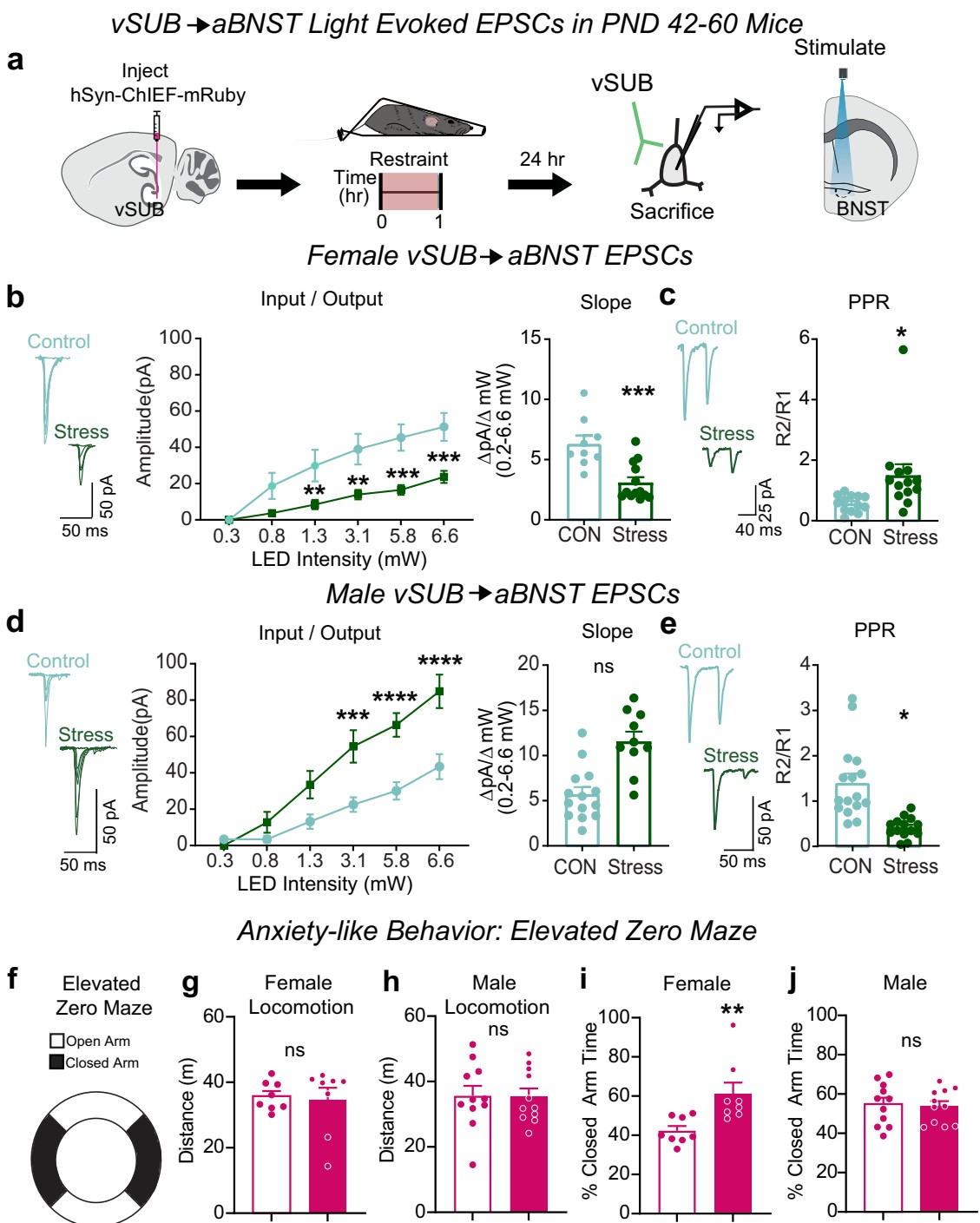

**Fig. 6 | Stress causes sex-specific changes to vSUB output to the anterior bed nucleus of the stria terminalis (aBNST) and anxiety-like behavior.**
**a** Experimental schema to optogenetically isolate vSUB input to aBNST and monitor light-evoked exciatory postsynaptic currents (EPSCs) in male and female mice (PND42-60) 24 h after restraint stress. **b, d** Input-output representative traces (left), summary graph (middle), and slope (right) for EPSCs recorded in the aBNST of female ((**b**) LED Intensity x Stress Group F(5, 100) = 8.563, ****$p$ < 0.0001, 0.3 mW, $p$ > 0.9999; 0.8 mW, $p$ = 0.1421; 1.3 mW, **$p$ = 0.0091; 3.1 mW, **$p$ = 0.0014; 5.8 mW, ***$p$ = 0.0002; 6.6 mW, ***$p$ = 0.0004; slope, ***$p$ = 0.0005 ($t$ = 4.181, $df$ = 20); control $n$ = 9/3, stress $n$ = 13/3) or male mice ((**d**) LED Intensity x Stress Group F(5, 110) = 10.84, ****$p$ < 0.0001, 0.3 mW, $p$ = 0.9984; 0.8 mW, $p$ = 0.7935; 1.3 mW, $p$ = 0.0588; 3.1 mW, ***$p$ = 0.0004; 5.8 mW, ****$p$ < 0.0001; 6.6 mW, ****$p$ < 0.0001; slope, $p$ = 0.06 ($t$ = 2.601, $df$ = 4); control $n$ = 14/3, stress $n$ = 10/3).
**c, e** Representative PPR traces (50 ms) (left) and PPR measurements (right) from

female ((**c**) *$p$ = 0.0207 ($t$ = 2.476, $df$ = 24), control $n$ = 13/3, stress $n$ = 13/3) or male ((**e**) *$p$ = 0.0348 ($t$ = 3.142, $df$ = 4), control $n$ = 16/3, stress $n$ = 14/3) mice. **f** Anxiety-like behavior was assessed in the elevated zero maze (EZM) and conveyed as the percent time spent in the closed arm. (**g, h**) Distance traveled during the EZM in females ((**g**) $p$ = 0.7923 ($t$ = 0.2684, $df$ = 14); control $n$ = 8, stress $n$ = 8) and males ((**h**) $p$ = 0.9573 ($t$ = 0.054117, $df$ = 20); control $n$ = 11, stress $n$ = 11). (**i, j**) Percent closed arm time in the EZM for females ((**i**) **$p$ = 0.0086 ($t$ = 3.055, df=14); control $n$ = 8, stress $n$ = 8) and males ((**j**) $p$ = 0.7830 ($t$ = 0.2792, $df$ = 20); control $n$ = 11, stress $n$ = 11). Data are represented as mean ± SEM and calculated from the total number of cells (electrophysiology) or total animals (behavior) as indicated in bars. Statistical significance was determined by a 2-way repeated measures ANOVA followed by Šidák's multiple comparisons test ((**b**) (middle), (**d**) (middle)), nested unpaired Student's $t$-test (two-tailed) ((**b**) (left)-(**c**, **d**) (left)-(**e**)), or unpaired Student's $t$-test (two-tailed) (**g**–**j**). Source data are provided as a Source Data file.

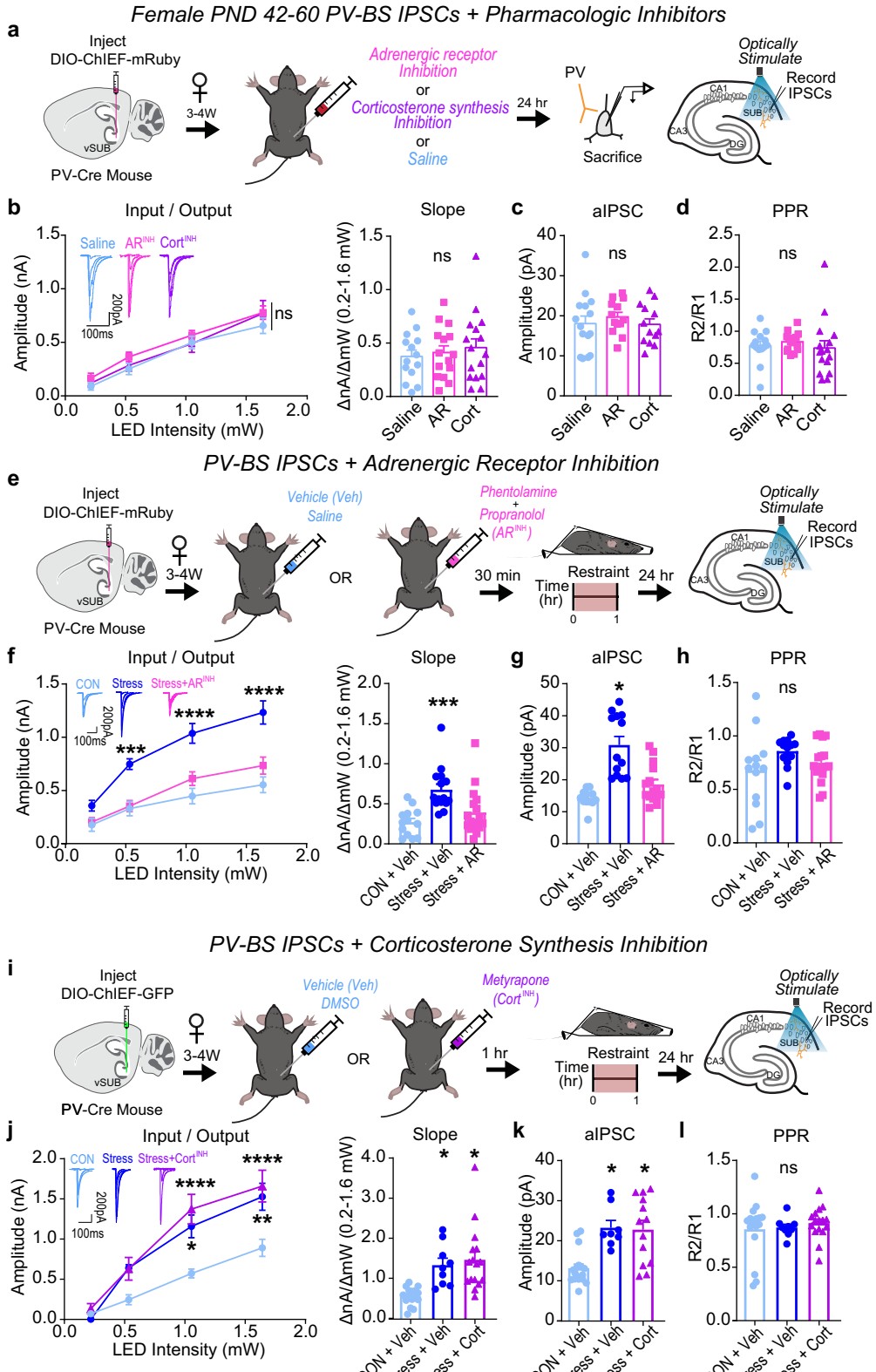

## Analysis of electrophysiology recordings

All recordings were acquired using Molecular Devices Multiclamp 700B amplifier and Digidata 1440 digitizer with Axon pClamp™ 9.0 Clampex software, lowpass filtered at 2 kHz and digitized at 10−20 kHz. Evoked PSC peak amplitudes from each recording were identified using Axon™ pClamp10 Clampfit software. To collect input/output curves, 8 sweeps (0.1 Hz) were averaged at each stimulation intensity. All final postsynaptic currents (PSC) values are displayed as the absolute value. The input/output slope was calculated using the SLOPE function in Microsoft Excel: (amplitude range/intensity range) from the linear range of the curves. The release probability was obtained over 10 sweeps (0.1 Hz) at each inter-stimulus interval (ISI) and was assessed by measurements of paired-pulse ratios (PPRs) at 25−1000 ms ISI. The PPR was measured by dividing the average PSC

**Fig. 7 | Adrenergic signaling drives stress-enhancement of parvalbumin inter-neuron (PV)-burst-spiking (BS) inhibition in females. a–d** Impact of adrenergic receptor (AR) or corticosterone (Cort) synthesis inhibitors in stress-naive females. **a** Experimental schema. **b** Inhibitory post-synaptic current (IPSC) input-output curves (left) and slope (right). (LED Intensity x Drug $F_{(6, 126)} = 0.3048$, $p = 0.9334$; slope, $F_{(2, 6)} = 0.2611$, $p = 0.7785$; saline $n = 14/3$, AR $n = 15/3$, Cort $n = 16/3$). **c** Strontium-evoked aIPSC amplitudes (c; $F_{(2, 6)} = 0.2684$, $p = 0.7733$; saline $n = 15/3$, AR $n = 13/3$, Cort $n = 13/3$). **d** Paired-pulse ratios (PPRs) (50 ms; $F_{(2, 42)} = 0.3617$, $p = 0.6986$; saline $n = 14/3$, AR $n = 15/3$, Cort $n = 16/3$). **e–h** AR inhibition in stress-exposed females. **e** Experimental schema. **f** IPSC input-output curves (left; LED Intensity x Condition $F_{(6, 123)} = 4.931$, ***$p = 0.0001$, Control+Saline vs Stress + AR, 0.213 mW, $p > 0.9999$; 0.528 mW, $p > 0.9999$; 1.050 mW, $p = 0.5795$; 1.640 mW, $p = 0.4584$; Control+Saline vs Stress + Saline, 0.213 mW, $p = 0.4963$; 0.528 mW, ***$p = 0.0006$; 1.050 mW, ****$p < 0.0001$; 1.640 mW, ****$p < 0.0001$) and slope (right; $F_{(2, 42)} = 8.709$, ***$p = 0.0007$, Control+Saline vs Stress+AR, $p = 0.3836$, Control + Saline vs Stress+Saline, ***$p = 0.0005$; Control + Saline $n = 13/3$, Stress+Saline $n = 15/3$, Stress + AR $n = 17/4$). **(g)** Strontium-evoked asynchronous IPSC (aIPSC) amplitudes ($F_{(2, 7)} = 6.955$, *$p = 0.0217$, Control+Saline vs Stress+AR $p = 0.5212$, Control+Saline vs Stress + Saline *$p = 0.0175$, Control + Saline $n = 12/3$, Stress+Saline $n = 13/3$,

Stress + AR $n = 15/4$). **h** PPRs (50 ms; $F_{(2, 7)} = 0.6844$, $p = 0.5532$, Control+Saline $n = 13/3$, Stress + Saline $n = 15/3$, Stress + AR $n = 17/3$). **i–l** Cort inhibition in stress-exposed females. **i** Experimental schema. **j** Input-output curve and representative traces (left; LED Intensity x Condition $F_{(6, 114)} = 5.539$, ****$p < 0.0001$, Control + DMSO vs Stress + Cort, 0.213 mW, $p > 0.9999$; 0.518 mW, $p = 0.1381$; 1.050 mW, ****$p < 0.0001$; 1.640 mW, ***$p < 0.0001$; Control+DMSO vs Stress+DMSO, 0.213 mW, $p > 0.9999$, 0.518 mW, $p = 0.2742$, 1.050 mW, *$p = 0.0188$, 1.640 mW, **$p = 0.0080$) and slope (right; $F_{(2, 6)} = 9.566$, *$p = 0.0136$, Control+DMSO vs Stress +Cort, *$p = 0.0123$, Control+DMSO vs Stress+DMSO, *$p = 0.0453$; Control + DMSO $n = 16/4$, Stress + DMSO $n = 9/2$, Stress+Cort $n = 16/3$). **k** Strontium-evoked aIPSCs ($F_{(2, 6)} = 11.29$, **$p = 0.0092$, Control+DMSO vs Stress + Cort *$p = 0.0123$, Control + DMSO vs Stress+DMSO *$p = 0.0195$; Control+DMSO $n = 17/4$, Stress + DMSO $n = 8/2$, Stress + Cort $n = 13/3$). **l** IPSC PPRs (50 ms; $F_{(2, 6)} = 0.1572$, $p = 0.8580$, Control + DMSO $n = 16/4$, Stress+DMSO $n = 9/2$, Stress + Cort $n = 16/3$). Data: mean ± SEM calculated from individual cells indicated in graphs. Significance: 2-way repeated measures ANOVA followed by Šidák's multiple comparisons test (**b–j** input-output curves) or nested 1-way ANOVA followed by Šidák's multiple comparisons test (**b–j** input-output slopes, **c-d**, **g-h**, **k-l**). Source data are provided as a Source Data file.

amplitude evoked by the second stimulus by the average PSC amplitude evoked by the first stimulus (R2/R1). For strontium experiments, the slices were stimulated over 15 sweeps (0.25 Hz) and the amplitudes of asynchronous PSC events that occurred within a 300 ms window immediately following the phasic PSC were analyzed using Clampfit event detection software[19].

## Fiber photometry procedure

PND30-35 or PND56-60 mice were injected with 200 μL of AAVrg-hSyn-Cre-P2A-tdTomato ($2.3 \times 10^{13}$ vg/mL) into the aBNST at the coordinates AP + 0.1; ML 0.8; DV -4.5 (in millimeters relative to the Bregma, midline, or dorsal surface of the brain) at a rate of 1.6 nL/sec. During the same surgery, 500 μL of AAV1-hSyn-FLEx[loxP]-jGCaMP7f ($1.2 \times 10^{13}$ vg/mL) was injected into the vSub at coordinates AP -4.04; ML 3.4; DV -3.5. The glass capillary was left in place for 5 min following injection before removal. After completing the two injections, a 400 μm diameter, 0.39NA, 3mm-long optical fiber was implanted directly over the vSUB. The implantation site was targeted at 0.1 mm dorsal to the injection site. Implants were secured to the skull with metal screws (Antrin Miniature Specialists), Metabond (Parkell), and Geristore dental epoxy (DenMat). Mice recovered for at least 2 weeks before experiments began (PND45-55 for adolescent and PND70-77 for adult), a rest period consistent with others[109–112]. Combined with post-hoc verification of correct viral targeting and fiber placement for every experimental animal, the preferential projection of vSUB to aBNST within vHIPP encouraged us that we are selectively targeting vSUB neurons (Supplementary Fig. 5a–c).

Mice were handled for 3 days before the start of recordings to reduce the novelty of handling. On recording days, recordings were performed from 8:30 a.m. to 12:00 p.m. to match potential circadian effects in the electrophysiology experiments. Prior to performing recordings, the top of the ferrule was cleaned using a 70% alcohol solution. Following this, each of the three periods was recorded, sequentially, each with a recording time of 1 h. At the end of each 1-h recording session, the mice were moved to a new cage to avoid the influence of odor from the previous session. Throughout the entire 3 h recording period, the mice had no access to water or food.

During the pre-restraint stage, following patch cable connection, mice were placed into a clean cage and allowed to move freely for 1 h. Mice were disconnected from the patch cable after this recording was complete. During the restraint stage, mice were enclosed in a conical plastic bag with open holes on the right top of their heads. The holes were only large enough to allow the ferrule to pass through. After enclosing the mouse, we wrapped the excess plastic portion around the mouse's body and secured the wrap with tape and staples, ensuring that

the mouse was able to comfortably breathe. After laying the mouse flat into the new cage, the mouse was again connected to the patch cord, and recording was performed for 1 h. Mice were again disconnected from the patch cable after this recording was complete. During the post-restraint recovery period, mice were freed from the restraints, at which time the mice were again connected to the patch cord, and another hour of recording was performed, as in the pre-restraint stage.

## Analysis of fiber photometry recordings

Fiber photometry recordings were conducted using previously described equipment[113]. In brief, 465 nm excitation light (0.95 mW) was used to stimulate $Ca^{2+}$-dependent emission, while 405 nm excitation light was used to control for motion artifacts via isosbestic emission. Both light sources were controlled by a RZ5P real-time processor (Tucker Davis Technologies) using Synapse software. The 465 nm light was modulated at 210 Hz, and the 405 nm light at 330 Hz, with both delivered simultaneously through an implanted 400 μm diameter optical fiber. All optical signals were bandpass filtered using a Fluorescence Mini Cube FMC4 (Doric) and measured with a femtowatt photoreceiver (Newport). Data were recorded at a sampling rate of 100 Hz.

Signal processing was performed with MATLAB (MathWorks Inc.). For the definition of timestamps for motion, during pre- and post-restraint sessions the motion of the mouse was tracked using Biobserve software. Following the video recording, a custom Python script calculated velocity of the mouse by taking the distance the center body coordinate of the mouse moved from frame to frame. The velocity threshold for a motion event was a sustained event with velocity >12.5 cm/s. For the restraint period, as mice could not locomote during their restraint, motion was defined visually by the observer as events where the mouse clearly moved their trunks as well as their heads. Events where the mouse only turned their heads were not counted. Motion events detected were aligned to the corresponding video and used to calculate the PSTHs.

All traces from individual mice were manually screened to exclude individuals with traces that contained high levels of noise and/or unclear signals. We used the isosbestic signal to correct for changes unrelated to calcium activity by fitting the 405 nm signal to the 465 nm signal using a linear regression model and subtracting the scaled 405 nm signal from the 465 nm signal to isolate calcium-dependent changes. Data points containing large motion artifacts were manually removed. The Guppy script was then used to calculate and create plots for the peri-stimulus time histogram (PSTH), area under the curve (AUC), signal amplitude, and frequency for each mouse. PSTHs were generated using Z-score normalization, with a 10 s window applied for transient detection. AUC was calculated for two time intervals, from

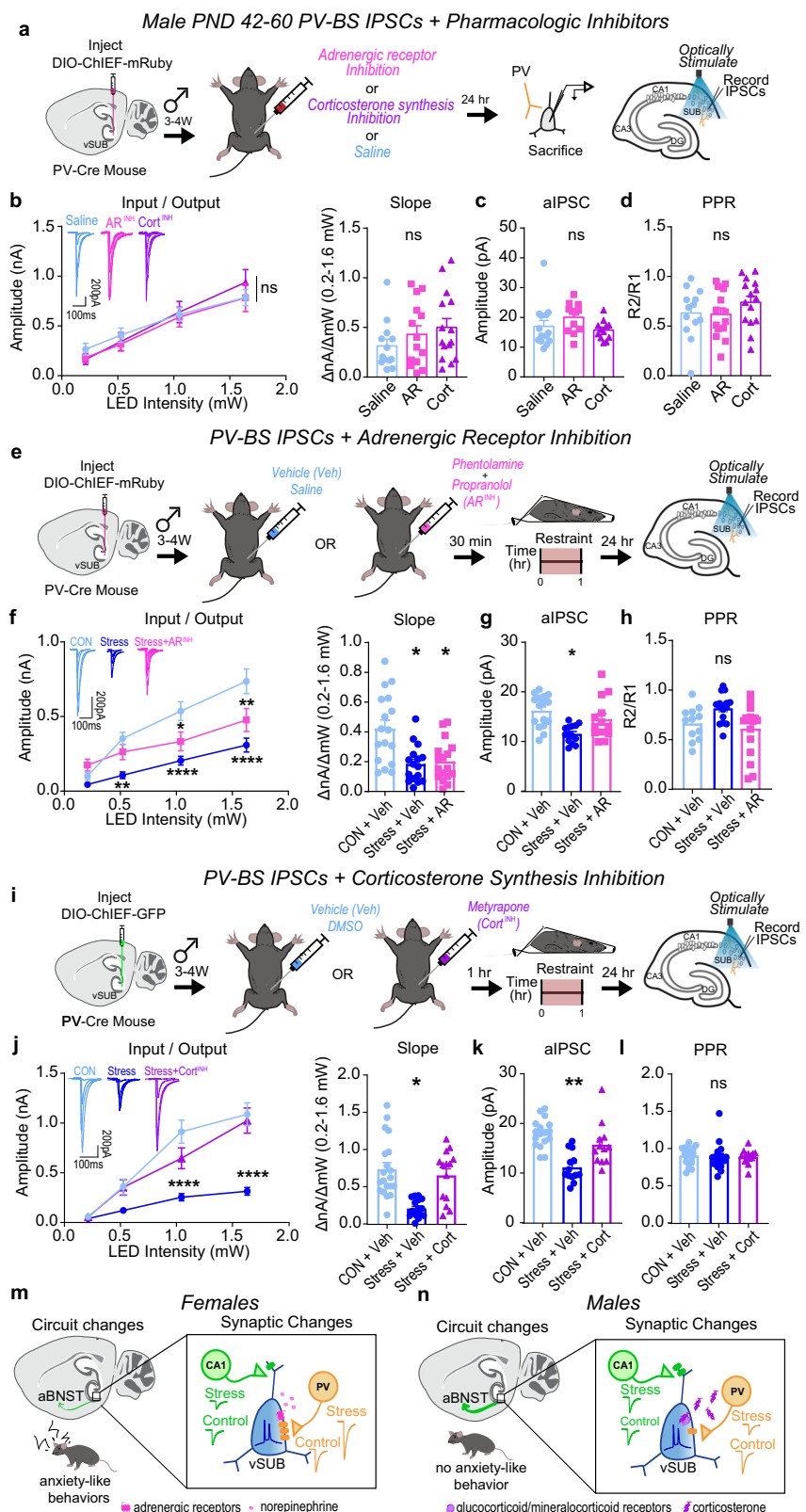

-3 s to 0 s and from 0 s to +3 s relative to each timestamp, enabling precise quantification of calcium dynamics.

## Quantification and statistical analyses

All statistical analyses were performed in Prism 10 (GraphPad). For two-group comparisons, we used a nested student-paired or unpaired 2-tailed Student's *t*-test (two-tailed) where stated in the figure legends. To estimate the relationship between locomotor activity and Ca²⁺ signal in fiber photometry experiments, a simple linear regression was used. For multiple group comparisons, we used a nested one-way ANOVA, one-way, or two-way ANOVA where stated in the figure legends. Repeated-measures ANOVA was also used where stated in the figure legends. Šídák or Tukey *post-hoc* comparisons were performed following ANOVAs. All experiments

**Fig. 8 | Corticosterone signaling drives stress-reduction of parvalbumin interneuron (PV)-burst-spiking (BS) inhibition in males. a–d** Impact of adrenergic receptor (AR) or corticosterone (Cort) synthesis inhibitors in stress-naive females. **a** Experimental schema. **b** Inhibitory post-synaptic current (IPSC) input-output curves (left; LED Intensity x Drug $F_{(6, 117)} = 0.9416$, $p = 0.4682$) and slope (right; $F_{(2, 6)} = 0.9140$, $p = 0.4503$; saline $n = 13/3$, AR $n = 14/3$, Cort $n = 15/3$). **c** Strontium-evoked asynchronous IPSC (aIPSC) amplitudes ($F_{(2, 6)} = 1.229$, $p = 0.3570$; saline $n = 14/3$, AR $n = 12/3$, Cort $n = 12/3$). **d** IPSC PPR (50 ms) ($F_{(2, 6)} = 0.4402$, $p = 0.6631$; saline $n = 13/3$, AR $n = 14/3$, Cort $n = 15/3$). **e–h** AR inhibition in stress-exposed males. **e** Experimental schema. **f** Input-output curve (left; LED Intensity x Stress Group $F_{(6, 141)} = 9.182$, ****$p > 0.0001$, Control+Saline vs Stress + AR, 0.213 mW $p = 0.9465$, 0.528 mW $p = 0.8516$, 1.050 mW *$p = 0.0412$, 1.640 mW **$p = 0.0033$; Control + Saline vs Stress+Saline, 0.213 mW $p = 0.9909$, 0.528 mW **$p = 0.0074$, 1.050 mW ****$p < 0.0001$, 1.640 mW ****$p < 0.0001$) and slope (right; $F_{(2, 7)} = 7.844$, *$p = 0.0163$, Control+Saline vs Stress+AR *$p = 0.0283$, Control + Saline vs Stress+Saline *$p = 0.0186$; Control + Saline $n = 17/4$, Stress+Saline $n = 16/3$, Stress + AR $n = 17/3$). **g** aIPSCs ($F_{(2, 7)} = 4.801$, *$p = 0.0487$, Control + Saline vs Stress+AR $p = 0.4003$, Control + Saline vs Stress+Saline *$p = 0.0344$, Control+Saline $n = 16/4$, Stress + Saline $n = 14/3$, Stress+AR $n = 13/4$). (**h**) IPSC PPRs (50 ms; $F_{(2, 7)} = 1.629$,

$p = 0.2625$, Control+Saline $n = 17/4$, Stress+Saline n = 16/3, Stress + AR $n = 17/3$). **i–l** Cort inhibition in stress-exposed males. **i** Experimental schema. **j** Input-output curve (left; LED Intensity x Stress Group $F_{(6, 144)} = 11.88$, ****$p < 0.0001$, Control + DMSO vs Stress + Cort, 0.213 mW $p > 0.9999$, 0.518 mW $p > 0.9999$, 1.050 mW $p = 0.1380$, 1.640 mW $p = 0.9988$; Control + DMSO vs Stress+DMSO, 0.213 mW $p > 0.9999$, 0.518 mW $p = 0.1890$, 1.050 mW ****$p < 0.0001$, 1.640 mW ****$p < 0.0001$) and slope (right; $F_{(2, 7)} = 5.235$, *$p = 0.0407$, Control + DMSO vs Stress + Cort $p = 0.8739$, Control+DMSO vs Stress+DMSO *$p = 0.0345$; Control+DMSO $n = 20/4$, Stress+DMSO $n = 14/3$, Stress + Cort $n = 14/3$). **k** aIPSC amplitudes ($F_{(2, 7)} = 13.65$, **$p = 0.0038$, Control + DMSO vs Stress+DMSO $p = 0.2359$, Control+DMSO vs Stress + DMSO *$p = 0.0025$, Control+DMSO $n = 16/3$, Stress+DMSO $n = 13/3$, Stress +Cort $n = 12/3$). **l** IPSC PPRs (50 ms; $F_{(2, 7)} = 0.02841$, $p = 0.9721$, Control+DMSO $n = 20/4$, Stress+DMSO $n = 16/3$, Stress+Cort $n = 14/3$). Summary models for females (**m**) and males (**n**). Data are mean ± SEM calculated from individual cells. Significance: 2-way repeated **m**easures ANOVA followed by Šidák's post-hoc test (**b–j** input-output curves) or nested 1-way ANOVA followed by Šidák's post-hoc test (**b–j** input-output slopes, **c**, **d**, **g**, **h**, **k**, **l**). Source data are provided as a Source Data file.

were replicated in at least 3 animals/condition/sex, unless otherwise noted, and littermate matched across groups. To confirm that all main electrophysiological findings were sufficiently powered, we ran a post-hoc power analysis (G*Power 3[114]); α = 0.05, and 1-β = 0.95) using our experimentally determined effects sizes that ranged between 1.2 and 2.4 (average effect size: 1.97). In experiments where significance was detected, we met or exceeded the predicted sample size determined by the calculated effect size. We attempted to collect roughly equal numbers of cells per animal per experiment, such that we are comfortable knowing that the collected data reflect the manifested synaptic adaptations in each animal and that each cell carries equal weight in the analysis. The robust regression and outlier removal test in Prism 10 was used to detect any outliers (5 cells in total were excluded). Whenever possible, the experimenter was blinded to animal sex, treatment, and cell identity during analysis. Cells of low quality were excluded from the electrophysiology analysis (i.e., unstable baseline, access resistance >10% of the membrane resistance). Statistical tests used for each experiment and exact $p$ values are present in the figure legends. In figures, *$p < 0.05$, **$p < 0.01$, ***$p < 0.001$, and ****$p < 0.0001$, ns $p > 0.05$. Samples can be found within figures or in figure legends and are notated: n/N = cell number/animal number. Averages are graphically expressed as arithmetic mean +/- SEM.

### Reporting summary
Further information on research design is available in the Nature Portfolio Reporting Summary linked to this article.

## Data availability
All fiber photometry data have been deposited at Zenodo (https://zenodo.org/records/15420702) and is publicly available as of the date of the publication. Source data generated in this study are provided in the Source Data file. Raw data will be provided upon request. Any additional information required to reanalyze the data reported in this paper is available from the lead contact upon request (Jason Aoto; jason.aoto@cuanschutz.edu). Source data are provided with this paper.

## Code availability
Guppy, a python toolbox for the analysis of fiber photometry data is publicly available (https://github.com/LernerLab/GuPPy). Custom MATLAB scripts for fiber photometry are publicly available (https://github.com/BeierLaboratory/FiberPhotometry). For scripts and step-by-step user guide see citation 60: Sherathiya, V. N. et al., 2021.

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

## Acknowledgements
This work was supported by the National Institutes of Health Grants R01MH116901 (to J.A.), R21MH129620 (to J.A.), F30DA057053 (to C.N.M.), 5T32GM007635 (to C.N.M.), and 5T32GM008497 (to C.N.M.); NIH R01DA054374 (to K.T.B), R01DA056599 (to K.T.B), R01NS130044 (to K.T.B), TRDRP T31KT1437 (to K.T.B) and T31IP1426 (to K.T.B), and One Mind OM-5596678 (to K.T.B). We thank Dr. Michael Mesches and Dr. Nicolas Busquet for providing their expertise regarding the behavioral core usage, experimental design, and analyses. We also thank Hannah Polatsek for her contributions to mouse behavioral data collection and members of the Aoto lab for thoughtful discussions.

## Author contributions
C.N.M. performed AAV injections, electrophysiology experiments, elevated zero maze behavior experiments, histology experiments, and performed data analyses. Y.L. performed the fiber photometry data acquisition and analysis. Y.L. performed the forced swim test and open field test experiments and conducted their analyses. K.T.B. performed fiber photometry analysis and aided in fiber photometry data interpretation. C.N.M. and J.A. are responsible for the study conception, experimental design, data interpretation, and manuscript generation. All authors provided input on the manuscript.

## Competing interests
The authors declare no competing interests.
