## [Transparent Peer Review file · Nature Communications]

Acute stress causes sex-specific changes to ventral subiculum synapses, circuitry, and anxiety-like behavior

Corresponding Author: Dr Jason Aoto

Version 0:

Reviewer comments:

Reviewer #1

(Remarks to the Author)

In this manuscript, Miller and colleagues investigate changes in the function of the ventral subiculum in male and female mice following acute stress. Using electrophysiology, they find that acute restraint stress causes changes to both excitatory and inhibitory synapses in the ventral subiculum. Interestingly, these changes occur in opposite directions in males and females, and in females last up to a week following stress. They find that these changes occur in BNST-projecting vSub neurons, and using fiber photometry find corresponding changes in the activity of BNST-projecting vSub neurons during and after restraint stress. Finally, the authors use systemic administration of either adrenergic antagonists or a blocker of glucocorticoid synthesis to show that the effects of acute stress on vSub synapses is dependent on adrenergic activity in females and HPA activation in males.

Overall, this study is very interesting and has the potential to be impactful on the field. The stark sex differences that the authors found are fascinating and would likely be of high interest to stress researchers. The ventral hippocampus is known to be a key regulator of emotional states and behavior, but very little work has focused on the ventral subiculum-this study addresses that gap. However, while there is a lot of potential here, I have significant concerns with this study as currently presented, most significantly the low number of experimental animals and discrepancies in the ages of animals used in different parts of the study. I have listed my concerns, as well as a number of minor improvements needed in the writing below.

1. The age of animals used is inconsistent throughout the study. For electrophysiology studies, mice were between 42 and 60 days old. While the paper states that these mice were "adults", these are actually adolescents, and this range covers quite a wide swathe of this period. Stress responses, limbic circuitry and effects of sex are highly dynamic over this period, and a 42-day old animal is unlikely to be the same as a 60-day old animal. It is unclear how this range is distributed over the experiments presented, and this should be clarified.
2. A related issue is that while the mice used for electrophysiology are adolescents, fiber photometry animals were fully adult and the time of the experiment. These two time points can really not be assumed to be the same. Authors should demonstrate that their key electrophysiological findings can be replicated in adult animals (or that their photometry findings can be replicated in adolescents, but this is likely far more challenging)
3. For all electrophysiology studies, the number of animals is too low and statistical accounting of cells taken from the same animal is inappropriate. Generally 3-4 animals are used per group for each data set, this is really too low-stress can be highly variable and I typically expect to see n's of 6-10 mice/group. Because the variable in question is what happens to the animal, cells from the same animal cannot be considered independent observations and considering them so violates assumptions underlying statistical tests. Authors should either collapse multiple cells into a single data point per animal or use nested statistics.
4. In the fiber photometry experiment, were any differences in struggling behavior seen between male and female animals?
5. For Figure 6, please show individual animal data points in f-h and m-o.
6. It would also be helpful to indicate in 6C and 6J that the data is being analyzed time locked to struggling bouts.
7. The methods state that metyrapone was dissolved in DMSO, yet the figures imply that the vehicle control was done with saline. Was a DMSO treated group included? This should be corrected if not-DMSO can have significant effects on animals on its own.
8. The manuscript focuses extensively on sex differences, however the data between males and females is never directly compared. For the most part, the differences are qualitative rather than quantitative so I think this is ok. However, in figure 6, the authors report a quantitative change-they find that calcium activity in the aBNST-projecting vSub cells during stress is

greater in males than in females. This should really be directly tested statistically using an ANOVA.

9. Description and discussion of figure 7 and 8 needs to be clearer about potential mechanisms-systemic administration of either drug is likely to affect a wide range of phenomena throughout the entire body, not just glucocorticoid and adrenergic signaling in the subiculum.

10. The authors repeatedly refer to changes in the “E/I balance” in the vSub, however this was never directly measured-they should limit their description of results to what was actually found in the experiments.

11. (Minor) The authors refer several times to previous findings that the ventral subiculum is sexually dimorphic-it would be helpful to briefly describe these findings for the readers.

12. (Minor) Please be careful with precision of language throughout the manuscript. An example of this is the phrase “sexually dimorphic”. Not every sex difference is a dimorphism, this should only refer to phenomena or structures that occur in one sex but not the other without overlap. Similarly, the sympathetic nervous system is primarily used to refer to peripheral signaling, effects of LC innervation of the CNS would not be considered part of this.

13. (Minor) Line 144-the authors state “how stressors impact PV synaptic inhibition in any brain region remains unexamined”. This is untrue, please see extensive work from the labs of Laurence Coutellier and James Herman among others

14. (Minor) Line 206-this subheading should be reversed, the data presented demonstrates that a disproportional number of aBNST-projecting cells from the vSub are BS, not the other way around

15. (Minor) Line 209- please provide citations for the role of the vSUB-aBNST circuit playing a role in stress hormone release and anxiety.

16. (Minor) Line 231-please be specific about findings in this section. Fiber photometry is a measure of calcium activity, an indirect proxy of neuronal activity

17. (Minor) I would recommend using a different abbreviation for “adrenergic receptor” then AR-I know this is technically correct but it is quite confusing given that the authors abbreviate “acute restraint stress” as “ARS”

18. (Minor) Line 388-please use a different phrase than “network activity”, this is not what is measured by fiber photometry

Reviewer #2

(Remarks to the Author)

In the manuscript by Miller et al, the authors investigated the impact of acute restraint stress (ARS) on the sexually dimorphic ventral subiculum (vSUB) local circuit and its output to the anterior bed nucleus of the stria terminalis (aBNST) as well as the impact of this circuitry on manifestation of anxiety-like behaviors. The authors observed sex-specific synaptic adaptations to parvalbumin (PV) expressing inhibitory interneurons-mediated inhibition in the vSUB in presence of acute stress wherein ARS potentiated inhibitory PV-BS synapses in female mice (immediate and long-lasting) and decreased PV-BS inhibition (only long-lasting) in male mice. Mechanistically, the stress-induced PV-BS inhibitory adaptations in females were driven by sympathetic nervous system signaling through adrenergic receptors whereas in males, these adaptations were driven by corticosterone signaling. Finally, the authors observed that ARS differentially affected anxiety-like behaviors in a sex-dependent manner. A detailed investigation of the microcircuit organization in the vSUB can provide us with novel insights into how sex can be an important predisposing factor to psychiatric disease etiology. The electrophysiological studies and analyses are quite comprehensive and rigorous. They analyzed input/output relationships, paired-pulse ratios, electrically-evoked postsynaptic currents, and long-term potentiation (only CA1-vSUB synapses) on two types of synapses. However, excitement of these findings is dampened by the lack of extensive behavioral analyses and pathology analyses in the vSUB (spine morphology, synapse density, expression levels of synaptic proteins) to confirm a role for ARS in vulnerability to psychiatric disorders. Furthermore, the inference that alterations in activity of microcircuit in vSUB can contribute to pathogenesis of psychiatric disorders is over-reaching. We have now listed our comments that can potentially improve the quality of the manuscript.

Major concerns:

1) The authors reported that acute restraint stress (ARS) induced anxiety-like behavior in female, but not in male mice using the elevated zero maze (EZM). The employment of a single behavioral paradigm and the inference that acute stress impacts synapses in the ventral hippocampus is a gross over-interpretation of the data. Conducting two more behavioral paradigms that measures anxiety-like behavior would add more robustness and validity to the authors' conclusions. Additionally, the authors should also report the amount of time that mice spend in the open arms and conduct an ANOVA analysis to determine the effect of arm on anxiety and whether there is a significant interaction between stress and arm on anxiety. This would be a more descriptive and holistic analysis of mice's response to an anxiogenic-inducing environment. Mice have a natural fear for large, open spaces and the current analysis of behavioral data does not represent this innate behavior of mice. From the current analysis, we do not know if female mice subjected to acute stress spend more time than the control group in the open arms. Additionally, the EZM is not an ideal paradigm to measure locomotor activity due to smaller surface area of the maze. Measuring locomotor activity in an open field test would be a more reliable measure of this activity.

2) For statistical analysis, the authors conducted statistical comparisons within each sex. Was there any specific reason to not conduct an ANOVA analysis by using both stress and sex as variables? Although the sex differences can be interpreted from observing the data within each sex independently, it would be more helpful to a direct comparison between male and female for the electrophysiological and behavioral analyses. This would add more robustness to statistical comparisons and results.

3) For the experiment that evaluated sex-specific mechanisms underlying stress-induced adaptations in PV-BS inhibition (Fig 7e-l) and (Fig 8e-l), it appears that both the control and the stress group received the pharmacological treatment, and a side-by-side comparison was undertaken between the two groups. However, it would have more sense if there were four groups for these experiments which would be Control + Saline, Control + drug, Stress + saline, and Stress + drug and to

conduct a two-way ANOVA analysis to determine the effects of the pharmacological agent on PV-BS inhibition. This would make it easier for the readers to assess a side-by-side comparison of how a specific pharmacological treatment can impact PV-BS inhibition in each sex. In the current version, the readers have to refer to an earlier figure (Figs 3 & 4) to see how PV-BS inhibition is altered in each sex and connect the dots on their own. In the current draft, we do not know if these pharmacological agents have any independent effects on PV-BS inhibition irrespective of stress induction. We suggest the authors to discuss these agents and provide any evidence that these drugs do not have any independent effects on the inhibitory activity of PV-BS synapses in the vSUB. Additionally, conducting a two-way ANOVA analysis and determining if there is any interaction between stress and drug would add more robustness and reliability to the data and inferences.

4) Is there any specific reason as to why the authors did not measure NMDA/AMPA ratio to address alterations in postsynaptic transmission at vCA1-vSUB synapses for Figs 1 & 2 since the vSUB receives glutamatergic input from the vCA1 region. It would be of interest to see if there are any changes in number of synapses (excitatory or inhibitory) or expression levels of synaptic proteins in the vSUB through cell biology or molecular biology methods. These studies would reinforce the authors' hypothesis that the differential effects of acute stress on synaptic function in the vSUB would contribute to pathogenesis of psychiatric disorders.

5) The authors suggested that the vSUB region can participate in coping behavior and represent a parallel pathway to induce struggling behavior. Have the authors considered undertaking tests that measure depressive behavior in response to acute stressors such as forced swim test or tail suspension test to investigate this hypothesis?

6) In the discussion section, the authors do not provide a potential mechanism underlying the sex-dependent effects of acute stress on anxiety behavior. It would be helpful if the authors link the electrophysiological findings to behavioral responses upon induction of acute stress since they observed sex differences in PV-BS inhibition in the vSUB.

7) For Fig 1 & 2, for panel 1h (currently 6 cells from 4 stressed females), do the authors think that increasing the sample size, would diminish the significance of results observed in Fig 2h (currently 11 cells from 4 stressed males). This is important for the author's point (in comment # 2) in regard to sex dimorphism and also address the same point that an analysis of sex and stress as two variables for ANOVA analysis would be crucial.

8) The authors specifically talk about the sexual dimorphic responses to ARS in the context of anxiety behaviors. However, psychiatric disorders represent a myriad of symptoms including social deficits, impaired cognitive functioning, reward deficits, perseverative behavioral patterns, reduced goal-directed behavior etc. Does the vSUB only regulate anxiety behavior or can potentially regulate other behaviors that are relevant to psychiatric disorders?

9) The female mice exhibited anxiety-like behavior 1 week post ARS. Considering that this is a significantly long interval of time, did the authors measure corticosterone levels in blood of female and male mice in the presence/absence of acute stress? I also wonder if there are any changes in synapse density (excitatory or inhibitory), expression levels of synaptic proteins (PSD-95, AMPA receptors, NMDA receptors) after 1 week post ARS. It would be helpful if the authors could provide any anatomical or biochemical evidence that acute stress can impact synapse density, spine morphology, or expression levels of synaptic proteins linked to psychiatric disease pathology.

10) The authors put forth the observation that acute stress can be encoded at the synapses in a sex-specific manner, and this could be potential mechanism underlying sex-dependent susceptibility to psychiatric disorders in the introduction section (lines 51-53). However, the cited articles (#5-8) do not fully support this argument since it only pertains to a specific disorder (post-traumatic stress disorder) which makes it a weak statement. They only cite data from clinical studies. The authors should provide clinical evidence pertaining to other psychiatric disorders and preclinical evidence that acute stress can differentially induce behavioral alterations in a sex-dependent manner since the authors also make the statement that different acute stress paradigms can induce sex-specific changes in behavior in mice (lines 386-387).

Minor comments:

1) Throughout the results section, the authors do not mention which statistical test was used for which experiment. The P values are reported in the figure captions, but the authors should also report "F" and "df" values derived from the ANOVA analysis in the results sections. It would be helpful if the authors report the P, F, and df values from ANOVA analysis and P values from Student's t-test in the results section. Furthermore, significant changes in EPSC amplitude were reported for Fig 3d, 4d, 5i, and 5k. Were these P values derived from ANOVA analysis or from a post-hoc test?

2) The authors should perform a quality control check for all of their figure captions for the main text figures and supplementary figures. There are a couple of errors in both the main text figure captions and supplementary figure captions. For instance, for Fig 5, panel "i" refers to EPSC data in female mice but on line 1005, it says panel "k". For Fig 6, panel "w" refers to distance travelled by female mice, but on line 1033, it says panel "x". For Fig 7, panel (f,j) on line 1045 is not bolded unlike the other panel designations in Fig 7 and in other figures. For Fig 8, panel "i" and "j" refers to IPSC data for sympathetic and corticosterone signaling experiments. However, it says panel "e" on line 1065 and panel "i" on line 1066. On line 1032, "P" is not italicized. There are also minor errors in the figure captions for the supplementary figures that needs to be rectified.

3) In the statistical analyses section, "P" is not capitalized but it is capitalized in the figure captions. There should be consistency in regard to the P value being capitalized or not.

4) For all the figure captions (main text & supplementary), it should read as “unpaired Student’s t-test” instead of “unpaired t-test”. The “t” needs to be italicized. The authors should also state if they used repeated measures ANOVA or not in the figure captions.

5) On lines 453-454, it should read as “All injectable solutions were freshly prepared on the day of the experiment”. At the moment, the sentence ends with “of” that makes the sentence sound incomplete.

Reviewer #3

(Remarks to the Author)

This study provides critical insights into the sex differences in stress adaptation at the synaptic and circuit levels, advancing our understanding of the mechanisms underlying stress-related psychiatric disorders like anxiety and depression. The findings have broad implications for developing sex-specific treatments for psychiatric diseases.

The study investigates the sex-specific effects of acute stress on the ventral subiculum (vSUB), a region linked to stress regulation, emotional processing and psychiatric disease. Using a combination of electrophysiology, optogenetics, and behavioral assays, the authors show that stress induces distinct changes in synaptic function and neuronal circuitry in male and female mice.

The majority of these experiments are technically well performed, and the results are informative. However, there are several concerns with the logic of some of the experiments and the interpretation of the results. Specifically, the study did great work in revealing sex-specific synaptic and circuitry changes in vSUB after 1 hour acute restraint stress. However, the data does not support the conclusion that acute stress causes sex-dependent anxiety-like behavior.

In its current form, the study is not suitable for publication in Nature Communications.

Major concerns:

1, All studies were done in ≤ 2 month old mice (line 486, line 536), which are not strictly considered as adult. According to the Jackson Laboratory, mature adult mice should be at least three months old because, although they are sexually mature by 35 days, relatively rapid maturational growth continues for most biological processes and structures until about three months. We are aware that patch-clamp experiments are more complicated in mature mice (> 3 months old). However, this relatively immature age of the mice used raised questions as to whether similar findings for behavioral outcomes would hold true in mature mice.

2, The rationale for examining synaptic change 24 hours after acute restraint stress is unclear. Why this specific time point was chosen? Whether this synaptic change in CA1-vSUB and vSUB-aBNST (Fig 1,2 and 5) connections is transient or long lasting, for instance, for days or even weeks, is not investigated. This comes to very crucial if the authors claim the long-term behavioral change in anxiety-like behavior, which most likely need a long-lasting synaptic plasticity change or circuit modulation among the vCA1, vSUB and aBNST.

Nevertheless, the authors have considered to perform such experiments in examining PV-BS IPSCs after a week. However, whether the local microcircuit connectivity shift affects long-range vCA1-vSUB-aBNST projections, which are involved in anxiety and stress behaviors, remain unclear. Long-term changes, such as those occurring over days or weeks, should be investigated to support claims about persistent behavioral effects.

3, Regarding the EZM behavioral test, the authors used white noise during habituation and experimentation. What is the sound level and frequency of this white noise? Could white noise itself be another type of stressor? Classically, anxiety tests, such as EPM, EZM or open field test, are carried out without such a background noise or even in a sound-attenuated arena.

Furthermore, the authors decide to analyze the first 5 minutes of the EZM and claimed this period as the most anxiogenic response period. However, the reference given “100. The relation between fear induced by novel stimulation and exploratory behavior - PubMed. Journal of comparative and physiological psychology 48 (1955 Aug). <https://doi.org/10.1037/h0043788>” does not support such claim. The apparatus in this paper is not an EZM and did not test anxiety. It is a misuse of a citation. The EZM was invented in 1990s, 40 years after the cited study. The authors must provide clear reasons to select only the first part of the behavior test (are results not holding afterwards?). Mouse behavior during the entire 10 minutes session should be analyzed.

4, The argument that stress causes anxiety-like behavior in females is not compelling. Previous study showed no gender bias regarding the time spent in close arm in EPM or EZM (Tucker LB, McCabe JT. Front Behav Neurosci. 2017 doi: 10.3389/fnbeh.2017.00013). However, the female control mice seem to stay less time in close arm than male control mice (fig 6s,x). It is thus reasonable to pool fig 6q and 6s together, as the same for fig 6v and 6x, to confirm whether ARS induced anxiety-like behavior in female mice, or ARS brought ‘braver or more explorative’ female mice back to normal behavior. A more sophisticated experiment for such claim would need to compare anxiety-like behavior on EZM for the same cohort of mice before and after ARS, to avoid biased behavior among groups.

5, The coordinates (AP -4.04; ML 3.4; DV -3.5) used for GCaMP7f virus injection for fiber photometry is clearly off target. The virus transduced the STR (subiculum, transition area), other than vSub. There are also many CA1 neurons transduced, as shown in the sampling image in Fig 6a. As BNST also receive projections from CA1, post hoc immunohistochemistry must be used to verify viral labeling specificity, for instance clearly showing the tip of fiber sitting above vSub. Furthermore, as there is no clear border between vCA1 and vSub, a thinner fiber 200 μ m will do much cleaner job than a 400 μ m fiber does.

Furthermore, calcium signal recordings started two weeks after implantation of a 400um fiber. It is not an ideal time as there are still inflammation going on after fiber implantation. Common procedure for such calcium activity recording is to wait at least 4 weeks. Otherwise, neurons can show abnormal hyperactivity, which can hinder proper conclusions.

6, It is necessary to consider also the elevated plus maze and open field tests to confirm the findings in EZM.

7, We have concerns about the variations of two cohorts of animals in figure 7e-l and fig 8e-l. If we are not mistaken, the controls in fig 7f-h and 7j-l are both injected with saline. I expect similar electrical properties of these two control cohorts. However, the values of slope, aIPSC and PPR are largely different between them. The same difference as observed in fig 8f-h vs fig 8j-l for male mice. This variation in control experiments likely indicates instability of the experimental settings, conductance measurements, or strong variability across different mouse cohorts.

Minor

1, Detailed methods for fiber photometry should be documented, such as sampling rate, excitation power, calcium traces filtering, peak detection etc.

2, ref 17, wherever possible, please cite peer-reviewed article rather than non-peer reviewed biorxiv.

3, line 208-209, "The vSUB-aBNST circuit is known to be critical for stress hormone (i.e. corticosterone) release and anxiety-like behavior," References are missing for this claim.

4, CA1 and vCA1 are both used throughout the text, please make sure they are accurately used.

Reviewer #4

(Remarks to the Author)

Reviewer #5

(Remarks to the Author)

Version 1:

Reviewer comments:

Reviewer #1

(Remarks to the Author)

The authors have responded to many of my concerns, and I commend them for doing challenging experiments to demonstrate that the effect of stress was age-independent. However despite many improvements to the manuscript, the fact remains that an n of 3 or 4 animals is far too low for a study where the variable is something that happens to the animal itself. The fact that other papers have been published in this journal with similar ns does not change my stance on this. Had I reviewed these papers I would have also seen them as underpowered, if I encountered them as a reader I would consider them premature findings and would hesitate to use them to inform my own work. Without increasing the n's in this study to an appropriate sample size I do not think it reaches the standards of this journal.

Reviewer #2

(Remarks to the Author)

This study provided a detailed and comprehensive analysis of the microcircuit organization in the vSUB. The authors carefully addressed each of the 10 previous major concerns and provided additional analyses and experimental data. These new data strengthened the observations of anxiety-like and depressive behaviors. The newly added analyses provided additional clarification on the PV-BS inhibition effects in both control and stress groups. Finally, the authors carefully addressed all of the minor comments with revisions and expanded literature citations.

Reviewer #3

(Remarks to the Author)

We acknowledge the authors' efforts in providing additional analyses to support their hypothesis. While most of our concerns have been addressed, the critical issue regarding the age of the experimental mice remains unresolved. The authors refer to their subjects as "sexually mature adolescent mice" (PND42-60) or "adult mice" (PND70-77), citing Brust et al., 2015, from *Frontiers in Zoology* (which is a review other than research article systematically examine sexual development). However, we still disagree with this classification.

It is well-established among researchers working with laboratory mice that animals aged PND42-60 are still undergoing rapid growth. During this period, significant changes occur in body size, skeletal structure and brain development. For

instance, a mouse at PND42 weighs approximately 70% of its weight at PND90. The extent of their sexual maturity at this stage is questionable, or at least not robust and being debated. The research community more commonly considers mice to be adults at around 3 months of age (PND90 or older), other than PND70-77 in this study.

Another concern is male and female have slightly different development timeframe, so we do believe a time point at fully adult age is necessary. Findings at adolescent age PND42 might be also influenced by neurodevelopment factors.

Given that the authors aim to address 'sexually dimorphic' questions, it is essential to use non-questionable, well-accepted adult ages. We believe that using adult mice aged 3-6 months is necessary to address scientific questions involving sex-specific changes, neural circuits and behavior.

We recommend that the authors replicate their key findings using well-accepted adult mice aged 3-6 months to substantiate their claims. Sex-specific studies should involve clearly recognized adult subjects rather than adolescent or early adulthood mice. We acknowledge that certain techniques, such as patch-clamp recordings, are challenging in 3-6-month-old mice and may not be necessary. However, behavioral assessments, neural circuit tracing and calcium imaging should be conducted in mice within this mature adult age range.

This study has the potential to make a significant contribution to the research field. However, the age of the research subjects currently undermines the strength of the conclusions. We hope the authors can provide robust data from appropriately aged subjects to support their claims.

We would like to thank the reviewers for their thoughtful comments. We were encouraged by the enthusiasm for the study and that all reviewers agreed that the findings will be impactful in the field. We have endeavored to address the experimental concerns raised by the reviewers. Briefly, we replicated our key vSUB local circuit findings in adult (PND70-77) animals and found that the sex-specific synaptic adaptations reported in sexually mature adolescent female and male (PND42-60) following acute restraint stress were present in adult animals (Supplementary Figs 1-2). Second, we repeated the *in vivo* fiber photometry experiment in PND45-55 female and male mice (Fig. 5). Similar to what we initially observed in PND70-77 mice, males exhibited heightened Ca^{2+} activity during the stress recovery period while female Ca^{2+} activity returned to baseline. Together, these data indicate that the sex-specific synaptic adaptations to the vSUB local circuit following acute restraint stress is not restricted to a single time window but instead occurs equally in adolescent and adult mice. Third, we expanded our behavioral analysis by performing the open field test as a secondary approach to measure anxiety-like behavior, and the forced swim test to measure helplessness and passive coping behaviors (Supplementary Fig. 7). Fourth, we added the stress+vehicle control conditions to our pharmacological interrogation into the relevant signaling pathways involved in the stress-induced sex-specific adaptations (Figs. 7 and 8, Supplementary Figs. 8-9). Additionally, all of our electrophysiological experiments were re-analyzed with nested statistics, which identified similar sex-differences in response to stress. We hope that the reviewers will agree that the manuscript is significantly improved, and acceptable for publication in Nature Communications.

Reviewer #1 (Remarks to the Author):

In this manuscript, Miller and colleagues investigate changes in the function of the ventral subiculum in male and female mice following acute stress. Using electrophysiology, they find that acute restraint stress causes changes to both excitatory and inhibitory synapses in the ventral subiculum. Interestingly, these changes occur in opposite directions in males and females, and in females last up to a week following stress. They find that these changes occur in BNST-projecting vSub neurons, and using fiber photometry find corresponding changes in the activity of BNST-projecting vSub neurons during and after restraint stress. Finally, the authors use systemic administration of either adrenergic antagonists or a blocker of glucocorticoid synthesis to show that the effects of acute stress on vSub synapses is dependent on adrenergic activity in females and HPA activation in males.

Overall, this study is very interesting and has the potential to be impactful on the field. The stark sex differences that the authors found are fascinating and would likely be of high interest to stress researchers. The ventral hippocampus is known to be a key regulator of emotional states and behavior, but very little work has focused on the ventral subiculum- this study addresses that gap. However, while there is a lot of potential here, I have significant concerns with this study as currently presented, most significantly the low number of experimental animals and discrepancies in the ages of animals used in different parts of the study. I have listed my concerns, as well as a number of minor improvements needed in the writing below.

We are excited that the reviewer finds the study very interesting and that it has the potential to be impactful on the field. Further, we appreciate that the reviewer acknowledges that the nature of the study will be of high interest to researchers in the stress field and in the field of ventral subiculum. We hope that our responses will address the major concern regarding the number of animals used per experiment and that our logical experiments will address the discrepancies in animal ages.

1. The age of animals used is inconsistent throughout the study. For electrophysiology studies, mice were between 42 and 60 days old. While the paper states that these mice were “adults”, these are actually adolescents, and this range covers quite a wide swathe of this period. Stress responses, limbic circuitry and effects of sex are highly dynamic over this period, and a 42-day old animal is unlikely to be the same as a 60-day old animal. It is unclear how this range is distributed over the experiments presented, and this should be clarified.

To address this concern, the exact ages of the mice used per experiment are included in supplemental data files alongside their animal averages. The PND42-60 range was selected because it encompasses sexually mature mice and spans the end of mid adolescence (PND35-47) and late adolescence (PND48-60) (Brust et al., 2015) as defined by the Jackson Laboratory. To further address this concern, we repeated these key electrophysiological studies of basal synaptic transmission in the vSUB local circuit (vCA1-vSUB and PV-vSUB) in adult mice (PND70-77). Our findings in sexually mature PND 42-60 mice hold true for fully adult mice (Figures S1, S2). We report at both time periods that acute restraint stress enhances PV-mediated inhibition and reduces excitatory synaptic strength onto BS neurons in females. In males, the same stress protocol selectively reduces PV-mediated inhibition of BS cells. We agree

with the reviewer that our electrophysiology data are collected from a wide swathe of ages (PND42-60), however, we find the reproducibility of the phenotypes across this age range and in older (PND70-77) animals encouraging that the sex-specific synaptic adaptations that we identified here are robust and not specific to a particular time point.

2. A related issue is that while the mice used for electrophysiology are adolescents, fiber photometry animals were fully adult and the time of the experiment. These two time points can really not be assumed to be the same. Authors should demonstrate that their key electrophysiological findings can be replicated in adult animals (or that their photometry findings can be replicated in adolescents, but this is likely far more challenging)

We appreciate the reviewer's comment regarding the age discrepancies between our electrophysiology data and our *in vivo* fiber photometry measurements. We also appreciate the acknowledgement that fiber photometry in younger animals is technically challenging. To strengthen our arguments, we replicated the fiber photometry in younger (PND45-55) mice and replicated our key electrophysiology experiments (vCA1-vSUB and PV-vSUB) between PND70-77. Remarkably, the findings we reported hold true in PND 42-60 mice and PND 70-77 mice for both the electrophysiology and fiber photometry experiments (Figures S2, S4, S6). In short, we observe increased calcium activity in males following motion after stress than before in sexually mature adolescent and adult mice.

3. For all electrophysiology studies, the number of animals is too low and statistical accounting of cells taken from the same animal is inappropriate. Generally 3-4 animals are used per group for each data set, this is really too low-stress can be highly variable and I typically expect to see n's of 6-10 mice/group. Because the variable in question is what happens to the animal, cells from the same animal cannot be considered independent observations and considering them so violates assumptions underlying statistical tests. Authors should either collapse multiple cells into a single data point per animal or use nested statistics.

We re-analyzed all of our experimental data using nested statistics as suggested by the reviewer. All reported electrophysiological experiments in the main and supplemental figures remain statistically significant. Despite our best efforts to record from the same number of regular- and burst-spiking cells per animal, there was the expected variability in the number of healthy cells monitored that varied experiment-by-experiment. By presenting our data as a per animal average, the animals where fewer cells were measured will be overrepresented. We feel that presenting our data as a per cell average is advantageous because it is more straightforward and more faithfully represents the collected data. Here are examples of papers studying stress published in Nature Communications in the last 4 years with electrophysiology data represented as cells and from similar animal numbers (Liu et al., 2020; Zhang et al., 2021; Linders et al., 2022; Rodrigues et al., 2022; Guo et al., 2024; Lee et al., 2024; Zhang et al., 2024).

4. In the fiber photometry experiment, were any differences in struggling behavior seen between male and female animals?

The struggling behaviors were defined by the same parameters in male and females and no overt sex-differences were observed (see below figure).

5. For Figure 6, please show individual animal data points in f-h and m-o.

We changed the fiber photometry graphs in PND42-60 and PND70-77 mice to show individual animal points behind the animal averages.

6. It would also be helpful to indicate in 6C and 6J that the data is being analyzed time locked to struggling bouts.

We have embedded descriptive labeling in the figure to indicate this.

7. The methods state that metyrapone was dissolved in DMSO, yet the figures imply that the vehicle control was done with saline. Was a DMSO treated group included? This should be corrected if not-DMSO can have significant effects on animals on its own.

We apologize to the reviewer for the vague description in the text and our mislabeled informatic panel. The reviewer is correct - the control group for the metyrapone experiments was injected with DMSO. We have updated the figure to add clarity for which vehicle was used in the pharmacological studies. We found that DMSO alone did not alter PV-vSUB synaptic transmission (Fig 7-8 & S8-9).

8. The manuscript focuses extensively on sex differences, however the data between males and females is never directly compared. For the most part, the differences are qualitative rather than quantitative so I think this is ok. However, in figure 6, the authors report a quantitative change-they find that calcium activity in the aBNST-projecting vSub cells during stress is greater in males than in females. This should really be directly tested statistically using an ANOVA.

We agree with the reviewer and ANOVA analyses do not reveal statistical significance, so we removed this claim from the manuscript.

9. Description and discussion of figure 7 and 8 needs to be clearer about potential mechanisms-systemic administration of either drug is likely to affect a wide range of phenomena throughout the entire body, not just glucocorticoid and adrenergic signaling in the subiculum.

We agree that the systemic administration of pharmacologic agents is acting on many brain regions in addition to the ventral subiculum and that the effects we observe may not be due to direct action on these receptors in the ventral subiculum. We further highlight this possibility and discuss how our findings maybe downstream results of other circuit changes.

10. The authors repeatedly refer to changes in the “E/I balance” in the vSub, however this was never directly measured-they should limit their description of results to what was actually found in the experiments.

The reviewer is correct that we did not directly measure E/I balance and we changed this language to “excitatory and inhibitory inputs.”

11. (Minor) The authors refer several times to previous findings that the ventral subiculum is sexually dimorphic-it would be helpful to briefly describe these findings for the readers.

We included this brief statement in the text at line 68 and 69 “exhibit basal sex differences in basal PV inhibitory strength and connectivity (Boxer 2021 Cell Reports).”

12. (Minor) Please be careful with precision of language throughout the manuscript. An example of this is the phrase “sexually dimorphic”. Not every sex difference is a dimorphism, this should only refer to phenomena or structures that occur in one sex but not the other without overlap. Similarly, the sympathetic nervous system is primarily used to refer to peripheral signaling, effects of LC innervation of the CNS would not be considered part of this.

We re-examined our usage of “sexual dimorphism” and changed to sex difference where appropriate. We changed the use of “sympathetic nervous system” to “adrenergic receptor signaling” in response to this comment.

13. (Minor) Line 144-the authors state “how stressors impact PV synaptic inhibition in any brain region remains unexamined”. This is untrue, please see extensive work from the labs of Laurence Coutellier and James Herman among others

We appreciate the reviewer for pointing our mistake we are familiar with the studies from these laboratories. We revised this statement to “Despite this, the impact of stress on PV inhibition in vSUB at the synaptic level remains unexamined” (lines 127-128).

14. (Minor) Line 206-this subheading should be reversed, the data presented demonstrates that a disproportional number of aBNST-projecting cells from the vSub are BS, not the other way around

The line on 189 reads “vSUB-BS cells disproportionately project to aBNST.” This appears to already convey the information the reviewer is suggesting.

15. (Minor) Line 209- please provide citations for the role of the vSUB-aBNST circuit playing a role in stress hormone release and anxiety.

The vCA1/vSUB-aBNST circuit regulates corticosterone release (Cole et al., 2022) and aspects of anxiety-like behaviors (Glangetas et al., 2017; Urien et al., 2022; Kopaeva et al., 2023). We added these citations in line 192.

16. (Minor) Line 231-please be specific about findings in this section. Fiber photometry is a measure of calcium activity, an indirect proxy of neuronal activity

We agree with the reviewer and changed “which is a correlative approach to link neuronal activity and behavior” to “To expand on our *ex vivo* findings, we employed fiber photometry in awake, behaving mice to monitor *in vivo* Ca²⁺ signals in aBNST-projecting vSUB cells, which is a measure of Ca²⁺ activity, an indirect proxy of neuronal activity⁵⁶.” In lines 210-211

17. (Minor) I would recommend using a different abbreviation for “adrenergic receptor” then AR-I know this is technically correct but it is quite confusing given that the authors abbreviate “acute restraint stress” as “ARS”

“AR” is a classic abbreviation for adrenergic receptors in the field, so to address this comment we will instead refer to acute restraint stress as “stress” instead of “ARS” to avoid confusion.

18. (Minor) Line 388-please use a different phrase than “network activity”, this is not what is measured by fiber photometry

We have changed lines 418-419 from “network activity” to “*in vivo* Ca²⁺ activity.”

Reviewer #2 (Remarks to the Author):

In the manuscript by Miller et al, the authors investigated the impact of acute restraint stress (ARS) on the sexually dimorphic ventral subiculum (vSUB) local circuit and its output to the anterior bed nucleus of the stria terminalis (aBNST) as well as the impact of this circuitry on manifestation of anxiety-like behaviors. The authors observed sex-specific synaptic adaptations to parvalbumin (PV) expressing inhibitory interneurons-mediated inhibition in the vSUB in presence of acute stress wherein ARS potentiated inhibitory PV-BS synapses in female mice (immediate and long-lasting) and decreased PV-BS inhibition (only long-lasting) in male mice. Mechanistically, the stress-induced PV-BS inhibitory adaptations in females were driven by sympathetic nervous system signaling through adrenergic receptors whereas in males, these adaptations were driven by corticosterone signaling. Finally, the authors observed that ARS differentially affected anxiety-like behaviors in a sex-dependent manner. A detailed investigation of the microcircuit organization in the vSUB can provide us with novel insights into how sex can be an important predisposing factor to psychiatric disease etiology. The electrophysiological studies and analyses are quite comprehensive and rigorous. They analyzed input/output relationships, paired-pulse ratios, electrically-evoked postsynaptic currents, and long-term potentiation (only CA1-vSUB synapses) on two types of synapses. However, excitement of these findings is dampened by the lack of extensive behavioral analyses and pathology analyses in the vSUB (spine morphology, synapse density, expression levels of synaptic proteins) to confirm a role for ARS in vulnerability to psychiatric disorders. Furthermore, the inference that alterations in activity of microcircuit in vSUB can contribute to pathogenesis of psychiatric disorders is over-reaching. We have now listed our comments that can potentially improve the quality of the manuscript.

We appreciate the reviewer’s appreciation regarding the novelty of this work and that our synaptic dissection is rigorous and comprehensive. As discussed below, we have made every effort to address all of the reviewer’s experimental concerns.

Major concerns:

1) The authors reported that acute restraint stress (ARS) induced anxiety-like behavior in female, but not in male mice using the elevated zero maze (EZM). The employment of a single behavioral paradigm and the inference that acute stress impacts synapses in the ventral hippocampus is a gross over-interpretation of the data. Conducting two more behavioral paradigms that measures anxiety-like behavior would add more robustness and validity to the authors' conclusions. Additionally, the authors should also report the amount of time that mice spend in the open arms and conduct an ANOVA analysis to determine the effect of arm on anxiety and whether there is a significant interaction between stress and arm on anxiety. This would be a more descriptive and holistic analysis of mice's response to an anxiogenic-inducing environment. Mice have a natural fear for large, open spaces and the current analysis of behavioral data does not represent this innate behavior of mice. From the current analysis, we do not know if female mice subjected to acute stress spend more time than the control group in the open arms. Additionally, the EZM is not an ideal paradigm to measure locomotor activity due to smaller surface area of the maze. Measuring locomotor activity in an open field test would be a more reliable measure of this activity.

We thank the reviewer for this suggestion. As suggested, we added the open field test (OFT) and found decreased center time exploration in the OFT after stress (lines 286-289). Further, to measure depressive-like behavior, performed the forced swim test and found that stress females have increased immobility which represents increased helplessness and passive coping to stress (Fig S7).

We have conducted the additional analyses suggested and found a significant effect between stress condition and EZM arm time spent for females. Further, we analyzed the total time spent in the open arm as the reviewer suggested and found that stress female mice spend less time in the open arm than control female mice (Fig 6), reinforcing an anxiety-like phenotype.

Finally, we analyzed locomotor behavior in the OFT and found no impact of stress on locomotion (Fig S7).

2) For statistical analysis, the authors conducted statistical comparisons within each sex. Was there any specific reason to not conduct an ANOVA analysis by using both stress and sex as variables? Although the sex differences can be interpreted from observing the data within each sex independently, it would be more helpful to a direct comparison between male and female for the electrophysiological and behavioral analyses. This would add more robustness to statistical comparisons and results.

We agree that it would ideal to perform two-way ANOVA to compare stress x sex. However, given the pragmatic limitations discussed below, we feel that a direct comparison between males and females is inappropriate. For all analyses of a single sex, we used cage and litter-mate matched animals across control and stress group. Unfortunately, it was too logistically challenging to also litter-mate match across sexes and, of course, could not cage-match.

3) For the experiment that evaluated sex-specific mechanisms underlying stress-induced adaptations in PV-BS inhibition (Fig 7e-l) and (Fig 8e-l), it appears that both the control and the stress group received the pharmacological treatment, and a side-by-side comparison was undertaken between the two groups. However, it would have more sense if there were four groups for these experiments which would be Control + Saline, Control + drug, Stress + saline, and Stress + drug and to conduct a two-way ANOVA analysis to determine the effects of the pharmacological agent on PV-BS inhibition. This would make it easier for the readers to assess a side-by-side comparison of how a specific pharmacological treatment can impact PV-BS inhibition in each sex. In the current version, the readers have to refer to an earlier figure (Figs 3 & 4) to see how PV-BS inhibition is altered in each sex and connect the dots on their own. In the current draft, we do not know if these pharmacological agents have any independent effects on PV-BS inhibition irrespective of stress induction. We suggest the authors to discuss these agents and provide any evidence that these drugs do not have any independent effects on the inhibitory activity of PV-BS synapses in the vSUB. Additionally, conducting a two-way ANOVA analysis and determining if there is any interaction between stress and drug would add more robustness and reliability to the data and inferences.

Due to the scale of the experiments and to conserve animal resources, we first examined whether the pharmacological agents alone impacted PV-BS inhibition without stress exposure. These experiments indicated that pharmacological agents alone did not have any independent effects on PV-BS inhibition

in control animals (Fig 7 a-d & Fig 8a-d). However, to aid in the ease of figure interpretation, we added additional groups of stress + vehicle the PV-vSUB inhibition experiments (Fig 7-8, S8-S9).

4) Is there any specific reason as to why the authors did not measure NMDA/AMPA ratio to address alterations in postsynaptic transmission at vCA1-vSUB synapses for Figs 1 & 2 since the vSUB receives glutamatergic input from the vCA1 region. It would be of interest to see if there are any changes in number of synapses (excitatory or inhibitory) or expression levels of synaptic proteins in the vSUB through cell biology or molecular biology methods. These studies would reinforce the authors' hypothesis that the differential effects of acute stress on synaptic function in the vSUB would contribute to pathogenesis of psychiatric disorders.

We apologize for the confusion. We measured postsynaptic strength using strontium-evoked measurements at vCA1-vSUB synapses (Figs. 1e and 2e). Using this approach, we found that postsynaptic strength was reduced in females without changes in males. Further, the frequency of these aEPSCs was unchanged in females (see below). The frequency of aEPSCs can reflect changes in release probability and/or synapse numbers. Our PPR measurements are unchanged between control and stress conditions, which suggests that the number of synapses is also unaltered.

We appreciate the reviewer's suggested experiment to quantify the expression levels of synaptic proteins to test for changes in synapse numbers in acute stress exposed female mice.

While this is an appealing avenue to pursue, our electrophysiology experiments indicate that there are no changes in synapse numbers. Further, we would need to analyze synapses made specifically onto either regular- or burst-spiking neurons. Unfortunately, reliable genetic tools to label regular- or burst-spiking neurons for identification do not exist. An addition to the complexity of trying to isolate regular- or burst-spiking cells in vSUB, the quantification of glutamate receptors (e.g. AMPARs and NMDARs) poses a problem for three key reasons. First, antibodies directed against extracellular epitopes of the obligate NMDAR subunit, GluN1, are not commercially available. Second, while there are reliable antibodies that recognize extracellular epitopes for AMPARs, due to the chemical fixation-induced membrane permeabilization that occurs during IHC (Glynn et al., 2011; Richter et al., 2018; Cheng et al., 2019; Idziak et al., 2023), we would not be able to differentiate between surface and total AMPAR pools. Third, it is unclear how fluorescent synaptic puncta, classically quantified in IHC using diffraction-limited confocal microscopy, contribute to synaptic transmission as puncta fluorescence is comprised of intracellular, non-synaptic, extra-synaptic and synaptic pools of receptors. Using this approach, it is difficult to ascertain whether a change in AMPAR puncta reflects a change in the pool of synaptic AMPARs that sense glutamate. These significant technical limitations and experimental confounds will likely limit our interpretation of protein quantification and may not provide insight into our already rigorous and comprehensive electrophysiological analysis.

There is growing appreciation that the nanoscopic organization of synaptic proteins likely plays a more critical role in defining synaptic transmission properties than total protein level. Specifically, the number, volume, and subsynaptic localization of receptor nanodomains are thought to be critically important for basal synaptic transmission and plasticity (MacGillavry et al., 2013; Nair et al., 2013; Tang et al., 2016; Lloyd et al., 2023). However, even at the nanoscale, not all nanodomains sense neurotransmitter and participate in transmission (Tang et al., 2016). To our knowledge, the application of super-resolution imaging modalities in slice preparations is challenging due to technical limitations. This is certainly an avenue that we are currently pursuing in a future study.

5) The authors suggested that the vSUB region can participate in coping behavior and represent a parallel pathway to induce struggling behavior. Have the authors considered undertaking tests that measure depressive behavior in response to acute stressors such as forced swim test or tail suspension test to investigate this hypothesis?

As recommended by the reviewer, we conducted a forced swim test to investigate whether the model of ARS impacts helplessness and coping behaviors in females (Fig S7). We found that stress females have increased immobility, suggesting that females exhibit a more passive coping style and increased helplessness after stress (lines 290-294).

6) In the discussion section, the authors do not provide a potential mechanism underlying the sex-dependent effects of acute stress on anxiety behavior. It would be helpful if the authors link the electrophysiological findings to behavioral responses upon induction of acute stress since they observed sex differences in PV-BS inhibition in the vSUB.

We appreciate the reviewer's comment - In lines 353-361 of the discussion, we discuss: "Our data indicate that vSUB output to the aBNST, a highly sexually dimorphic region integrated within stress and anxiety circuitry⁷¹⁻⁷⁶, is disproportionately dominated by BS cells. aBNST neurons are predominantly inhibitory and participate in feed-forward inhibition of downstream targets that govern the stress response and anxiety. In rodents exposed to aversive contexts, the activity of aBNST-projecting vSUB neurons decreases¹⁷ while experimentally increasing the activity of aBNST-projecting vSUB neurons decreases corticosterone during restraint stress and reduces anxiety-like behavior^{16,18}. Thus, an intriguing possibility is that the activity of vSUB BS cells play an influential role in the response to stress. Here, we found that stress produces a diametrically opposite net effect on BS cell activity and in vSUB-aBNST output."

7) For Fig 1 & 2, for panel 1h (currently 6 cells from 4 stressed females), do the authors think that increasing the sample size, would diminish the significance of results observed in Fig 2h (currently 11 cells from 4 stressed males). This is important for the author's point (in comment # 2) in regard to sex dimorphism and also address the same point that an analysis of sex and stress as two variables for ANOVA analysis would be crucial.

For Fig 1h, we added animals to the experimental groups to 12 stress RS cells and 15 controls RS cells and ensured this effect is not driven by low sample size. We are encouraged by the robustness of the effect as it remains when reanalyzing with nested statistics and is also observed in when repeated in older animals (PND 70-77) (Fig S1).

8) The authors specifically talk about the sexual dimorphic responses to ARS in the context of anxiety behaviors. However, psychiatric disorders represent a myriad of symptoms including social deficits, impaired cognitive functioning, reward deficits, perseverative behavioral patterns, reduced goal-directed behavior etc. Does the vSUB only regulate anxiety behavior or can potentially regulate other behaviors that are relevant to psychiatric disorders?

The vSUB is implicated in the pathogenesis of many psychiatric disorders including anxiety, major depression, substance use disorder relapse, bipolar disorder, and schizophrenia (Baset and Huang, 2024). We focused on anxiety-like behavior because it is a common symptom of acute stress disorders, has robust sex-differences in prevalence, and is relatively easy to measure. However, understanding how acute stress impacts behavioral measurements relevant to other psychiatric conditions will be valuable to understand as a follow up to this study and exceeds the scope of the current study.

9) The female mice exhibited anxiety-like behavior 1 week post ARS. Considering that this is a significantly long interval of time, did the authors measure corticosterone levels in blood of female and male mice in the presence/absence of acute stress? I also wonder if there are any changes in synapse density (excitatory or inhibitory), expression levels of synaptic proteins (PSD-95, AMPA receptors, NMDA receptors) after 1 week post ARS. It would be helpful if the authors could provide any anatomical or biochemical evidence that acute stress can impact synapse density, spine morphology, or expression levels of synaptic proteins linked to psychiatric disease pathology.

We removed the 1-week datasets for behavior and electrophysiology from the manuscript in order to focus our study on the impact of acute restraint stress at the 24 hr time point. While we agree that these experiments would add strength to the argument that the stress adaptations, we observe in the vSUB is linked to psychiatric disease pathology, the scope of this paper is to create a foundational understanding of sex-differences in acute stress adaptations in the vSUB.

10) The authors put forth the observation that acute stress can be encoded at the synapses in a sex-specific manner, and this could be potential mechanism underlying sex-dependent susceptibility to psychiatric disorders in the introduction section (lines 51-53). However, the cited articles (#5-8) do not fully support this argument since it only pertains to a specific disorder (post-traumatic stress disorder) which makes it a weak statement. They only cite data from clinical studies. The authors should provide clinical evidence pertaining to other psychiatric disorders and preclinical evidence that acute stress can differentially induce behavioral alterations in a sex-dependent manner since the authors also make the statement that different acute stress paradigms can induce sex-specific changes in behavior in mice (lines 386-387).

We include clinical citations for acute stress disorder and post-traumatic stress disorders, both of which are major risk factors for mood disorders, suicide, and substance use disorders (Fanai and Khan, 2024). We cite preclinical finding in the results (lines 294-296) that show females are more likely to have lasting (24-hr post stress) anxiety-like behavior than males following acute stress (Freitas et al., 2014; Liu et al., 2023; Zhang et al., 2024a). To address this concern, we changed the language to “pathogenesis of stress-related disorders.” In lines 51-53.

Minor comments:

1) Throughout the results section, the authors do not mention which statistical test was used for which experiment. The P values are reported in the figure captions, but the authors should also report “F” and “df” values derived from the ANOVA analysis in the results sections. It would be helpful if the authors report the P, F, and df values from ANOVA analysis and P values from Student’s t-test in the results section. Furthermore, significant changes in EPSC amplitude were reported for Fig 3d, 4d, 5i, and 5k. Were these P values derived from ANOVA analysis or from a post-hoc test?

The statistical tests, F, df, and p values are now included in the figure legends and in reporting section of this journal. The statistical and post-hoc tests are also reported in the methods section. Changes in PSC amplitudes in Input/Output curves were determined from Sidak post- hoc comparisons following ANOVA repeated measure analysis (line 664-678).

2) The authors should perform a quality control check for all of their figure captions for the main text figures and supplementary figures. There are a couple of errors in both the main text figure captions and supplementary figure captions. For instance, for Fig 5, panel “i” refers to EPSC data in female mice but on line 1005, it says panel “k”. For Fig 6, panel “w” refers to distance travelled by female mice, but on line 1033, it says panel “x”. For Fig 7, panel (f,j)

on line 1045 is not bolded unlike the other panel designations in Fig 7 and in other figures. For Fig 8, panel “f” and “j” refers to IPSC data for sympathetic and corticosterone signaling experiments. However, it says panel “e” on line 1065 and panel “i” on line 1066. On line 1032, “P” is not italicized. There are also minor errors in the figure captions for the supplementary figures that needs to be rectified.

We apologize for these errors and thank the reviewer for bringing these to our attention. We have addressed these changes.

3) In the statistical analyses section, “P” is not capitalized but it is capitalized in the figure captions. There should be consistency in regard to the P value being capitalized or not.

We have changed a p-values to lowercase.

4) For all the figure captions (main text & supplementary), it should read as “unpaired Student’s t-test” instead of “unpaired t-test”. The “t” needs to be italicized. The authors should also state if they used repeated measures ANOVA or not in the figure captions.

Repeated measures ANOVA analysis are included in the figure captions as needed. We italicized for t-tests and included “Student’s”.

5) On lines 453-454, it should read as “All injectable solutions were freshly prepared on the day of the experiment”. At the moment, the sentence ends with “of” that makes the sentence sound incomplete.

We have made these suggested text changes (line 486).

Reviewer #3 (Remarks to the Author):

This study provides critical insights into the sex differences in stress adaptation at the synaptic and circuit levels, advancing our understanding of the mechanisms underlying stress-related psychiatric disorders like anxiety and depression. The findings have broad implications for developing sex-specific treatments for psychiatric diseases. The study investigates the sex-specific effects of acute stress on the ventral subiculum (vSUB), a region linked to stress regulation, emotional processing and psychiatric disease. Using a combination of electrophysiology, optogenetics, and behavioral assays, the authors show that stress induces distinct changes in synaptic function and neuronal circuitry in male and female mice.

The majority of these experiments are technically well performed, and the results are informative. However, there are several concerns with the logic of some of the experiments and the interpretation of the results. Specifically, the study did great work in revealing sex-specific synaptic and circuitry changes in vSUB after 1 hour acute restraint stress. However, the data does not support the conclusion that acute stress causes sex-dependent anxiety-like behavior. In its current form, the study is not suitable for publication in Nature Communications.

We thank the reviewer for appreciating that our study provides critical insights and our findings have broad implications for the treatment of sex-specific disorders.

Major concerns:

1, All studies were done in ≤ 2 month old mice (line 486, line536), which are not strictly considered as adult. According to the Jackson Laboratory, mature adult mice should be at least three months old because, although they are sexually mature by 35 days, relatively rapid maturational growth continues for most biological processes and structures until about three months. We are aware that patch-clamp experiments are more complicated in mature mice (> 3 months old). However, this relatively immature age of the mice used raised questions as to whether similar findings for behavioral outcomes would hold true in mature mice.

To address this concern, we repeated these key electrophysiological studies of basal synaptic transmission in the vSUB local circuit (vCA1-vSUB and PV-vSUB) in adult mice (PND70-77). Our findings in sexually mature PND 42-60 mice hold true for fully adult mice (Figures S1, S2). We report at both time periods that acute restraint stress enhances PV-mediated inhibition and reduces excitatory synaptic strength onto BS neurons in females. In males, the same stress protocol selectively reduces PV-mediated inhibition of BS cells. We agree with the reviewer that our electrophysiology data are collected from a wide swathe of ages (PND42-60), however, we find the reproducibility of the

phenotypes across this age range and in older (PND70-77) animals encouraging that the sex-specific synaptic adaptations that we identified here are robust and not specific to a particular time point.

2, The rationale for examining synaptic change 24 hours after acute restraint stress is unclear. Why this specific time point was chosen? Whether this synaptic change in CA1-vSUB and vSUB-aBNST (Fig 1,2 and 5) connections is transient or long lasting, for instance, for days or even weeks, is not investigated. This comes to very crucial if the authors claim the long-term behavioral change in anxiety-like behavior, which most likely need a long-lasting synaptic plasticity change or circuit modulation among the vCA1, vSUB and aBNST.

Nevertheless, the authors have considered to perform such experiments in examining PV-BS IPSCs after a week. However, whether the local microcircuit connectivity shift affects long-range vCA1-vSUB-aBNST projections, which are involved in anxiety and stress behaviors, remain unclear. Long-term changes, such as those occurring over days or weeks, should be investigated to support claims about persistent behavioral effects.

As discussed in point 9 for Reviewer 2, we removed the 1-week PV-oIPSC data and 1-week behavioral data from the manuscript and now restrict our claims to the acute period. Although it would be highly informative to have time course studies of synaptic changes in all studies examining stress effects, we largely focused on one timepoint for feasibility and to allow us to interrogate the changes more deeply. We elected the 1-day time period so that we could capture changes driven transcriptionally or translationally and we were initially unsure whether a single stressor would induce lasting changes.

3, Regarding the EZM behavioral test, the authors used white noise during habituation and experimentation. What is the sound level and frequency of this white noise? Could white noise itself be another type of stressor? Classically, anxiety tests, such as EPM, EZM or open field test, are carried out without such a background noise or even in a sound-attenuated arena.

Furthermore, the authors decide to analyze the first 5 minutes of the EZM and claimed this period as the most anxiogenic response period. However, the reference given "100. The relation between fear induced by novel stimulation and exploratory behavior - PubMed. Journal of comparative and physiological psychology 48 (1955 Aug). <https://doi.org/10.1037/h0043788>; does not support such claim. The apparatus in this paper is not an EZM and did not test anxiety. It is a misuse of a citation. The EZM was invented in 1990s, 40 years after the cited study. The authors must provide clear reasons to select only the first part of the behavior test (are results not holding afterwards?). Mouse behavior during the entire 10 minutes session should be analyzed.

We apologize for the error in citation. The correct citations for the justification of using the first 5 min of the EZM test and that 5 min is a standard total duration for the EZM procedure (Walf and Frye, 2007; Tucker and McCabe, 2017). However, to address this criticism, we now show the EZM data by the full 10 minutes. Using this analysis approach, our results are still significant across the 10-minute time period (Fig 6). The same mild (40-50 dB) white noise source and volume was used for every EZM experiment. Our behavioral room is attached to a hallway that is used by other vivarium users and the mild white noise reduces disturbances from the ambient noise, human traffic, and human conversations in the hallway. We have included the decibel level in the methods section. We did not use decibel volumes associated with increased anxiety-like behavior (Zhvanina et al., 2020; Peng et al., 2023). This is now included on line 493.

4, The argument that stress causes anxiety-like behavior in females is not compelling. Previous study showed no gender bias regarding the time spend in close arm in EPM or EZM (Tucker LB, McCabe JT. Front Behav Neurosci. 2017 doi: 10.3389/fnbeh.2017.00013). However, the female control mice seem to stay less time in close arm than male control mice (fig 6s,x). It is thus reasonable to pool fig 6q and 6s together, as the same for fig6v and 6x, to confirm whether ARS induced anxiety-like behavior in female mice, or ARS brought 'braver or more explorative' female mice back to normal behavior.

A more sophisticated experiment for such claim would need to compare anxiety-like behavior on EZM for the same cohort of mice before and after ARS, to avoid biased behavior among groups.

The males and females used in the behavioral experiments were not littermate matched nor were they all run during the same cohorts. Thus, it would be inappropriate to directly compare their behaviors. Importantly, within each sex the control and stress groups were equally distributed among cohorts and cage and littermate matched. We agree with the reviewer that littermate matching across sexes and testing both sexes equally across cohort would have been optimal, but was unfortunately not feasible

given the logistics of behavior room reservations and availability of enough pups per litter to sufficiently support littermate matching across sexes.

While we considered using a within subject design for assessing anxiety-like behavior, we were concerned that the mice would behave differently in the EZM upon repeated exposure. Although the effect of carryover testing is less clear for the EZM relative to the EPM, there is evidence that repeated exposure can meaningfully change time spent in open arms and locomotion (Cook et al., 2002; Tucker and McCabe, 2017). We now include the open field test as additional anxiety-like behavior measurements to assess the robustness of the phenotype in females (Fig S7).

5, The coordinates (AP -4.04; ML 3.4; DV -3.5) used for GCaMP7f virus injection for fiber photometry is clearly off target. The virus transduced the ST_r (subiculum, transition area), other than vSub. There are also many CA1 neurons transduced, as shown in the sampling image in Fig 6a. As BNST also receive projections from CA1, post hoc immunochemistry must be used to verify viral labeling specificity, for instance clearly showing the tip of fiber sitting above vSub. Furthermore, as there is no clear border between vCA1 and vSub, a thinner fiber 200µm will do much cleaner job than a 400µm fiber does.

Furthermore, calcium signal recordings started two weeks after implantation of a 400um fiber. It is not an ideal time as there are still inflammation going on after fiber implantation. Common procedure for such calcium activity recording is to wait at least 4 weeks. Otherwise, neurons can show abnormal hyperactivity, which can hinder proper conclusions.

Given the size of the ventral subiculum, we believe that a 400um is a reasonable choice. This is the typical fiber size for fiber photometry (smaller fibers e.g., 200 um can be used for fiber photometry, though they are more typically employed for optogenetic stimulation (Cui et al., 2013; Chen et al., 2015; Beutler et al., 2020; Cai et al., 2020; Marcus et al., 2020; Gray et al., 2021; Karigo et al., 2021; Andersen et al., 2023; Fuzesi et al., 2023; Terstege et al., 2023; Serikov et al., 2024; Tamboli et al., 2024). While dorsal subiculum may be relatively small and thus difficult to specifically capture using a larger fiber, the 400um fiber enables us to maximize the signal that we obtain, while remaining restricted to the vSub.

The coordinates used in this study were modified from those have been used previously by the laboratory to successfully target vSUB (Boxer et al., 2021; Boxer et al., 2023). Prior to electrophysiology experiments, injection coordinates are validated by fluorescence microscopy and mis-injected animals are excluded from the study. The specificity of recording from vSub and not CA1 cells is also supported by the quantification of the projections from CA1 and vSUB to BNST. We quantified the number of cells in vSUB and vCA1 (anatomically defined in the Allen Brain Explorer) and roughly 88% of aBNST-projecting vHIPP neurons are located in vSUB and ~12% are in vCA1. We include this quantification below and also include it in the manuscript (Supplementary Figure 5). Importantly, these quantifications are supported by recent reports indicating that vSUB provides the primary hippocampal output to BNST (Ding et al., 2020; Sun et al., 2023). To address the reviewer's concerns about injection targeting and fiber placement, we now clarify that every animal used for fiber photometry was examined post-hoc for the correct targeting of virus injections and fiber placement. We have included a representative coronal brain section that better shows the enrichment of GCaMP7f expression in vSUB and more clearly reveals the correct positioning of the fiber directly above vSUB (Fig. 5h).

While longer timeframes from virus injection to calcium imaging are sometimes used in research, this is not standard, and typically, not required. We cite several references, below, that perform fiber photometry recordings after only a few weeks post-injection, not the full month stated by the reviewer. Longer time periods can be advantageous for increasing expression of the fluorescent indicator, such as when calcium signal is recorded from axonal terminals; however, we found that that was not necessary in our case (see representative vSUB-aBNST images taken immediately following data acquisition). The following papers collected data from animals 2-weeks after the implantation of 400µm fibers or are published protocols papers that recommend testing animals 2-weeks after the implantation of 400µm fibers: Cui et al., *Nature*. 2013; Chen et al., *Neuron*. 2015; Beutler et al., *Elife*. 2020; Cai et al., *Elife*. 2020; Marcus et al., *Neuron*. 2020; Gray et al., *Sci Adv*. 2021; Karigo et al., *Nature*. 2021; Andersen et al., *Bio. Protoc*. 2023; Fuzesi et al., *Nat. Commun*. 2023; Terstege et al., *STAR Protoc*. 2023; Serikov et al., *STAR Protoc*. 2024; Tamboli et al., *STAR Protocol*. 2024.

6, It is necessary to consider also the elevated plus maze and open field tests to confirm the findings in EZM.

In response to the reviewer's concerns, we performed the OFT and FST (experiments see response to Reviewer 2, comment 1). EZM and EPM are similar assays to measure anxiety-like behavior, however, the interpretation of anxiety-like behavior in EPM is complicated by the open center area (Shepherd et al., 1994; Cook et al., 2001; Milner and Crabbe, 2008). Instead, we focused our efforts on the OFT, which is a different measure of anxiety and the FST to measure a different dimension of stress-induced behavioral adaptations. We observed anxiety-like behavior (without a change in locomotion) in the OFT in females after stress (Fig. S7a-c).

7, We have concerns about the variations of two cohorts of animals in figure 7e-l and fig 8e-l. If we are not mistaken, the controls in fig 7f-h and 7j-l are both injected with saline. I expect similar electrical properties of these two control cohorts. However, the values of slope, aIPSC and PPR are largely different between them. The same difference as observed in fig 8f-h vs fig 8j-l for male mice. This variation in control experiments likely indicates instability of the experimental settings, conductance measurements, or strong variability across different mouse cohorts.

These are excellent observations. It is not uncommon for differences in the absolute values of synaptic transmission to differ between distinct sets of experiments due to the variability inherent to ex vivo slice electrophysiology that can be explained by, for example, distinct experiments performed months/years apart, slight differences in solutions, and/or tissue handling. Despite best efforts to control for this inherent variability, publications that present their data in the form of absolute values have reported differences in presynaptic release (Gulyas et al., 2010), evoked synaptic transmission (Baimel et al., 2019; Copenhaver and LeGates, 2024), and even unitary currents (Lee et al., 2015). To control for the quality of cells electrophysiologically monitored, we continuously measured cell parameters by using a -5 mV test pulse - cells where the access resistance was $> 20M\Omega$ (uncompensated) and/or a exhibited a $>20\%$ change in series resistance and/or exhibited a holding current of $> +100$ pA were excluded from analysis. As is common in the field, our experimental design also controls for variability between distinct experiments as we used littermates for individual experiments and vehicle treated animals were interleaved with drug treated animals during consecutive days of data collection.

We apologize for the confusion regarding the chemical used in control conditions. AR inhibitors (phentolamine and propranolol) were solubilized in saline while the corticosterone synthesis inhibitor (metyrapone) was resuspended in DMSO. We have added these details to figures 7 and 8. Importantly, the conditions where saline was used as the control vehicle (Figs. 7f-h and 8f-h) have similar synaptic properties while conditions where DMSO was used as the control vehicle (Figs. 7j-l and 8j-l) are similar. Although commonly used as a solvent in pharmacological studies there is limited evidence suggesting that systemic administration of DMSO may alter hippocampal spine density, GABAergic transmission (Nakahiro et al., 1992) and hippocampus-dependent behaviors (Penazzi et al., 2017). We attempted to address these potential issues by including a DMSO vehicle control in all relevant experiments. We made the differences in vehicles used in control groups clearer in the figures to address this.

Additionally, there are several key experimental differences between the adrenergic blockade experiments (7f-h/8f-h) and the corticosterone blockade experiments (7j-l/8j-l). First, we used different virus preparations to perform 7/8f-h and 7/8j-l (see diagram in figures for virus details). Throughout the paper, we used AAV-DIO-ChIEF-mRuby, which depleted our stock after *in vivo* injections to complete 7/8f-h. For 7/8j-l, we used a different virus AAV-DIO-ChIEF-GFP, which we previously used (Boxer et al., 2023). Although both viruses are virtually identical except for the fluorescent protein and made concurrently, the heavy usage of the ChIEF-mRuby variant in this project and others (Boxer et al., 2021) might have introduced freeze-thaw cycles (despite our best efforts) that reduced viral titer. Thus, although the viruses changed between these experiments, they were importantly given equally across experimental group.

Minor

1, Detailed methods for fiber photometry should be documented, such as sampling rate, excitation power, calcium traces filtering, peak detection etc.

We have added the requested information in the methods section (Lines 633-649).

2, ref 17, wherever possible, please cite peer-reviewed article rather than non-peer reviewed biorxiv.

We apologize for the mistake. We now cite the peer-reviewed article as citation 19:

Boxer EE, Seng C, Lukacsovich D, Kim J, Schwartz S, Kennedy MJ, Földy C, Aoto J. Neurexin-3 defines synapse- and sex-dependent diversity of GABAergic inhibition in ventral subiculum. *Cell Rep*. 2021 Dec 7;37(10):110098. doi: 10.1016/j.celrep.2021.110098. PMID: 34879268; PMCID: PMC8763380.

3, line 208-209, “The vSUB-aBNST circuit is known to be critical for stress hormone (i.e. corticosterone) release and anxiety-like behavior,” References are missing for this claim.

We apologize for the accidental omission. The vCA1/vSUB-aBNST circuit regulates corticosterone release (Cole et al., 2022) and aspects of anxiety-like behaviors (Glangetas et al., 2017; Urien et al., 2022; Kopaeva et al., 2023). We added these citations to line 192.

4, CA1 and vCA1 are both used throughout the text, please make sure they are accurately used.

We have adjusted the text to address this concern.

References

- Andersen M, Tsopanidou A, Radovanovic T, Compere VN, Hauglund N, Nedergaard M, Kjaerby C (2023) Using Fiber Photometry in Mice to Estimate Fluorescent Biosensor Levels During Sleep. *Bio Protoc* 13:e4734. 10.21769/BioProtoc.4734
- Baimel C, McGarry LM, Carter AG (2019) The Projection Targets of Medium Spiny Neurons Govern Cocaine-Evoked Synaptic Plasticity in the Nucleus Accumbens. *Cell Rep* 28:2256-2263 e2253. 10.1016/j.celrep.2019.07.074
- Beutler LR, Corpuz TV, Ahn JS, Kosar S, Song W, Chen Y, Knight ZA (2020) Obesity causes selective and long-lasting desensitization of AgRP neurons to dietary fat. *Elife* 9. 10.7554/eLife.55909
- Boxer EE, Kim J, Dunn B, Aoto J (2023) Ventral Subiculum Inputs to Nucleus Accumbens Medial Shell Preferentially Innervate D2R Medium Spiny Neurons and Contain Calcium Permeable AMPARs. *J Neurosci* 43:1166-1177. 10.1523/JNEUROSCI.1907-22.2022
- Boxer EE, Seng C, Lukacsovich D, Kim J, Schwartz S, Kennedy MJ, Földy C, Aoto J (2021) Neurexin-3 defines synapse- and sex-dependent diversity of GABAergic inhibition in ventral subiculum. *Cell Rep* 37:110098. 10.1016/j.celrep.2021.110098
- Cai LX, Pizano K, Gundersen GW, Hayes CL, Fleming WT, Holt S, Cox JM, Witten IB (2020) Distinct signals in medial and lateral VTA dopamine neurons modulate fear extinction at different times. *Elife* 9. 10.7554/eLife.54936
- Chen Y, Lin YC, Kuo TW, Knight ZA (2015) Sensory detection of food rapidly modulates arcuate feeding circuits. *Cell* 160:829-841. 10.1016/j.cell.2015.01.033
- Cheng R, Zhang F, Li M, Wo X, Su YW, Wang W (2019) Influence of Fixation and Permeabilization on the Mass Density of Single Cells: A Surface Plasmon Resonance Imaging Study. *Front Chem* 7:588. 10.3389/fchem.2019.00588
- Cole AB, Montgomery K, Bale TL, Thompson SM (2022) What the hippocampus tells the HPA axis: Hippocampal output attenuates acute stress responses via disynaptic inhibition of CRF+ PVN neurons. *Neurobiol Stress* 20:100473. 10.1016/j.ynstr.2022.100473
- Cook MN, Williams RW, Flaherty L (2001) Anxiety-related behaviors in the elevated zero-maze are affected by genetic factors and retinal degeneration. *Behav Neurosci* 115:468-476.
- Copenhaver AE, LeGates TA (2024) Sex-Specific Mechanisms Underlie Long-Term Potentiation at HippocampusMedium Spiny Neuron Synapses in the Medial Shell of the Nucleus Accumbens. *J Neurosci* 44. 10.1523/JNEUROSCI.0100-24.2024
- Cui G, Jun SB, Jin X, Pham MD, Vogel SS, Lovinger DM, Costa RM (2013) Concurrent activation of striatal direct and indirect pathways during action initiation. *Nature* 494:238-242. 10.1038/nature11846
- Ding SL, Yao Z, Hirokawa KE, Nguyen TN, Graybuck LT, Fong O, Bohn P, Ngo K, Smith KA, Koch C, Phillips JW, Lein ES, Harris JA, Tasic B, Zeng H (2020) Distinct Transcriptomic Cell Types and Neural Circuits of the Subiculum and Prosubiculum along the Dorsal-Ventral Axis. *Cell Rep* 31:107648. 10.1016/j.celrep.2020.107648
- Fuzesi T, Rasiah NP, Rosenegger DG, Rojas-Carvajal M, Chomiak T, Daviu N, Molina LA, Simone K, Sterley TL, Nicola W, Bains JS (2023) Hypothalamic CRH neurons represent physiological memory of positive and negative experience. *Nat Commun* 14:8522. 10.1038/s41467-023-44163-5

- Glangetas C, Massi L, Fois GR, Jalabert M, Girard D, Diana M, Yonehara K, Roska B, Xu C, Luthi A, Caille S, Georges F (2017) NMDA-receptor-dependent plasticity in the bed nucleus of the stria terminalis triggers long-term anxiolysis. *Nat Commun* 8:14456. 10.1038/ncomms14456
- Glynn MW, Elmer BM, Garay PA, Liu XB, Needleman LA, El-Sabeawy F, McAllister AK (2011) MHCI negatively regulates synapse density during the establishment of cortical connections. *Nat Neurosci* 14:442-451. 10.1038/nn.2764
- Gray SR, Ye L, Ye JY, Paukert M (2021) Noradrenergic terminal short-term potentiation enables modality-selective integration of sensory input and vigilance state. *Sci Adv* 7:eabk1378. 10.1126/sciadv.abk1378
- Gulyas AI, Szabo GG, Ulbert I, Holderith N, Monyer H, Erdelyi F, Szabo G, Freund TF, Hajos N (2010) Parvalbumin-containing fast-spiking basket cells generate the field potential oscillations induced by cholinergic receptor activation in the hippocampus. *J Neurosci* 30:15134-15145. 10.1523/JNEUROSCI.4104-10.2010
- Guo F, Fan J, Liu JM, Kong PL, Ren J, Mo JW, Lu CL, Zhong QL, Chen LY, Jiang HT, Zhang C, Wen YL, Gu TT, Li SJ, Fang YY, Pan BX, Gao TM, Cao X (2024) Astrocytic ALKBH5 in stress response contributes to depressive-like behaviors in mice. *Nat Commun* 15:4347. 10.1038/s41467-024-48730-2
- Idziak A, Inavalli V, Bancelin S, Arizono M, Nagerl UV (2023) The Impact of Chemical Fixation on the Microanatomy of Mouse Organotypic Hippocampal Slices. *eNeuro* 10. 10.1523/ENEURO.0104-23.2023
- Karigo T, Kennedy A, Yang B, Liu M, Tai D, Wahle IA, Anderson DJ (2021) Distinct hypothalamic control of same- and opposite-sex mounting behaviour in mice. *Nature* 589:258-263. 10.1038/s41586-020-2995-0
- Kopaeva L, Yakimov A, Urien L, Bauer EP (2023) Chemogenetic activation of the ventral subiculum-BNST pathway reduces context fear expression. *Learn Mem* 30:164-168. 10.1101/lm.053797.123
- Lee IB, Lee E, Han NE, Slavuj M, Hwang JW, Lee A, Sun T, Jeong Y, Baik JH, Park JY, Choi SY, Kwag J, Yoon BJ (2024) Persistent enhancement of basolateral amygdala-dorsomedial striatum synapses causes compulsive-like behaviors in mice. *Nat Commun* 15:219. 10.1038/s41467-023-44322-8
- Lee SH, Ledri M, Toth B, Marchionni I, Henstridge CM, Dudok B, Kenesei K, Barna L, Szabo SI, Renkecz T, Oberoi M, Watanabe M, Limoli CL, Horvai G, Soltesz I, Katona I (2015) Multiple Forms of Endocannabinoid and Endovanilloid Signaling Regulate the Tonic Control of GABA Release. *J Neurosci* 35:10039-10057. 10.1523/JNEUROSCI.4112-14.2015
- Linders LE, Patrikiou L, Soiza-Reilly M, Schut EHS, van Schaffelaar BF, Boger L, Wolterink-Donselaar IG, Luijendijk MCM, Adan RAH, Mezei FJ (2022) Stress-driven potentiation of lateral hypothalamic synapses onto ventral tegmental area dopamine neurons causes increased consumption of palatable food. *Nat Commun* 13:6898. 10.1038/s41467-022-34625-7
- Liu WZ, Zhang WH, Zheng ZH, Zou JX, Liu XX, Huang SH, You WJ, He Y, Zhang JY, Wang XD, Pan BX (2020) Identification of a prefrontal cortex-to-amygdala pathway for chronic stress-induced anxiety. *Nat Commun* 11:2221. 10.1038/s41467-020-15920-7
- Marcus DJ, Bedse G, Gaulden AD, Ryan JD, Kondev V, Winters ND, Rosas-Vidal LE, Altemus M, Mackie K, Lee FS, Delpire E, Patel S (2020) Endocannabinoid Signaling Collapse Mediates Stress-Induced Amygdalo-Cortical Strengthening. *Neuron* 105:1062-1076 e1066. 10.1016/j.neuron.2019.12.024
- Milner LC, Crabbe JC (2008) Three murine anxiety models: results from multiple inbred strain comparisons. *Genes Brain Behav* 7:496-505. 10.1111/j.1601-183X.2007.00385.x
- Nakahiro M, Arakawa O, Narahashi T, Ukai S, Kato Y, Nishinuma K, Nishimura T (1992) Dimethyl sulfoxide (DMSO) blocks GABA-induced current in rat dorsal root ganglion neurons. *Neurosci Lett* 138:5-8. 10.1016/0304-3940(92)90459-k
- Richter KN et al. (2018) Glyoxal as an alternative fixative to formaldehyde in immunostaining and super-resolution microscopy. *EMBO J* 37:139-159. 10.15252/embj.201695709
- Rodrigues D, Jacinto L, Falcao M, Castro AC, Cruz A, Santa C, Manadas B, Marques F, Sousa N, Monteiro P (2022) Chronic stress causes striatal disinhibition mediated by SOM-interneurons in male mice. *Nat Commun* 13:7355. 10.1038/s41467-022-35028-4
- Serikov A, Martsishevskaya I, Shin W, Kim J (2024) Protocol for in vivo dual-color fiber photometry in the mouse thalamus. *STAR Protoc* 5:102931. 10.1016/j.xpro.2024.102931
- Shepherd JK, Grewal SS, Fletcher A, Bill DJ, Dourish CT (1994) Behavioural and pharmacological characterisation of the elevated "zero-maze" as an animal model of anxiety. *Psychopharmacology (Berl)* 116:56-64. 10.1007/BF02244871
- Tamboli S, Topolnik D, Radhakrishnan R, Veilleux-Lemieux D, Topolnik L (2024) Protocol for synchronized wireless fiber photometry and video recordings in rodents during behavior. *STAR Protoc* 5:103407. 10.1016/j.xpro.2024.103407

- Terstege DJ, Dawson M, Jamani NF, Tsutsui M, Epp JR, Sargin D (2023) Protocol for the integration of fiber photometry and social behavior in rodent models. STAR Protoc 4:102689. 10.1016/j.xpro.2023.102689
- Urien L, Cohen S, Howard S, Yakimov A, Nordlicht R, Bauer EP (2022) Aversive Contexts Reduce Activity in the Ventral Subiculum- BNST Pathway. Neuroscience 496:129-140. 10.1016/j.neuroscience.2022.06.019
- Zhang X, Liu Y, Hong X, Li X, Meshul CK, Moore C, Yang Y, Han Y, Li WG, Qi X, Lou H, Duan S, Xu TL, Tong X (2021) NG2 glia-derived GABA release tunes inhibitory synapses and contributes to stress-induced anxiety. Nat Commun 12:5740. 10.1038/s41467-021-25956-y
- Zhang Y, Shen J, Xie F, Liu Z, Yin F, Cheng M, Wang L, Cai M, Herzog H, Wu P, Zhang Z, Zhan C, Liu T (2024) Feedforward inhibition of stress by brainstem neuropeptide Y neurons. Nat Commun 15:7603. 10.1038/s41467-024-51956-9

Reviewer #1 (Remarks to the Author):

The authors have responded to many of my concerns, and I commend them for doing challenging experiments to demonstrate that the effect of stress was age-independent. However despite many improvements to the manuscript, the fact remains that an n of 3 or 4 animals is far too low for a study where the variable is something that happens to the animal itself. The fact that other papers have been published in this journal with similar ns does not change my stance on this. Had I reviewed these papers I would have also seen them as underpowered, if I encountered them as a reader I would consider them premature findings and would hesitate to use them to inform my own work. Without increasing the n's in this study to an appropriate sample size I do not think it reaches the standards of this journal.

We appreciate the reviewer's desire that results published in journals such as Nature Communications to be rigorous and reproducible, especially in light of the crisis on reproducibility in biology. Our approach to ensuring that our results meet the strictest standards of reproducibility is to apply commonly used and rigorous statistical power calculations to ensure that our experiments are sufficiently powered. Power analysis is the gold standard and most accepted method for determining sample sizes required to achieve statistical significance. As discussed below, given the effect size that we observed for our completed experiments, we have exceeded the estimated sample sizes necessary to achieve a minimum of 95% power, the standard for achieving statistical significance. Importantly, the use of a minimum of 3 independent animals and presenting cells as individual data points has been, and still is, the gold standard for slice electrophysiology for decades that has been widely implemented across the addiction, stress and sex hormone/innate behavior fields. Therefore, our animal numbers are in line with the modern expectations for these experiments. Together, following the rigorous statistical standards in the field, our sample sizes all result in at least 95% power, indicating that we have met or exceeded the conventional metrics for rigor and reproducibility.

Reviewer #1 states: "However despite many improvements to the manuscript, the fact remains that an n of 3 or 4 animals is far too low for a study where the variable is something that happens to the animal itself."

Given the effect sizes of the sexually dimorphic synaptic phenotypes observed after acute stress, we respectfully disagree. We ran a post-hoc power analysis (G*Power; $\alpha = 0.05$, and $1-\beta = 0.95$) using our average calculated effect size of 1.42 (ranging between 1.3 to 1.9) from the main electrophysiology figures and determined the average estimated sample size is ~12 cells. In all cases where significance was detected, we exceeded the predicted sample size and achieved a power of ~0.96-0.98. We did this to collect roughly the same number of cells per animal, such that we are comfortable knowing that the collected data reflect the manifested synaptic adaptations in each animal and that each cell carries equal weight in the analysis.

Further, a minimum of 3-4 animals has been and continues to be the gold standard to assess synaptic adaptations driven by "variables that happens to the animal itself" across multiple areas of neuroscience. These studies have been published in high-profile peer-reviewed journals and include, but are not limited to the fields of addiction, sex hormones/innate behavior, and stress. In addiction studies, where non-contingent and contingent psychostimulants are appreciated to produce variable outcomes, papers routinely present their data as individual cells collected from 3-4 animals (Thomas et al., 2001; Britt et al., 2012; Creed et al., 2016; Zhu et al., 2017; Xin et al., 2019; Lefevre et al., 2020; Giannotti et al., 2021; Gong et al., 2021; Li et al., 2021; Inbar et al., 2022; Pomrenze et al., 2022; Gangal et al., 2023). More recently, electrophysiology has been employed to measure the impact of circulating sex hormones on circuits that control innate behaviors (Stagkourakis et al., 2020; Hernandez-Vivanco et al., 2025). To further add to the list of stress studies that used a similar number of animals published in high profile journals (including Nature Portfolio Journals): (Joffe et al., 2019; Liu et al., 2020; Stagkourakis et al., 2020; Bartsch et al., 2021; Pignatelli et al., 2021; Cole et al., 2022; Joffe et al., 2022; Pomrenze et al., 2022; Rodrigues et al., 2022; Zhao et al., 2022; Guo et al., 2024; Lee et al., 2024; Lucantonio et al., 2025), (Lucantonio et al., 2025), two new articles were recently published in Neuron and Nature

Communications: (Lucantonio et al., 2025; van Doeselaar et al., 2025) that also use similar animal numbers as in our study.

Reviewer #2 (Remarks to the Author):

This study provided a detailed and comprehensive analysis of the microcircuit organization in the vSUB. The authors carefully addressed each of the 10 previous major concerns and provided additional analyses and experimental data. These new data strengthened the observations of anxiety-like and depressive behaviors. The newly added analyses provided additional clarification on the PV-BS inhibition effects in both control and stress groups. Finally, the authors carefully addressed all of the minor comments with revisions and expanded literature citations.

We thank the reviewer for their positive comments.

Reviewer #3 (Remarks to the Author):

We acknowledge the authors' efforts in providing additional analyses to support their hypothesis. While most of our concerns have been addressed, the critical issue regarding the age of the experimental mice remains unresolved. The authors refer to their subjects as "sexually mature adolescent mice" (PND42-60) or "adult mice" (PND70-77), citing Brust et al., 2015, from *Frontiers in Zoology* (which is a review other than research article systematically examine sexual development). However, we still disagree with this classification.

It is well-established among researchers working with laboratory mice that animals aged PND42-60 are still undergoing rapid growth. During this period, significant changes occur in body size, skeletal structure and brain development. For instance, a mouse at PND42 weighs approximately 70% of its weight at PND90. The extent of their sexual maturity at this stage is questionable, or at least not robust and being debated. The research community more commonly considers mice to be adults at around 3 months of age (PND90 or older), other than PND70-77 in this study.

Another concern is male and female have slightly different development timeframe, so we do believe a time point at fully adult age is necessary. Findings at adolescent age PND42 might be also influenced by neurodevelopment factors.

Given that the authors aim to address 'sexually dimorphic' questions, it is essential to use non-questionable, well-accepted adult ages. We believe that using adult mice aged 3-6 months is necessary to address scientific questions involving sex-specific changes, neural circuits and behavior.

We recommend that the authors replicate their key findings using well-accepted adult mice aged 3-6 months to substantiate their claims. Sex-specific studies should involve clearly recognized adult subjects rather than adolescent or early adulthood mice. We acknowledge that certain techniques, such as patch-clamp recordings, are challenging in 3-6-month-old mice and may not be necessary. However, behavioral assessments, neural circuit tracing and calcium imaging should be conducted in mice within this mature adult age range.

This study has the potential to make a significant contribution to the research field. However, the age of the research subjects currently undermines the strength of the conclusions. We hope the authors can provide robust data from appropriately aged subjects to support their claims.

To address the reviewer's concern regarding the developmental discrepancies, we are happy to highlight and reflect in the manuscript that our analyses occurred over a broad range of developmental periods ranging from sexually-mature to late adolescent animals. Our study within this age range does not diminish its significant contribution to the field as other influential and high impact papers studying sex-differences used adolescent rats (Huang and Woolley, 2012; Tabatadze et al., 2015) and mice (Muir et al., 2024). There is a significant and growing need to understand how acute stress drives sexually dimorphic changes in brain function in adolescents (Ordaz and Luna, 2012; Seo et al., 2017; Yohn and

Blendy, 2017; Kim et al., 2021; Sisk and Gee, 2022). Understanding how acute stress alters the behavior of human adolescents is topical and important.

We strongly believe that our rigorous analyses that identify robust sexually dimorphic functional adaptations in response to acute stress at PND42-60 and PND70-77 strengthens the veracity of our findings. Further, the age-range at which the electrophysiological and *in vivo* fiber photometry experiments occurred includes and extends beyond the generally accepted onset window (PND42-PND56, per Jackson Labs) of sexual maturity in mice. Further, the nearly identical functional phenotypes at PND42-60 and PND70-77 argues against the notion that the slightly different developmental timeframe between male and female mice influences our analyses. One could always continue testing at different ages to assess whether phenotypes are sensitive to the developmental time windows in which they are conducted, however our manuscript already includes two fully-powered timepoints (powered for both males and females), thus we believe that we have already gone above and beyond what is standard. We believe the best way to address this is to modify the text to specifically point out the age windows used.

We appreciate that the Reviewer #3 acknowledges the considerable difficulty of performing slice electrophysiology in older (3-6 month) animals. We feel that the strength of our manuscript is our ability to assess synapse function and correlate it with animal behavior and *in vivo* fiber photometry. Without the complementary electrophysiology in vSUB circuitry, we feel that animal behavior and fiber photometry in adults is too correlative.

References

- Bartsch JC, von Cramon M, Gruber D, Heinemann U, Behr J (2021) Stress-Induced Enhanced Long-Term Potentiation and Reduced Threshold for N-Methyl-D-Aspartate Receptor- and beta-Adrenergic Receptor-Mediated Synaptic Plasticity in Rodent Ventral Subiculum. *Front Mol Neurosci* 14:658465. 10.3389/fnmol.2021.658465
- Britt JP, Benaliouad F, McDevitt RA, Stuber GD, Wise RA, Bonci A (2012) Synaptic and behavioral profile of multiple glutamatergic inputs to the nucleus accumbens. *Neuron* 76:790-803. 10.1016/j.neuron.2012.09.040
- Cole AB, Montgomery K, Bale TL, Thompson SM (2022) What the hippocampus tells the HPA axis: Hippocampal output attenuates acute stress responses via disinaptic inhibition of CRF+ PVN neurons. *Neurobiol Stress* 20:100473. 10.1016/j.ynstr.2022.100473
- Creed M, Ntamati NR, Chandra R, Lobo MK, Luscher C (2016) Convergence of Reinforcing and Anhedonic Cocaine Effects in the Ventral Pallidum. *Neuron* 92:214-226. 10.1016/j.neuron.2016.09.001
- Gangal H, Xie X, Huang Z, Cheng Y, Wang X, Lu J, Zhuang X, Eshoh A, Huang Y, Chen R, Smith LN, Smith RJ, Wang J (2023) Drug reinforcement impairs cognitive flexibility by inhibiting striatal cholinergic neurons. *Nat Commun* 14:3886. 10.1038/s41467-023-39623-x
- Giannotti G, Gong S, Fayette N, Heinsbroek JA, Orfila JE, Herson PS, Ford CP, Peters J (2021) Extinction blunts paraventricular thalamic contributions to heroin relapse. *Cell Rep* 36:109605. 10.1016/j.celrep.2021.109605
- Gong S, Fayette N, Heinsbroek JA, Ford CP (2021) Cocaine shifts dopamine D2 receptor sensitivity to gate conditioned behaviors. *Neuron* 109:3421-3435 e3425. 10.1016/j.neuron.2021.08.012
- Guo F, Fan J, Liu JM, Kong PL, Ren J, Mo JW, Lu CL, Zhong QL, Chen LY, Jiang HT, Zhang C, Wen YL, Gu TT, Li SJ, Fang YY, Pan BX, Gao TM, Cao X (2024) Astrocytic ALKBH5 in stress response contributes to depressive-like behaviors in mice. *Nat Commun* 15:4347. 10.1038/s41467-024-48730-2
- Hernandez-Vivanco A, de la Vega-Ruiz R, Montes-Mellado A, Azcoitia I, Mendez P (2025) Activational and organizational effects of sex hormones on hippocampal inhibitory neurons. *J Neurosci*. 10.1523/JNEUROSCI.1764-24.2025

- Huang GZ, Woolley CS (2012) Estradiol acutely suppresses inhibition in the hippocampus through a sex-specific endocannabinoid and mGluR-dependent mechanism. *Neuron* 74:801-808. 10.1016/j.neuron.2012.03.035
- Inbar K, Levi LA, Kupchik YM (2022) Cocaine induces input and cell-type-specific synaptic plasticity in ventral pallidum-projecting nucleus accumbens medium spiny neurons. *Neuropsychopharmacology* 47:1461-1472. 10.1038/s41386-022-01285-6
- Joffe ME, Santiago CI, Engers JL, Lindsley CW, Conn PJ (2019) Metabotropic glutamate receptor subtype 3 gates acute stress-induced dysregulation of amygdalo-cortical function. *Mol Psychiatry* 24:916-927. 10.1038/s41380-017-0015-z
- Joffe ME, Maksymetz J, Luschinger JR, Dogra S, Ferranti AS, Luessen DJ, Gallinger IM, Xiang Z, Branthwaite H, Melugin PR, Williford KM, Centanni SW, Shields BC, Lindsley CW, Calipari ES, Siciliano CA, Niswender CM, Tadross MR, Winder DG, Conn PJ (2022) Acute restraint stress redirects prefrontal cortex circuit function through mGlu(5) receptor plasticity on somatostatin-expressing interneurons. *Neuron* 110:1068-1083 e1065. 10.1016/j.neuron.2021.12.027
- Kim S, Gacek SA, Mocchi MM, Redei EE (2021) Sex-Specific Behavioral Response to Early Adolescent Stress in the Genetically More Stress-Reactive Wistar Kyoto More Immobile, and Its Nearly Isogenic Wistar Kyoto Less Immobile Control Strain. *Front Behav Neurosci* 15:779036. 10.3389/fnbeh.2021.779036
- Lee IB, Lee E, Han NE, Slavuj M, Hwang JW, Lee A, Sun T, Jeong Y, Baik JH, Park JY, Choi SY, Kwag J, Yoon BJ (2024) Persistent enhancement of basolateral amygdala-dorsomedial striatum synapses causes compulsive-like behaviors in mice. *Nat Commun* 15:219. 10.1038/s41467-023-44322-8
- Lefevre EM, Pisansky MT, Toddes C, Baruffaldi F, Pravetoni M, Tian L, Kono TJY, Rothwell PE (2020) Interruption of continuous opioid exposure exacerbates drug-evoked adaptations in the mesolimbic dopamine system. *Neuropsychopharmacology* 45:1781-1792. 10.1038/s41386-020-0643-x
- Li Y, Simmler LD, Van Zessen R, Flakowski J, Wan JX, Deng F, Li YL, Nautiyal KM, Pascoli V, Luscher C (2021) Synaptic mechanism underlying serotonin modulation of transition to cocaine addiction. *Science* 373:1252-1256. 10.1126/science.abi9086
- Liu WZ, Zhang WH, Zheng ZH, Zou JX, Liu XX, Huang SH, You WJ, He Y, Zhang JY, Wang XD, Pan BX (2020) Identification of a prefrontal cortex-to-amygdala pathway for chronic stress-induced anxiety. *Nat Commun* 11:2221. 10.1038/s41467-020-15920-7
- Lucantonio F, Roeglin J, Li S, Lu J, Shi A, Czerpaniak K, Fiocchi FR, Bontempi L, Shields BC, Zarate CA, Jr., Tadross MR, Pignatelli M (2025) Ketamine rescues anhedonia by cell-type- and input-specific adaptations in the nucleus accumbens. *Neuron*. 10.1016/j.neuron.2025.02.021
- Muir J, Iyer ES, Tse YC, Sorensen J, Wu S, Eid RS, Cvetkovska V, Wassef K, Gostlin S, Vitaro P, Spencer NJ, Bagot RC (2024) Sex-biased neural encoding of threat discrimination in nucleus accumbens afferents drives suppression of reward behavior. *Nat Neurosci* 27:1966-1976. 10.1038/s41593-024-01748-7
- Ordaz S, Luna B (2012) Sex differences in physiological reactivity to acute psychosocial stress in adolescence. *Psychoneuroendocrinology* 37:1135-1157. 10.1016/j.psyneuen.2012.01.002
- Pignatelli M, Tejada HA, Barker DJ, Bontempi L, Wu J, Lopez A, Palma Ribeiro S, Lucantonio F, Parise EM, Torres-Berrio A, Alvarez-Bagnarol Y, Marino RAM, Cai ZL, Xue M, Morales M, Tamminga CA, Nestler EJ, Bonci A (2021) Cooperative synaptic and intrinsic plasticity in a disynaptic limbic circuit drive stress-induced anhedonia and passive coping in mice. *Mol Psychiatry* 26:1860-1879. 10.1038/s41380-020-0686-8
- Pomrenze MB, Cardozo Pinto DF, Neumann PA, Llorach P, Tucciarone JM, Morishita W, Eshel N, Heifets BD, Malenka RC (2022) Modulation of 5-HT release by dynorphin mediates social deficits during opioid withdrawal. *Neuron* 110:4125-4143 e4126. 10.1016/j.neuron.2022.09.024
- Rodrigues D, Jacinto L, Falcao M, Castro AC, Cruz A, Santa C, Manadas B, Marques F, Sousa N, Monteiro P (2022) Chronic stress causes striatal disinhibition mediated by SOM-interneurons in male mice. *Nat Commun* 13:7355. 10.1038/s41467-022-35028-4
- Seo D, Ahluwalia A, Potenza MN, Sinha R (2017) Gender differences in neural correlates of stress-induced anxiety. *J Neurosci Res* 95:115-125. 10.1002/jnr.23926

- Sisk LM, Gee DG (2022) Stress and adolescence: vulnerability and opportunity during a sensitive window of development. *Curr Opin Psychol* 44:286-292. 10.1016/j.copsyc.2021.10.005
- Stagkourakis S, Spigolon G, Liu G, Anderson DJ (2020) Experience-dependent plasticity in an innate social behavior is mediated by hypothalamic LTP. *Proc Natl Acad Sci U S A* 117:25789-25799. 10.1073/pnas.2011782117
- Tabatadze N, Huang G, May RM, Jain A, Woolley CS (2015) Sex Differences in Molecular Signaling at Inhibitory Synapses in the Hippocampus. *J Neurosci* 35:11252-11265. 10.1523/JNEUROSCI.1067-15.2015
- Thomas MJ, Beurrier C, Bonci A, Malenka RC (2001) Long-term depression in the nucleus accumbens: a neural correlate of behavioral sensitization to cocaine. *Nat Neurosci* 4:1217-1223. 10.1038/nn757
- van Doeselaar L, Abromeit A, Stark T, Menegaz D, Ballmann M, Mitra S, Yang H, Rehawi G, Huettl RE, Bordes J, Narayan S, Harbich D, Deussing JM, Rammes G, Czisch M, Knauer-Arloth J, Eder M, Lopez JP, Schmidt MV (2025) FKBP51 in glutamatergic forebrain neurons promotes early life stress inoculation in female mice. *Nat Commun* 16:2529. 10.1038/s41467-025-57952-x
- Xin W, Mironova YA, Shen H, Marino RAM, Waisman A, Lamers WH, Bergles DE, Bonci A (2019) Oligodendrocytes Support Neuronal Glutamatergic Transmission via Expression of Glutamine Synthetase. *Cell Rep* 27:2262-2271 e2265. 10.1016/j.celrep.2019.04.094
- Yohn NL, Blendy JA (2017) Adolescent Chronic Unpredictable Stress Exposure Is a Sensitive Window for Long-Term Changes in Adult Behavior in Mice. *Neuropsychopharmacology* 42:1670-1678. 10.1038/npp.2017.11
- Zhao D, Wang D, Wang W, Dai J, Cui M, Wu M, Liu C, Liu J, Meng F, Wang K, Hu F, Liu D, Qiu C, Li W, Li C (2022) The altered sensitivity of acute stress induced anxiety-related behaviors by modulating insular cortex-paraventricular thalamus-bed nucleus of the stria terminalis neural circuit. *Neurobiol Dis* 174:105890. 10.1016/j.nbd.2022.105890
- Zhu F, Wu Q, Li J, Grycel K, Liu B, Sun X, Zhou L, Jiao R, Song R, Khan YM, Wang Q, Wang L, Xu Y, Li J, Zhang B, Zhou Z (2017) A single dose of cocaine potentiates glutamatergic synaptic transmission onto locus coeruleus neurons. *Cell Calcium* 67:11-20. 10.1016/j.ceca.2017.07.007